# On the necessity of adaptive regularisation: Optimal anytime online learning on $\ell_p$-balls

**Emmeran Johnson**
Department of Mathematics
Imperial College London
emmeran.johnson17@imperial.ac.uk

**David Martínez-Rubio**
Signal Theory and Communications Department
Carlos III University of Madrid
dmrubio@ing.uc3m.es

**Ciara Pike-Burke**
Department of Mathematics
Imperial College London
c.pike-burke@imperial.ac.uk

**Patrick Rebeschini**
Department of Statistics
University of Oxford
patrick.rebeschini@stats.ox.ac.uk

## Abstract

We study online convex optimisation on $\ell_p$-balls in $\mathbb{R}^d$ for $p > 2$. While always sub-linear, the optimal regret exhibits a shift between the high-dimensional setting ($d > T$), when the dimension $d$ is greater than the time horizon $T$ and the low-dimensional setting ($d \le T$). We show that Follow-the-Regularised-Leader (FTRL) with time-varying regularisation which is adaptive to the dimension regime is anytime optimal for all dimension regimes. Motivated by this, we ask whether it is possible to obtain anytime optimality of FTRL with fixed non-adaptive regularisation. Our main result establishes that for separable regularisers, adaptivity in the regulariser is necessary, and that any fixed regulariser will be sub-optimal in one of the two dimension regimes. Finally, we provide lower bounds which rule out sub-linear regret bounds for the linear bandit problem in sufficiently high-dimension for all $\ell_p$-balls with $p \ge 1$.

## 1 Introduction

We study Online Convex Optimisation (OCO) [16, 45], a sequential game where in each round $t = 1, \ldots, T$, a learner selects a point $x_t$ in a convex set $V \subset \mathbb{R}^d$ and suffers a convex loss $\ell_t(x_t)$, the full loss $\ell_t$ is then revealed to the learner before the next point $x_{t+1}$ is selected. The learner competes against the best fixed point in hindsight and aims to minimise its regret against this competitor: $R_T = \sum_{t=1}^{T} \ell_t(x_t) - \min_{u \in V} \sum_{t=1}^{T} \ell_t(u)$. Optimal performance is known to depend on parameters of the problem such as the geometry of the set $V$ and constraints on the losses $\ell_t$ [28, 23, 25, 42].

We consider the setting where the action set $V = \left\{ x \in \mathbb{R}^d : \|x\|_p \le 1 \right\}$ is an $\ell_p$-ball in $d$-dimensional space with $p > 2$ and losses are $L$-Lipschitz with respect to $\|\cdot\|_p$ (ensuring $R_T \le 2LT$). The study of $\ell_p$-geometries with $p > 2$ has been the focus of many works in optimisation [10, 28, 18, 24, 25] as it covers sets with varying levels of curvature, offering insights for more general spaces.

In this work, we study the behaviour of the Follow-The-Regularised-Leader (FTRL) (and Online Mirror Descent (OMD)) family of algorithms in achieving anytime optimal regret guarantees. Anytime refers to the absence of knowledge of the time horizon $T$. We focus on two regimes: the high-dimensional setting where $d > T$ and the low-dimensional setting where $d \le T$. For $\ell_p$-balls with $p > 2$, the optimal regret exhibits a shift from the high-dimensional setting to the low-dimensional setting (see Table 1). If $T$ is unknown, then so is the dimension regime when the game begins. We show that anytime optimality can be achieved with FTRL by using adaptive

39th Conference on Neural Information Processing Systems (NeurIPS 2025).

regularisation that in early high-dimensional rounds uses a uniformly-convex regulariser of degree $p$ (see Definition 2.2) and switches to a strongly-convex regulariser in round $t_0 \approx d$. Despite achieving the anytime optimal regret through adaptive regularisation, it remains an open question whether this can be achieved through OMD or FTRL with a single fixed regulariser. This would be desirable since it would provide algorithmic simplicity as well as an understanding of how to appropriately regularise $\ell_p$-balls across all dimension-regimes simultaneously. Therefore, we aim to answer the following question:

*Can OMD or FTRL with a fixed regulariser be anytime optimal for OCO on $\ell_p$-balls with $p > 2$ ?*

To answer this question, it is natural to first consider the regularisers that are optimal in one of the dimension regimes. However, we give algorithmic-dependent lower bounds that show that these are not anytime optimal (Proposition 4.1, Proposition 4.5). More generally, we also show that any strongly-convex regulariser is provably sub-optimal in the high dimensional setting (Theorem 4.2).

We then turn to **our main result** which provides a negative answer to the above question for separable regularisers (that separate additively over dimension: $\psi(x) = \sum_{i=1}^{d} g_i(x_i)$). The result (Theorem 4.6) states that a separable regulariser (with OMD or FTRL) cannot be anytime optimal. This also establishes that the adaptive regularisation used in our procedure to achieve anytime optimality is necessary for separable regularisers. The class of separable regularisers covers a wide range of regularisers including all of the form $\|x\|_r^r$ for any $r \geq 1$ which are commonly used in OMD and FTRL [10, 28, 43]. Moreover, the result holds for any separate *coordinate-wise* decreasing step-sizes, showing that the widely used diagonal versions of Adagrad-style algorithms [12] are also anytime sub-optimal and emphasising the relevance of this result on practical methods. As far as we are aware, results on the failure of fixed regularization are novel in online learning. However, algorithmic specific lower bounds (like the ones we have for specific regularizers in Proposition 4.1 and Proposition 4.5) have appeared in prior work (e.g. Theorems 3 & 4 in [37]).

For any online learning problem, the learner may not know if the game will end in low or high dimensions, and designing optimal procedures in both cases is important. Our results show that achieving this is not straightforward for $\ell_p$-balls, highlighting that this problem should not be overlooked in online learning more broadly and pointing out that the question on universality of mirror descent started in prior works [39] is not completely answered (see discussion in Section 4.3). We note that the $\ell_p/\ell_q$ setting (i.e. $V = \mathcal{B}_p, \partial\ell_t(x) \subset \mathcal{B}_q$) is relevant in the literature, e.g., see the open problem in [17]. Our work highlights the difficulty of this setting for $p = q > 2$.

Finally, we consider the linear bandit problem where only the loss evaluated at $x_t$ is observed (Section 5). We would like to approach this problem in the same way as the full-information problem by characterising the algorithms achieving optimal regret guarantees across all dimension regimes. However, while sub-linear regret bounds have been established in the low-dimensional setting by [25], we show that the bandit setting is fundamentally different and much more difficult since sub-linear regret bounds become unachievable when the dimension is large enough (Theorem 5.1).

Table 1: Optimal regret rates for OCO on $\ell_p$-balls ($p > 2$) from known results. [28] show the regret bound of $O(LT^{1/2}d^{1/2-1/p})$ achieved by OMD using a strongly-convex regulariser is optimal for $d = O(T)$. When $d \gg T$ this bound is vacuous since $R_T \leq 2LT$. For $d > T$, OMD with a uniformly-convex regulariser of degree $p$ guarantees a regret of $O(LT^{1-1/p})$ [40] and is optimal [34, 18, 17, 40]. These guarantees can also be achieved with FTRL using the same regularisers [36, 4].

|  | $d \leq T$ (low-dim) | $d > T$ (high-dim) |
| --- | --- | --- |
| Optimal Regret Rate | $L\sqrt{Td^{1-2/p}}$ | $LT^{1-1/p}$ |
| Regularisation (OMD or FTRL) | Strongly-convex | Uniformly-convex |

## 1.1 Contributions

We highlight our contributions for OCO on $\ell_p$-balls for $p > 2$ below. Note that the case $p \in [1, 2]$ is already understood across all dimension regimes and does not present shifts in the rate of regret with

respect to the time horizon $T$ unlike in the $p > 2$ case, see Remark 2.1, [28]. We focus here on FTRL but the results also hold for OMD, and we include these in Appendix H.

- We consider anytime bounds where the time horizon $T$ is not known in advance and show *FTRL with adaptive regularisation achieves anytime optimality* (Theorem 3.1).
- We establish algorithmic-specific lower bounds for instances of FTRL that show that the fixed regulariser achieving optimality in low-dimensions ($\|x\|_2^2$) is provably sub-optimal in high-dimensions (Proposition 4.1), and the fixed regulariser achieving optimality in high-dimensions ($\|x\|_p^p$) is provably sub-optimal in low-dimensions (Proposition 4.5). We also provide a more general result on the *sub-optimality of any strongly-convex regulariser in high-dimensions* (Theorem 4.2).
- **Our main result:** For **separable regularisers**, or regularisers that are within a multiplicative constant of these, we show that **adaptive regularisation for OMD or FTRL is necessary to achieve anytime optimality** (Theorem 4.6), ruling out the existence of a single anytime optimal separable regulariser.
- In Section 2, we connect results from the literature to *fully characterise optimality for $\ell_p$-balls with $p > 2$ across all dimension regimes* . In particular, we highlight that FTRL with a strongly-convex regulariser achieves the optimal regret in low-dimensions and FTRL with a uniformly-convex regulariser of degree $p$ achieves the optimal regret in high-dimensions.
- Finally, for bandit feedback where only $\ell_t(x_t)$ is revealed to the learner in each round $t$ instead of the full loss, we establish lower bounds for all convex $\ell_p$-balls ($p \geq 1$) showing *any linear bandit learner suffers linear regret when the dimension is large enough* (Theorem 5.1).

We also include some simulations in Appendix A which validate some of our theoretical findings.

## 1.2 Related works

**High-dimensional Online Learning:** The setting where $d > T$ has been considered mostly for the stochastic linear bandit problem [20, 30, 7, 29, 26], where the stochastic linear refers to the losses being fixed and linear but observed with i.i.d. noise as opposed to the harder fully adversarial nature of our setting. Beyond stochastic linear bandits, little attention has been given to the high-dimensional setting. Although the high-dimensional setting was not explicitly studied in [28], the results provided for OCO on $\ell_p$-balls for $p \in [1, 2]$ fully characterise regret optimality across all dimensions (see Remark 2.1). Similarly, the results we present for the high-dimensional case of $\ell_p$-balls with $p > 2$ in Section 2.3 follow from prior work not explicitly studying the high-dimensional setting [40, 4].

**Uniform Convexity of functions** (Definition 2.2, see also [46, 1, 35, 24]) is the key ingredient to obtain dimension-independent regret bounds in high-dimensions. Uniformly-convex functions have been considered as regularisers for offline [10] and online optimisation [40, 4], and also as objectives [22] and losses [39]. **Uniform convexity of sets**[1] [8, 19, 2] allows for interpolation of the set curvature between strong convexity and absence of curvature. In optimisation, curvature of the action set such as strong convexity typically leads to accelerated convergence rates [21, 33, 14, 32]. Uniformly-convex sets then allow interpolation between the faster rates of strongly-convex sets and the slower rates of sets without curvature [11, 23, 25, 42] (see [24] for an overview). In this work, we consider a natural class of uniformly-convex sets, $\ell_p$-balls for $p > 2$. These balls also interpolate between strongly-convexity ($p = 2$) and absence of curvature ($p = \infty$), and in high dimensions we recover a connection between curvature and faster rates ($T^{1-1/p}$), which is absent in low-dimensions if we consider the dimension as fixed where the rate is $O(\sqrt{T})$ for all values of $p \geq 2$.

## 1.3 Notation

We use the following notation: $r_\star$ is the dual of $r \geq 1$ satisfying $1/r + 1/r_\star = 1$, $\|x\|_r = \left(\sum_{i=1}^d |x_i|^r\right)^{1/r}$ denotes the $\ell_r$ norm, $\|x\|_\star$ is the dual norm of $\|x\|$, $\phi_r(x) = \frac{1}{r}\|x\|_r^r$ for $r \geq 2$, $x_{t,i}$ denotes the $i$-th entry of a vector $x_t$ with a time index $t$, $e_i$ denotes the $i$-th canonical basis vector, and $\partial f(x)$ is the set of sub-gradients of a function $f$ at $x$. For a function $f : \mathbb{R} \to \mathbb{R}$, we write $f(x) = O(g(x))$ (resp. $\Omega(g(x))$) where $\exists c > 0, N \in \mathbb{R}_{>0}$ such that for all $x > N$, $f(n) \leq cg(n)$ (resp. $f(n) \geq cg(n)$). We use $\widetilde{O}$ and $\widetilde{\Omega}$ when we ignore logarithmic factors.

---

[1]Uniform convexity for sets and uniform convexity for functions are connected through an equivalence of uniform convexity between a set and the set-induced norm [24].

## 2 Preliminaries

In this section, we review results from prior works on OCO for $\ell_p$-balls and connect them to fully characterise the optimal rates across all dimension regimes. The action set is $V = \mathcal{B}_p = \left\{ x \in \mathbb{R}^d : \|x\|_p \leq 1 \right\}$, the unit $\ell_p$-ball with $p > 2$. We assume we have $L$-Lipschitz losses in $\ell_p$ norm (i.e. $\|g_t\|_{p_\star} \leq L$ for $g_t \in \partial \ell_t(x_t)$), ensuring the regret incurred in a single round is bounded by $2L$, and the overall regret by $2LT$.

We first present a general regret bound for FTRL using a uniformly-convex regulariser (Section 2.1). We focus here on FTRL because of its advantages over OMD (in unbounded domains, the regret of OMD can be linear while FTRL maintains sub-linear regret [37]), although the results we discuss also hold for OMD and we include these in Appendix H. We then consider these bounds with specific regularisers and provide matching lower bounds to establish the optimal regret in the low-dimensional setting (Section 2.2) and high-dimensional setting (Section 2.3). The results from this section follow from prior work and we include the missing proofs in Appendix C.

**Remark 2.1.** *We focus on $\ell_p$-balls for $p > 2$ because the case $p \in [1, 2]$ is already understood [28]. [28] show that when $p \in [1, 2]$, OMD with regulariser $\psi(x) = \|x\|_a^2/2(a - 1)$ and $a = \max\left\{1 + 1/\log(2d), p\right\}$ achieves a regret of $O(\sqrt{T/(a - 1)})$ and this is optimal for all $d$ except if $T < 1/(a - 1)$ for which sub-linear regret is not possible.*

### 2.1 FTRL and uniformly-convex regularisation

In this section, we review the analysis of Follow-the-Regularised-Leader (FTRL) using a uniformly-convex regulariser [4] which will lead to the regret guarantees in the subsequent sections. First, we provide the definition of a uniformly-convex function from [35] (note there are also other standard equivalent definitions, see e.g. [24]).

**Definition 2.2** ([35]). *A differentiable function $f$ on a closed convex set $V$ is $\mu$-uniformly-convex on $V$ of degree $p > 2$ w.r.t. a norm $\|\cdot\|$ if there exists $\mu > 0$ such that for all $x, y \in V$,*

$$f(y) \geq f(x) + \langle \nabla f(x), y - x \rangle + \frac{\mu}{p}\|y - x\|^p.$$

Uniform convexity generalises strong convexity by weakening the condition of a quadratic lower bound allowing functions that are locally much flatter. In particular, uniform convexity with $p = 2$ recovers strong convexity. Though FTRL is usually considered with a strongly-convex regulariser, its analysis can be generalised to uniformly-convex regularisers [4], as seen in the following theorem which we write in a general form to ensure results in following sections directly follow from it.

**Theorem 2.3.** *Let $V \subset \mathbb{R}^d$ be convex and consider proper convex losses $(\ell_t)_{t=1}^T$. For $t \geq 1$, let $\psi_t : \mathbb{R}^d \to \mathbb{R}$ be a proper, closed and differentiable $\mu_t$-uniformly-convex function on $V$ of degree $r_t \geq 2$ with respect to a norm $\|\cdot\|_{|t}$ (we use this notation to avoid confusion with the $\ell_p$ norm $\|\cdot\|_p$, we will denote the dual norm as $\|\cdot\|_{|t\star}$). At time-step $t = 1, ..., T$, FTRL on linearised losses with time-varying regularisers $(\psi_t)_{t=1}^T$ outputs the following points with $g_t \in \partial \ell_t(x_t)$ for all $t$,*

$$x_t = \arg\min_{x \in V} \left\{ \psi_t(x) + \sum_{s=1}^{t-1} \langle g_s, x \rangle \right\}. \tag{1}$$

*Then for any $u \in V$ and $g_t \in \partial \ell_t(x_t)$, the points played by FTRL satisfy the following regret bound:*

$$\sum_{t=1}^T \ell_t(x_t) - \ell_t(u) \leq \psi_T(u) - \min_{x \in V} \psi_1(x) + \sum_{t=1}^T \left\{ \frac{(r_t - 1)}{r_t \mu_t^{\frac{1}{r_t - 1}}} \|g_t\|_{|t\star}^{\frac{r_t}{r_t - 1}} + \psi_t(x_{t+1}) - \psi_{t+1}(x_{t+1}) \right\}.$$

A version of this result can be found in [4] We include the proof in Appendix C.1 for completeness. If we consider a fixed regulariser with a step-size ($\psi_t(x) = \frac{1}{\eta_{t-1}}\psi(x)$ for a fixed $\psi$) so that the condition of uniform convexity is fixed for all rounds, we get the following result (proof in Appendix C.2).

**Corollary 2.4.** *Let $V \subset \mathbb{R}^d$ be convex and consider proper convex losses $(\ell_t)_{t=1}^T$. Let $\psi : \mathbb{R}^d \to \mathbb{R}_{\geq 0}$ be a proper, closed and differentiable $\mu$-uniformly-convex function on $V$ of degree $r \geq 2$ with respect to a norm $\|\cdot\|$. Assume $V$ is bounded and let $D$ be such that $\sup_{x \in V} \psi(x) \leq D$. Assume the losses are*

$L_{\|\cdot\|}$-*Lipschitz with respect to* $\|\cdot\|$. *Consider using FTRL in* (1) *with regularisers* $\psi_t(x) = \frac{1}{\eta_{t-1}}\psi(x)$ *and* $\eta_{t-1} = \frac{D^{1/r_\star}\mu^{1/r}}{L_{\|\cdot\|}(r_\star-1)^{1/r_\star}t^{1/r_\star}}$. *Then for any* $u \in V$,

$$\sum_{t=1}^{T}\ell_t(x_t) - \ell_t(u) \le \frac{r^{1/r}r_\star^{1/r_\star}}{\mu^{1/r}}L_{\|\cdot\|}D^{1/r}T^{1/r_\star}.$$

The degree $r$ of uniform convexity in the above regret bound offers a trade-off between the dependence on the horizon $T$ and the diameter $D$, while leaving the Lipschitz constant unaffected. In particular, a larger $r$ will shrink the dependence on the diameter at the cost of a worst rate w.r.t. $T$. This can give better regret bounds for high-dimensional problems where the dimension dependence arises through the diameter $D$, as is the case for $\ell_p$-balls (see Section 2.3). With $r = 2$, we recover the standard regret bound of FTRL using a strongly-convex regulariser, $R_T \le 2L_{\|\cdot\|}\sqrt{DT/\mu}$.

It is also possible to obtain bounds that are Lipschitz-adaptive (do not require knowledge of $L_{\|\cdot\|}$) and adapt to the sequence of sub-gradients (scale with $\sum_{t=1}^{T}\|g_t\|_\star^{r_\star}$ instead of $T^{1/r_\star}$) using a gradient clipping technique (see the blog-post by [41] and Section 4 in [9]).

## 2.2 Low-dimensional regime

In this section, we consider OCO on $\mathcal{B}_p$ in the low-dimensional setting where $d \le T$. Using FTRL with strongly-convex regularisation in this setting achieves the optimal regret. Specifically, we consider the squared $\ell_2$-norm $\phi_2(x) = \frac{1}{2}\|x\|_2^2$ as the regulariser. This is 1-strongly-convex on $V$ with respect to $\|\cdot\|_2$ and we can apply Corollary 2.4 with $r = 2$. We have that $L_{\|\cdot\|_2} \le L$ since the losses are $L$-Lipschitz with respect to $\|\cdot\|_p$ and $p_\star \le 2$ so $\|g_t\|_2 \le \|g_t\|_{p_\star} \le L$. The diameter of $\mathcal{B}_p$ measured by $\phi_2$ is $D = \sup_{x \in \mathcal{B}_p}\phi_2(x) = \sup_{x \in \mathcal{B}_p}\frac{1}{2}\|x\|_2^2 = \frac{1}{2}d^{1-2/p}$. So FTRL guarantees $R_T \le L\sqrt{2d^{1-2/p}T}$, which is optimal up to constants for $d \le T$ as shown by the theorem below.

**Theorem 2.5.** *Fix* $d \le T$ *and let* $\mathcal{A}$ *be any algorithm for OCO on* $V = \mathcal{B}_p$. *There exists a sequence of* $L$-*Lipschitz losses w.r.t.* $\|\cdot\|_p$ *such that* $\mathcal{A}$ *suffers a regret of at least* $\frac{1}{\sqrt{6}}L\sqrt{d^{1-2/p}T}$.

Optimality in low-dimension was established by [28] (both upper and lower bounds). However, the lower bound we present above contains better constants and a simpler analysis stemming from the "probabilistic" method instead of the reductions from estimation to testing used by [28]. The proof can be found in Appendix C.3.

## 2.3 High-dimensional regime

In this section, we consider OCO on $\mathcal{B}_p$ in the high-dimensional setting where $d > T$. We saw in the previous section that the optimal regret in low-dimensions of $O(\sqrt{d^{1-2/p}T})$ is polynomial in the dimension. As $d \to \infty$ for fixed $T$, this polynomial dependence on the dimension cannot remain optimal since $R_T \le 2LT$ is bounded. Nevertheless, we will see that sub-linear regret bounds (in $T$) are possible for any $d > T$, even when $d$ is such that $\sqrt{d^{1-2/p}T} \ge T$. However, this is not achievable using strongly-convex regularisation (we delay discussion of this failure to Section 4). Instead, in this section we consider uniformly-convex regularisation of degree $p > 2$ that enforces less curvature, allowing points in the corners of $\mathcal{B}_p$ to be more appropriately regularised in high dimensions. This allows us to obtain optimal regret bounds for the high-dimensional regime.

We consider FTRL on $\mathcal{B}_p$ with the regulariser $\phi_p(x) = \frac{1}{p}\|x\|_p^p$. The following proposition ensures that $\phi_p$ is uniformly-convex of degree $p$ with respect to $\|\cdot\|_p$. This is a well-known result derived from Clarkson's inequality. We include a proof in Appendix C.5 to provide clarity on the constant of uniform convexity.

**Proposition 2.6.** *Fix* $p > 2$. $\phi_p(x) = \frac{1}{p}\|x\|_p^p$ *is* $2^{1-p}$-*uniformly-convex of degree* $p$ *w.r.t.* $\|\cdot\|_p$ *on* $\mathcal{B}_p$.

We can now apply Corollary 2.4 with $r = p$. We have that $L_{\|\cdot\|_p} = L$ since the losses are $L$-Lipschitz with respect to $\|\cdot\|_p$. The diameter of $\mathcal{B}_p$ measured by $\phi_p$ is $D = 1/p$. So FTRL guarantees $R_T \le L(2p_\star T)^{1/p_\star}$, which is optimal up to constants for $d > T$ as shown by the theorem below.

**Theorem 2.7.** *Fix $d > T$ and let $\mathcal{A}$ be any algorithm for OCO on $V = \mathcal{B}_p$. There exists a sequence of $L$-Lipschitz losses w.r.t. $\|\cdot\|_p$ such that $\mathcal{A}$ suffers a regret of at least $LT^{1/p_\star}$.*

This lower bound follows from an online-to-batch conversion of the lower bound for high-dimensional (offline) Lipschitz convex optimisation [34, 18, 17] or an instantiation of the lower bound by [40] (Lemma 15). We include the details of the latter in Appendix C.4.

## 3 Anytime optimality through adaptive regularisation

In the previous section (Section 2), we saw that the optimal regret is achieved with strongly-convex regularisation in the low-dimensional setting ($d \leq T$) and with uniformly-convex regularisation in the high-dimensional setting ($d > T$). To be optimal, the learner with knowledge of the dimension $d$ and the time horizon $T$ can evaluate whether $d > T$ or $d \leq T$ and select the appropriate regularisation based on whether the problem is high or low dimensional. However, choosing the correct regulariser relies on knowing whether the problem is high ($d > T$) or low ($d \leq T$) dimensional, which itself relies on knowing the horizon $T$. In this section, we consider how to achieve anytime optimal regret bounds which hold without knowledge of $T$.

We consider FTRL with regularisation that adapts to the dimension regime. Fix $t_0 = 3^{-2p/(p-2)}d$. Then, in early high-dimensional rounds, the uniformly-convex $\phi_p$ is used, until the threshold $t_0$ when the low-dimensional regime is reached and the regulariser switches to the strongly-convex $\phi_2$. In both cases, the step-size used is the one in Corollary 2.4. Specifically, with $\phi_r(x) = \frac{1}{r}\|x\|_r^r$ for $r \geq 2$, we consider FTRL with regulariser at time $t$ given by

$$\psi_t(x) = \begin{cases} \frac{1}{\eta_{t-1}}\phi_p(x), & \eta_{t-1} = \frac{1}{L(2p_\star t)^{1/p_\star}}, & \text{if } t \leq t_0, \\ \frac{1}{\eta_{t-1}}\phi_2(x), & \eta_{t-1} = \frac{\sqrt{d^{1-2/p}}}{L\sqrt{2t}}, & \text{if } t > t_0. \end{cases} \tag{2}$$

FTRL with this sequence of regularisers is anytime optimal as shown by the following theorem.

**Theorem 3.1.** *Let $V = \mathcal{B}_p$ ($p > 2$) and consider proper convex losses $(\ell_t)_{t=1}^T$ that are $L$-Lipschitz with respect to $\|\cdot\|$. Consider FTRL with regularisers given in (2). Then*

$$R_T \leq \begin{cases} L\left(2p_\star T\right)^{1/p_\star}, & \text{if } T \leq t_0, \\ L\sqrt{2Td^{1-2/p}}, & \text{if } T > t_0 \end{cases}$$

The proof is in Appendix D and consists of a careful application of Theorem 2.3. The time-step where the regulariser changes is handled by the specific value of the threshold $t_0 = 3^{-2p/(p-2)}d$. This value allows us to recover the same low-dimensional bound (including constants) as in Section 2.2 achieved using strongly-convex regularisation from the start. For the high-dimensional setting, there is no switch in regulariser so the algorithm and regret bounds are identical to those in Section 2.3. In other words, *being agnostic to the dimension regime comes at no cost to the regret bound*. The above procedure can be used with gradient-clipping techniques discussed by [41] to obtain a Lipschitz-adaptive anytime optimal algorithm. OMD can also be used to achieve anytime optimality with similar adaptive regularisation (see Appendix H).

Our anytime-optimal procedure is like a restarting technique except the step-size in the later low-dimensional time-steps accounts for the earlier time-steps. This makes constants not degrade. In potentially more complicated settings requiring many switches in regularisation, not accounting for earlier time-steps in the step-size may come at the cost of more than just constants. A doubling-trick approach could also be used, though at the cost of worse constants (see e.g. Appendix F).

## 4 Necessity of adaptive regularisation

In the previous section, we demonstrated that adaptive regularisation achieves anytime optimal regret bounds for OCO on $\ell_p$-balls with $p > 2$. In this section, we show that for separable regularisers, adaptive regularisation is necessary for anytime optimality. We first show that the regularisers we have considered up to now are provably anytime sub-optimal: in Section 4.1 we show that strong convexity fails in high-dimension; in Section 4.2, we show that the uniformly-convex regulariser $\phi_p = \frac{1}{p}\|x\|_p^p$ fails in low-dimension. Then, we present the main result of this section on the failure of using a fixed separable regulariser in Section 4.3. All the missing proofs for this section can be found in Appendix E.

## 4.1 Failure of strong convexity in high-dimensions

We saw in Section 2.2 that the strongly-convex regulariser $\phi_2(x) = \frac{1}{2}\|x\|_2^2$ achieves the optimal $O(\sqrt{d^{1-2/p}T})$ regret guarantee in the low-dimensional setting ($d \leq T$) but this bound is sub-optimal in the high-dimensional setting ($d > T$) because of the polynomial dependence on the dimension. We show that such a sub-optimal polynomial dependence on the dimension necessarily appears in the regret bound for any strongly-convex regulariser on $\mathcal{B}_p$. This occurs since these strongly-convex regularisers can be shown to take values that scale polynomially with $d$ for points in the corners of the $\ell_p$-balls [10, Example 4.1]. The following two results establish that these sub-optimal regret bounds are not loose and that strongly-convex regularisers are provably sub-optimal in high-dimensions. The first is a lower bound specific to FTRL with regulariser $\phi_2$.

**Proposition 4.1.** *There exists a sequence of linear L-Lipschitz losses (in $\ell_p$-norm) for which FTRL with regulariser $\psi_t(x) = \frac{1}{\eta_{t-1}}\phi_2(x)$ and any sequence of decreasing $\eta_{t-1}$ suffers regret*

$$R_T \geq L \cdot \min\Big(\frac{T}{16}, \frac{1}{8}\sqrt{Td^{1-2/p}}\Big).$$

The above proposition shows the regret of FTRL with regulariser $\phi_2$ scales polynomially with $d$ until it is linear in $T$. This demonstrates the analysis from Section 2.2 is in fact tight and this algorithm is sub-optimal in high-dimensions. We now state a more general result that shows that using FTRL with any strongly-convex regulariser fails if the dimension is large enough. This also establishes that strongly-convex regularisers cannot be anytime optimal (see Section 4.3).

**Theorem 4.2.** *Consider a sign invariant[2] regulariser $\psi$ that is $\mu$-strongly-convex with respect to an arbitrary norm $\|\cdot\|$ (s.t. $\|e_i\| = 1$ for all i) and attains its minimum value 0 at $x = 0$. Consider $V = \mathcal{B}_p$ with $p > 2$ and assume losses are L-Lipschitz in $\ell_p$-norm. If $d \geq (4T/\mu)^{p/(p-2)}$, there exists a sequence of linear L-Lipschitz losses (in $\ell_p$-norm) for which FTRL with regulariser $\psi_t(x) = \frac{1}{\eta_{t-1}}\psi(x)$ and any sequence of decreasing $\eta_{t-1}$ suffers regret $R_T \geq \frac{1}{8}LT$.*

The above theorem is a consequence of the following two lemmas (proofs in Appendix E).

**Lemma 4.3.** *Consider a sign-invariant function $\psi$ that is $\mu$-strongly-convex with respect to an arbitrary norm $\|\cdot\|$ (s.t. $\|e_i\| = 1$ for all i) and attaining its minimum value 0 at $x = 0$. Then $\psi(x) \geq \frac{\mu}{2}\|x\|_2^2$.*

**Lemma 4.4.** *Consider $V = \mathcal{B}_p$ with $p > 2$ and assume losses are L-Lipschitz in $\ell_p$-norm. Let $\psi$ be a convex function satisfying for some $\mu > 0$ and any $x \in \mathbb{R}^d$, $\psi(x) \geq \frac{\mu}{2}\|x\|_2^2$. If $d \geq (4T/\mu)^{p/(p-2)}$, there exists a sequence of linear L-Lipschitz losses (in $\ell_p$-norm) for which FTRL with regulariser $\psi_t(x) = \frac{1}{\eta_{t-1}}\psi(x)$ and any sequence of decreasing $\eta_{t-1}$ suffers regret $R_T \geq \frac{1}{8}LT$.*

## 4.2 Failure of uniform convexity in low-dimension

We saw in Section 2.3 that the uniformly-convex regulariser of degree $p$, $\phi_p(x) = \frac{1}{p}\|x\|_p^p$, achieves optimal regret guarantees in the high-dimensional setting ($d > T$). The next result shows its regret guarantees are provably sub-optimal in low-dimensions, where the optimal rate is $O(\sqrt{Td^{1-2/p}})$.

**Proposition 4.5.** *There exists a sequence of linear L-Lipschitz losses (in $\ell_p$-norm) for which FTRL with regulariser $\psi_t(x) = \frac{1}{\eta_{t-1}}\phi_p(x)$ and any sequence of decreasing $\eta_{t-1}$ suffers regret*

$$R_T \geq L \cdot \min\Big(\frac{T}{8p}, \frac{T^{1/p_\star}}{8}\Big).$$

We remark that a general result for uniformly-convex function of degree $p$ as we had for strong convexity in Theorem 4.2 does not hold. This is because uniform convexity is a condition on the minimum curvature and so is not the reason for the failure of $\phi_p$ in the low-dimensional setting. The reason for the failure is that $\phi_p$ is only uniformly-convex of degree $p$ and does not satisfy some stronger curvature condition. Regularisers with stronger curvature conditions such as strong convexity

---

[2]A function $f : \mathcal{X} \subset \mathbb{R}^d \to \mathbb{R}$ is sign invariant if for any $s \in \{-1, 1\}^d$, $f(s \cdot x) = f(x)$ for all $x \in \mathcal{X}$ where $s \cdot x$ denotes coordinate-wise multiplication.

that are optimal in the low-dimensional setting naturally also satisfy uniform convexity of degree $p > 2$ on $\mathcal{B}_p$. The combination of the failure of strong convexity in high-dimensions and the necessity for strong curvature conditions in low-dimensions is the key insight for showing the necessity of adaptivity for separable regularisers in the next section.

### 4.3   Failure of fixed separable regularisation

In this section, we study the anytime optimality of FTRL under a fixed regulariser. While the previous two sections established that the two specific regularisers we considered in Section 2 (which are each optimal for one regime) are not able to guarantee this, it does not rule out the existence of a regulariser that could. We consider the class of separable regularisers defined as

$$\Psi = \Big\{ \psi : \mathcal{B}_p \to \mathbb{R} : \psi(x) = \sum_{i=1}^{d} g(x_i), g \in \mathcal{F} \Big\}, \tag{3}$$

where $\mathcal{F} = \big\{ g : \mathbb{R} \to \mathbb{R}_{\geq 0}; \text{convex, sign-invariant}, 0 = \arg\min_{x \in \mathbb{R}} g(x), g(0) = 0, g(1) = 1 \big\}$.

The function class $\mathcal{F}$ is a set of 1-dimensional even regularisers scaled to be in $[0, 1]$ for $x \in [-1, 1]$ (e.g. $x^r$ for $r \geq 1$). The class $\Psi$ covers a wide range of regularisers including all of the form $\|x\|_r^r$ for any $r \geq 1$. We can now state our main result on the failure of fixed separable regularisation. This result also holds for OMD with minor modification to the proof (see Appendix H).

**Theorem 4.6.** *FTRL with regulariser $\psi_t(x) = \frac{1}{\eta_{t-1}} \psi(x)$ for $\psi \in \Psi$ and any sequence of decreasing $\eta_{t-1}$ cannot be optimal across all dimensions. Specifically there are no constants $c_h, c_l > 0$ such that for all $T$, $R_T \leq c_h L T^{1-1/p}$ for all $d > T$ and $R_T \leq c_l L \sqrt{T} d^{1-2/p}$ for all $d \leq T$.*

*Proof.* Let's assume there are constants $c_h, c_l > 0$ such that for all $T$, $R_T \leq c_h L T^{1-1/p}$ for all $d > T$ and $R_T \leq c_l L \sqrt{T} d^{1-2/p}$ for all $d \leq T$ and show a contradiction. We begin with a lemma showing the necessity of quadratic growth of a regulariser achieving optimal regret in the 1-dimensional case (proof in Appendix E.5).

**Lemma 4.7.** *Consider $d = 1$ ($V = \mathcal{B}_p = [-1, 1]$) and $\psi \in \mathcal{F}$. FTRL with regulariser $\psi_t(x) = \frac{1}{\eta_{t-1}} \psi(x)$ and arbitrary decreasing step-size $\eta_{t-1}$ can only guarantee $R_T \leq cL\sqrt{T}$ for some constant $c > 0$ and all sufficiently large $T$ if for all $x \in [-1, 1]$, $\psi(x) \geq \frac{\psi(1/2)}{100c^2} x^2$.*

For $d = 1$, we have $\psi(x) = g(x)$ and under our assumption, $R_T \leq c_l L \sqrt{T}$ for all $T$, so for all $x \in [-1, 1]$, $g(x) \geq \frac{g(1/2)}{100c_l^2} x^2$ by Lemma 4.7. Hence in the general $d$-dimensional setting for $x \in \mathcal{B}_p$:

$$\psi(x) = \sum_{i=1}^{d} g(x_i) \geq \frac{g(1/2)}{100c_l^2} \sum_{i=1}^{d} x_i^2 = \frac{g(1/2)}{100c_l^2} \|x\|_2^2.$$

We can now use this lower bound with Lemma 4.4 (from Section 4.1) and $\mu = \frac{g(1/2)}{50c_l^2}$. This gives that if $d \geq \left( \frac{200c_l^2 T}{g(1/2)} \right)^{p/(p-2)}$ then there exists a sequence of linear $L$-Lipschitz losses (in $\ell_p$-norm) for which $R_T \geq \frac{1}{8} LT$, contradicting that $R_T \leq c_h L T^{1-1/p}$ for all $d > T$.   $\square$

The above result establishes the need for regularisation adaptive to the dimension-regime for anytime optimality when using FTRL with separable regularisers from $\Psi$. In particular, having seen that regularisers $\frac{1}{2}\|x\|_2^2$ (strong convexity) and $\frac{1}{p}\|x\|_p^p$ (uniform convexity of degree $p$) fail to achieve optimal anytime regret in the previous sections, we may be tempted to consider $\frac{1}{r}\|x\|_r^r$ for $r \in (2, p)$ that could trade-off the optimalities of strong-convexity in low-dimension and of uniform-convexity in high-dimension. However, Theorem 4.6 rules out this possibility. See also Proposition E.1 in Appendix E for a precise characterisation of the regret when using $\frac{1}{r}\|x\|_r^2$.

**Remark 4.8.** *We can slightly relax the constraint of separable regularisers in Theorem 4.6. Firstly, in the definition of $\Psi$ (3), a different $g_i \in \mathcal{F}$ can be used for each coordinate. The result then still holds but the quadratic lower bound on $\psi(x)$ stemming from Lemma 4.7 in the proof will scale*

*with* $\min_{1\leq i\leq d} g_i(1/2)$ *and the dimension from which the regret becomes linear in* $T$ *scales with* $\min_{1\leq i\leq d} g_i(1/2)^{p/(p-2)}$. *Secondly, Theorem 4.6 also holds more generally for regularisers* $\psi$ *which are within constants of a separable regulariser:* $c_1\psi'(x) \leq \psi(x) \leq c_2\psi'(x)$ *for* $\psi' \in \Psi$ *and constants* $c_1, c_2 > 0$. *Finally, by Theorem 4.2, the result holds for any strongly-convex regulariser.*

**Remark 4.9.** *In Appendix E, we prove a more general version of Lemma 4.4 for coordinate-wise step-sizes, where the FTRL update is allowed to have a different step-size* $\eta_{t-1,i}$ *for each coordinate:* $x_t = \arg\min_{x \in V}\left\{\psi(x) + \sum_{i=1}^{d} \eta_{t-1,i}\cdot x_i \sum_{s=1}^{t-1} g_{s,i}\right\}$. *Using this version in the proof of Theorem 4.6 gives the same result for any sequence of coordinate-wise decreasing step-sizes. This extension establishes the failure of a wider range of methods, in particular the diagonal versions of Adagrad-style algorithms [12].*

**Remark 4.10.** *We discuss the connection of our result to the universality of FTRL. It was shown by [39] that for a fixed* $r \in [2, \infty)$ *and constant* $C > 0$, *for any OCO problem for which a regret upper bound of* $CT^{1-1/r}$ *can be guaranteed for all* $T$, *then for any* $T > e^{r-1}$, *there exists a regulariser with which OMD/FTRL can guarantee a regret bound of* $\widetilde{O}(T^{1-1/r})$[3]. *We present this result in more detail in Appendix F and also provide an extension to include the setting where* $r$ *may change according to* $T$ *as in our setting. However, this latter result uses a doubling trick where different regularisers are used across separate intervals. This poses a question on the universality of FTRL with fixed regularisation for more general OCO problems where the optimal rate of regret is horizon or dimension dependent. Theorem 4.6 offers a negative answer to this question for separable regularisers. However, the case of more general regularisers remains open. Note that in our setting, the non-separable regulariser from [39] is within a constant fraction of* $\|x\|_p^p$ *so, Theorem 4.6 still applies (see Remark 4.8). To extend our result beyond separable regularisers, one possible approach is to extend Lemma 4.7 on the quadratic-growth of the regulariser beyond the* 1-*dimensional case.*

**Remark 4.11.** *The generality of the failure of separable regularization beyond* $\ell_p$-*ball structures is likely related to the concept of quadratically convex sets [28] for which several results are known on the possibility of getting regret that grows as* $\sqrt{T}$ *in the low-dimensional case, while it is likely that the regret is better in the high dimensional case (when taking into account dependence on other quantities like dimension). This is an interesting direction of future research. We also note that our results hold for* $\ell_p$-*balls translated away from 0, provided we consider regularisers satisfying our conditions on this translated ball (e.g. reaches its minimum in the centre of the translated* $\ell_p$-*ball).*

## 4.4 Proof intuitions

Many of the results discussed in this section so far are based on the same loss construction, with the following linear losses $\ell_t(x) = L \cdot x^T g_t$ where $g_t \in \mathcal{B}_{p_\star}$ is defined as

$$g_t = \begin{cases} (-1)^t \cdot e_1, & t \leq \frac{T}{2}, \\ -v, & t > \frac{T}{2}, \end{cases}$$

where $v \in \mathcal{B}_{p_\star}$ is a vector with equal entries defined as $v_{t,i} = d^{-1/p_\star}$.

**The above construction is motivated by the following intuition:** The gradients of the losses in the first half of the rounds cancel each other. The competitor is thus only dependent on the losses in the second half of the rounds which are constant and place the competitor in the corner of the $\ell_p$-ball. This two phase construction captures a bias-variance-like trade-off of FTRL with fixed regularisation:

- If the step-size is small, FTRL will not suffer much regret in the first half of the rounds but in the second half it will not be able to reach the corner of the ball (the competitor) sufficiently quickly and suffer large-regret (it has high "bias").
- If the step-size is large then FTRL will be able quickly reach the corner of the ball (the competitor) in the second half of the rounds but in the first it will suffer large regret because it moves too fast through the space and every other round it will get to close to $e_1$ which will make it suffer large loss in the next round (it has high "variance").

This construction is designed so that the performance of FTRL is at its best when the step-size adequately balances the trade-off between the losses from both halves of the rounds. Analysing the resulting regret gives us many of our results.

---

[3]This result was recently improved by [15] to omit any log-factors in $T$ for the case of online linear optimisation and $r = 2$

In the case of $\phi_2$ and $\phi_p$, by computing the points played by FTRL we can explicitly compute the regret suffered by FTRL for all step-sizes and obtain the lower bounds in Proposition 4.1 and Proposition 4.5 respectively. For Lemma 4.7, we exploit that in the 1-dimensional case the losses in the second half of the rounds are equal to the losses in the odd first half of the rounds, which enables a direct comparison between points played in the first and second half of the rounds. This comparison allows us to establish that in order to obtain $\sqrt{T}$ regret, the regulariser must have quadratic growth. The full details for all the proofs are in Appendix E.

## 5 Bandit feedback

In this section, we consider OCO on $\ell_p$-balls with bandit feedback and linear losses. In the bandit feedback environment, the learner only observes the loss evaluated at the point played $\ell_t(x_t)$ instead of the full loss function $\ell_t$. Similarly to the full information setting studied above, the optimal regret with bandit feedback also depends on the dimension regime. However, our main result in this section will show that for bandit feedback, sub-linear regret is not possible in the high dimensional regime.

The linear bandit problem has been extensively studied (see e.g. [27]). A $\widetilde{O}(d^{1/2}T^{1/2})$ pseudo-regret[4] bound was established by [6] for $\ell_p$-balls ($p \in (1, 2]$) and more generally for strongly-convex action sets by [25]. For $p = 1$, the same bound can be achieved via a reduction to the multi-armed bandit problem (see [6]). [25] also established $\widetilde{O}(d^{1/p}T^{1-1/p})$ pseudo-regret bounds for uniformly-convex sets of degree $p$, which apply to $\ell_p$-balls with $p > 2$. However, since these regret guarantees are dimension-dependent they become vacuous in high-dimensions. The following result shows that with bandit feedback sub-linear regret bounds on $\ell_p$-balls are unachievable in high-dimensions.

**Theorem 5.1.** *Fix $T$, $\delta > 0$ and $p \geq 1$. For any dimension $d$ sufficiently large and any OCO algorithm with bandit feedback on $V = \mathcal{B}_p$, there exists a sequence of random linear losses $(\ell_t)_{t\in[T]}$ with sub-gradients $(g_t)_{t\in[T]}$ such that $\|g_t\|_{p_\star} \leq L$ for all rounds $t$ with probability at least $1 - \delta$ and $\mathbb{E}[\bar{R}_T] \geq \frac{LT}{80}$, where the expectation is with respect to the randomness of the losses.*

The proof can be found in Appendix G and is based on information-theoretic arguments. We note that lower bounds in Chapter 24 of [27] give linear regret for high-dimensional stochastic linear bandits with $p = 1, 2$. However, as discussed in their Chapter 29, the noise in the stochastic case is outside the inner-product so these bounds do not apply for adversarial linear bandits. We also remark that dimension-dependent bandit lower bounds from prior work such as the one in [6] only hold in the low-dimensional setting (e.g. in [6], their lower bound Theorem 4 only holds for $T \geq d^{2/(1-q/2)}$ and does not give linear regret in high dimension).

## 6 Conclusion

In this work, we studied OCO on $\ell_p$-balls in $\mathbb{R}^d$ for $p > 2$, distinguishing between high-dimensional ($d > T$) and low-dimensional ($d \leq T$) regimes. In high-dimensions, FTRL achieves the optimal regret of $O(T^{1-1/p})$ with a uniformly-convex regulariser, while in low-dimensions it achieves the optimal regret of $O(T^{1/2}d^{1/2-1/p})$ with a strongly-convex regulariser. Importantly, we proved neither regulariser is optimal across both regimes. Therefore, when the dimension regime is unknown, we showed that FTRL with adaptive regularisation is anytime optimal. Furthermore, *we established that this adaptivity is necessary to achieve anytime optimality for separable regularisers.* This is a first step in answering a question on the universality of FTRL with fixed regularisation for general OCO problems. However, it remains open whether there exists a fixed regulariser providing anytime optimality or whether adaptivity for non-separable regularisers is necessary. Our results demonstrate that existing separable regularisers impose intrinsic limitations on FTRL and open up an interesting avenue of research to discover more sophisticated alternatives that potentially give algorithms that are fundamentally different. The challenge in generalising our proof technique to rule out the existence of these alternative non-separable regularisers is in extending Lemma 4.7 on the quadratic-growth of the regulariser beyond the 1-dimensional case. Finally, for the linear bandit problem, we ruled out the possibility for sub-linear regret bounds in high-dimension. These results underscore the role of dimension and geometry in achieving optimal performance in OCO.

---

[4]Pseudo-regret is defined as $\bar{R}_T = \mathbb{E}\left[\sum_{t=1}^{T} \ell_t(x_t)\right] - \min_{x \in V} \mathbb{E}\left[\sum_{t=1}^{T} \ell_t(x)\right]$ where the expectation is with respect to the randomness in the learner's actions.

## Acknowledgments and Disclosure of Funding

Emmeran Johnson is funded by EPSRC through the Modern Statistics and Statistical Machine Learning (StatML) CDT (grant no. EP/S023151/1). David Martínez-Rubio was partially supported by a 2025 Leonardo Grant for Scientific Research and Cultural Creation from the BBVA Foundation. Patrick Rebeschini was funded by UK Research and Innovation (UKRI) under the UK government's Horizon Europe funding guarantee (grant no. EP/Y028333/1).

We would like to thank the reviewers and meta-reviewers for their time and feedback.

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

# A Experiments

In this section, we present a numerical experiment in Figure 1 to validate some of our theoretical results on the optimality and sub-optimality of fixed and adpative regularisation for FTRL. We run FTRL with different regularisers on the loss construction used in the proofs of our lower bounds from Section 4, which is described in Appendix E.1.

For fixed $T$, we observe that the regret using FTRL with $\phi_p$ is constant across dimension, while the regret of FTRL with $\phi_2$ increases with dimension. In particular, $\phi_2$ outperforms $\phi_p$ in low-dimension while $\phi_p$ outperforms $\phi_2$ in high dimensions. This validates our results that $\phi_p$ is optimal in high dimensions (Section 2.3) but not in low-dimension (Section 4.2) and that $\phi_2$ is optimal in low-dimension (Section 2.2) but not in high dimensions (Section 4.1). Furthermore, the adaptive procedure from Section 3 performs well in both low and high dimensions. However, this experiment suggests that the theoretical threshold $t_0 = 3^{-2p/(p-2)}d$ from Theorem 3.1 is perhaps overly conservative in the transient setting between low and high dimensions (at least for this loss construction) and that a larger threshold $t_0 = 2d$ performs better here.

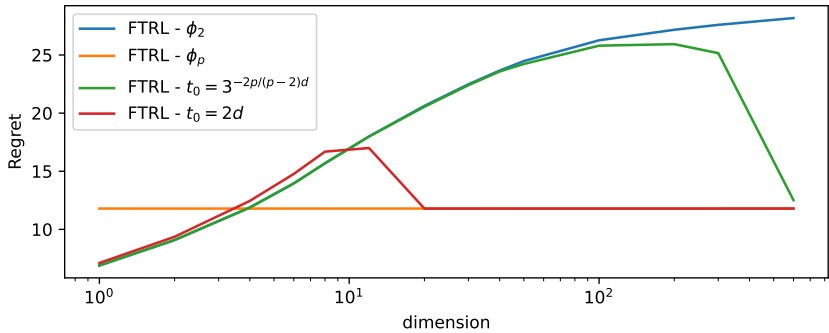

Figure 1: Comparison of FTRL with different regularisation. We fix $T = 40$ (and $L = 1$, $p = 10$) and vary the dimension. FTRL - $\phi_2$ refers to FTRL using the regulariser $\phi_2 = \frac{1}{2}\|x\|_2^2$ from Section 2.2 with $\eta_{t-1} = \sqrt{\frac{d^{1-2/p}}{2t}}$. FTRL - $\phi_p$ refers to FTRL using the regulariser $\phi_p = \frac{1}{p}\|x\|_p^p$ from Section 2.3 with $\eta_{t-1} = \frac{1}{(2p_\star t)^{1/p_\star}}$. The final two correspond to using the procedure from Section 3 with adaptive regularisation. The first with the threshold $t_0 = 3^{-2p/(p-2)}d$ from Theorem 3.1, while the second uses the threshold $t_0 = 2d$.

**Implementation Details:** The experiment was run on google colab with the default settings (including CPU) and takes around 5 minutes run. All the details of the loss construction and algorithms are provided or referenced above. The closed-form updates are provided in Appendix B. Note that the Bregman projections onto $\ell_p$-balls are not available analytically, we use the minimize function from the scipy.optimize library to compute the projections numerically (with method='SLSQP').

# B  Closed-form update of FTRL with specific uniformly convex regulariser and related lemmas

Consider a regulariser $\psi$ differentiable on $\mathbb{R}^d$. Define the Bregman divergence of $\psi$ as $D_\psi(x, y) = \psi(x) - \psi(y) - \langle \nabla\psi(y), x - y \rangle$ for all $x, y \in \mathbb{R}^d$.

**Lemma B.1.** *Fix $r \geq 2$. Let $\psi(x) = \frac{1}{r}\|x\|_r^r$. Let $V = \mathcal{B}_p$. Let $g_t \in \partial\ell_t(x_t)$. The update rule of FTRL using $\psi_t(x) = \frac{1}{\eta_{t-1}}\psi(x)$ as regularisers is*

$$\tilde{G}_{t+1} = -\eta_t \sum_{s=1}^{t} g_s$$

$$x_{t+1} = \arg\min_{x \in \mathcal{B}_p} D_\psi\left(x, sign(\tilde{G}_{t+1})|\tilde{G}_{t+1}|^{r_\star - 1}\right),$$

*where sign, power and absolute value functions are applied component-wise to vectors.*

*Proof.* Given $g_t \in \partial\ell_t(x_t)$, the update of FTRL with regulariser $\psi_t$ is (see (1))

$$x_{t+1} = \arg\min_{x \in V}\left\{\eta_t \langle \sum_{s=1}^{t} g_s, x \rangle + \psi(x)\right\}.$$

By Theorem 6.15 in [36], this update is equivalent to

$$\tilde{x}_{t+1} = \arg\min_{x \in \mathbb{R}^d}\left\{\eta_t \langle \sum_{s=1}^{t} g_s, x \rangle + \psi(x)\right\},$$

$$x_{t+1} = \arg\min_{x \in \mathcal{B}_p} D_\psi\left(x, \tilde{x}_{t+1}\right).$$

Now by Theorem 6.13 of [36], the first minimisation (over $\mathbb{R}^d$) is equivalent to

$$\tilde{x}_{t+1} = \nabla\psi^\star\left(-\eta_t \sum_{s=1}^{t} g_s\right),$$

where $\psi^\star$ is the Fenchel conjugate of $\psi$.

For an arbitrary norm $\|\cdot\|$, the Fenchel conjugate of $f(x) = \frac{1}{r}\|x\|^r$ is $f^\star(x) = \frac{1}{r_\star}\|x\|_\star^{r_\star}$ (see Lemma 2.2 in [24]). Therefore the Fenchel conjugate of $\psi(x)$ is $\psi^\star(x) = \frac{1}{r_\star}\|x\|_{r_\star}^{r_\star}$ and $\nabla\psi^\star(x) = sign(x)|x|^{r_\star - 1}$. Combining everything gives the result. $\qquad\square$

We now provide two lemmas pertaining to the Bregman projections of the FTRL update in Lemma B.1 for specific cases that will be of use in the proofs in Appendix E.

**Lemma B.2.** *Consider $z = c \cdot w$ where $w$ is a vector with all entries equal to $1$ and $c > d^{-1/p}$ so that $z \notin \mathcal{B}_p$. The Bregman projection $argmin_{x \in \mathcal{B}_p} D_\psi(x, z)$ with $\psi(x) = \frac{1}{r}\|x\|_r^r$ of $z$ is $d^{-1/p} \cdot w$, the rescaled version of $w$ that has $\ell_p$-norm equal to 1.*

*Proof.* We make use of Lemma 5.4 in [5]: if $f$ is a convex and differentiable function on $\mathcal{B}_p$ then $x$ is a minimiser of $f(x)$ in $\mathcal{B}_p$ if and only if $\nabla f(x)^T(y - x) \geq 0$ for all $y \in \mathcal{B}_p$. Consider

$$f(x) = D_\psi(x, z) = \psi(x) - \psi(z) - \nabla\psi(z)^T(x - z),$$
$$\nabla f(x) = \nabla\psi(x) - \nabla\psi(z),$$
$$[\nabla\psi(x)]_i = sign(x_i)|x_i|^{r-1}$$

Consider $x = d^{-1/p} \cdot w$. From the lemma mentioned above, it is enough to show that $\nabla f(x)^T(y - x) \geq 0$ for all $y \in \mathcal{B}_p$:

$$
\begin{aligned}
\nabla f(x)^T(y - x) &= (\nabla \psi(x) - \nabla \psi(z))^T(y - x) \\
&= (d^{-(r-1)/p} \cdot w - c^{r-1} \cdot w)^T(y - x) \\
&= (c^{r-1} - d^{-(r-1)/p})w^T(x - y) \\
&= (c^{r-1} - d^{-(r-1)/p})(d^{1-1/p} - \sum_{i=1}^{d} y_i) \\
&\geq (c^{r-1} - d^{-(r-1)/p})(d^{1-1/p} - \|y\|_1) \\
&\geq (c^{r-1} - d^{-(r-1)/p})(d^{1-1/p} - d^{1-1/p}\|y\|_p) \\
&\geq 0,
\end{aligned}
$$

where we used that $c^{r-1} - d^{-(r-1)/r} > 0$ and $\|y\|_1 \leq d^{1-1/p}\|y\|_p \leq d^{1-1/p}$ for all $y \in \mathcal{B}_p$. $\qquad\square$

**Lemma B.3.** *Consider $z = c \cdot e_1$ where $e_1$ is the first canonical basis vector and $|c| > 1$ so that $z \notin \mathcal{B}_p$. The Bregman projection $\operatorname{argmin}_{x \in \mathcal{B}_p} D_\psi(x, z)$ with $\psi(x) = \frac{1}{r}\|x\|_r^r$ of $z$ is $\operatorname{sign}(c) \cdot e_1$.*

*Proof.* As in the proof of Lemma B.2, it is enough to show that $\nabla f(x)^T(y - x) \geq 0$ for all $y \in \mathcal{B}_p$, with $x = \operatorname{sign}(c) \cdot e_1$, and

$$
\begin{aligned}
f(x) &= D_\psi(x, z) = \psi(x) - \psi(z) - \nabla \psi(z)^T(x - z), \\
\nabla f(x) &= \nabla \psi(x) - \nabla \psi(z), \\
[\nabla \psi(x)]_i &= \operatorname{sign}(x_i)|x_i|^{r-1}. \\
\nabla f(x)^T(y - x) &= (\nabla \psi(x) - \nabla \psi(z))^T(y - x) \\
&= (\nabla \psi(\operatorname{sign}(c) \cdot e_1) - \nabla \psi(c \cdot e_1))^T(y - x) \\
&= \operatorname{sign}(c) \cdot (|c|^{r-1} - 1)e_1^T(x - y) \\
&= \operatorname{sign}(c) \cdot (|c|^{r-1} - 1)(\operatorname{sign}(c) - y_1) \\
&\geq 0,
\end{aligned}
$$

where we used that $|c| > 1$ and $y_1 \leq 1$ for all $y \in \mathcal{B}_p$. $\qquad\square$

# C Proofs for Section 2

## C.1 Proof of Theorem 2.3

We follow and extend the analysis of FTRL from [36] (Section 7) which is closely related to the analysis in [31]. FTRL with uniformly convex regularisation was orginally considered in [4] based on the analysis in [31]. Existence and unicity of the update can be handled along the same lines as Theorem 6.8 in [36] with uniform convexity.

The analysis begins with the following expression for the regret. We refer the reader to [36] for the proof.

**Lemma C.1.** *Lemma 7.1 of [36] Denote* $F_t(x) = \psi_t(x) + \sum_{s=1}^{t-1} \ell_s(x)$ *and set* $x_t \in \arg\min_{x \in V} F_t(x)$. *Consider* $\psi_{T+1} = \psi_T$. *Then, for any* $u \in V$ *we have*

$$\sum_{t=1}^{T} \ell_t(x_t) - \ell_t(u) \le \psi_T(u) - \min_{x \in V} \psi_1(x) + \sum_{t=1}^{T} \Big\{ F_t(x_t) - F_{t+1}(x_{t+1}) + \ell_t(x_t) \Big\} \qquad (4)$$

To bound the terms $F_t(x_t) - F_{t+1}(x_{t+1}) + \ell_t(x_t)$, we use the uniform convexity of the regularisers. In particular, we require the following result on uniformly convex functions, which is an extension of Corollary 7.7 of [36].

**Lemma C.2.** *Let* $f : \mathbb{R}^d \to \mathbb{R}$ *be closed, proper, sub-differentiable and* $\mu$-*uniformly convex of degree* $r$ *w.r.t. a norm* $\|\cdot\|$. *Let* $x^\star = \arg\min_{x \in \mathrm{dom} f} f(x)$. *Then for all* $x \in \mathrm{dom} f$ *and* $g \in \partial f(x)$, *we have*

$$f(x) - f(x^\star) \le \frac{r-1}{r\mu^{1/(r-1)}} \|g\|_\star^{r/(r-1)}.$$

*Proof.* By the uniform convexity of $f$, we have

$$
\begin{aligned}
f(x^\star) &= \min_{z \in \mathrm{dom} f} f(z) \\
&\ge \min_{z \in \mathrm{dom} f} \Big\{ f(x) + \langle g, z - x \rangle + \frac{\mu}{r} \|z - x\|^r \Big\} \\
&\ge f(x) + \min_{z \in \mathbb{R}^d} \Big\{ \langle g, z - x \rangle + \frac{\mu}{r} \|z - x\|^r \Big\} \\
&= f(x) + \min_{z \in \mathbb{R}^d} \Big\{ \langle g, z \rangle + \frac{\mu}{r} \|z\|^r \Big\} \\
&= f(x) - \mu \max_{z \in \mathbb{R}^d} \Big\{ \langle \frac{-g}{\mu}, z \rangle - \frac{1}{r} \|z\|^r \Big\} \\
&= f(x) - \frac{\mu}{r_\star} \Big\| \frac{-g}{\mu} \Big\|_\star^{r_\star} \\
&= f(x) - \mu^{1-r_\star} \frac{\|g\|_\star^{r_\star}}{r_\star} \\
&= f(x) - \frac{r-1}{r\mu^{1/(r-1)}} \|g\|_\star^{r/(r-1)}
\end{aligned}
$$

where we used that the fenchel conjugate of $\frac{\|x\|^r}{r}$ is $\frac{\|x\|_\star^{r_\star}}{r_\star}$ Rearranging gives the result. $\qquad\square$

Since $\psi_t$ is proper, closed, differentiable and $\mu_t$-uniformly convex of degree $r_t$ with respect to $\|\cdot\|_{|t}$ and the losses are proper and convex, $F_t(x) + \ell_t(x) = \psi_t(x) + \sum_{s=1}^{t} \ell_s(x)$ is also proper, closed, sub-differentiable and $\mu_t$-uniformly convex of degree $r_t$ with respect to $\|\cdot\|_{|t}$. Applying Lemma C.2 to $F_t + \ell_t$, we have with $x_t^\star = \arg\min_{x \in V} F_t(x) + \ell_t(x)$

$$
\begin{aligned}
F_t(x_t) - F_{t+1}(x_{t+1}) + \ell_t(x_t) &= \Big( F_t(x_t) + \ell_t(x_t) \Big) - \Big( F_t(x_{t+1}) + \ell_t(x_{t+1}) \Big) + \psi_t(x_{t+1}) - \psi_{t+1}(x_{t+1}) \\
&\le \Big( F_t(x_t) + \ell_t(x_t) \Big) - \Big( F_t(x_t^\star) + \ell_t(x_t^\star) \Big) + \psi_t(x_{t+1}) - \psi_{t+1}(x_{t+1}) \\
&\le \frac{r_t - 1}{r_t \mu_t^{1/(r_t-1)}} \|g_t\|_{|t\star}^{r_t/(r_t-1)} + \psi_t(x_{t+1}) - \psi_{t+1}(x_{t+1}),
\end{aligned}
$$

where we used that $g_t \in \partial(F_t + \ell_t)(x_t)$ since $g_t \in \partial\ell_t(x_t)$ and $x_t = \arg\min_{x \in V} F_t(x)$. We omit some technical details but the steps from [36] extend to our setting. Plugging the above into (4) gives Theorem 2.3.

## C.2 Proof of Corollary 2.4

Since $\psi$ is $\mu$-uniformly convex function on $V$ of degree $r$ with respect to $\|\cdot\|$, then the regulariser used by FTRL in round $t$, $\psi_t = \frac{1}{\eta_{t-1}}\psi$ is $\frac{\mu}{\eta_{t-1}}$-uniformly convex function on $V$ of degree $r$ with respect to $\|\cdot\|$. Since $\eta_t \leq \eta_{t-1}$, $\psi_t(x_{t+1}) - \psi_{t+1}(x_{t+1}) = \left(\frac{1}{\eta_{t-1}} - \frac{1}{\eta_t}\right)\psi(x_{t+1}) \leq 0$. By the Lipschitz condition on the losses, we have $\|g_t\|_\star \leq L_{\|\cdot\|}$. Applying Theorem 2.3, we have

$$\sum_{t=1}^T \ell_t(x_t) - \ell_t(u) \leq \frac{\psi(u)}{\eta_{T-1}} + \frac{L_{\|\cdot\|}^{r_\star}}{r_\star \mu^{r_\star-1}} \sum_{t=1}^T \eta_{t-1}^{r_\star-1}$$

$$\leq \frac{L_{\|\cdot\|} D^{1-1/r_\star}(r_\star-1)^{1/r_\star} T^{1/r_\star}}{\mu^{1/r}} + \frac{L_{\|\cdot\|}^{r_\star}}{r_\star \mu^{r_\star-1}} \sum_{t=1}^T \left(\frac{D^{1/r_\star}\mu^{1/r}}{L_{\|\cdot\|}(r_\star-1)^{1/r_\star}t^{1/r_\star}}\right)^{r_\star-1}$$

$$\leq \frac{L_{\|\cdot\|} D^{1/r}(r_\star-1)^{1/r_\star} T^{1/r_\star}}{\mu^{1/r}} + \frac{L_{\|\cdot\|} D^{1/r}}{r_\star(r_\star-1)^{1/r}\mu^{(r_\star-1)(1-1/r)}} \sum_{t=1}^T \frac{1}{t^{1/r}}$$

$$= \frac{L_{\|\cdot\|} D^{1/r}}{\mu^{1/r}}\left((r_\star-1)^{1/r_\star} T^{1/r_\star} + \frac{1}{r_\star(r_\star-1)^{1/r}} \sum_{t=1}^T \frac{1}{t^{1/r}}\right).$$

Now note that

$$\sum_{t=1}^T \frac{1}{t^{1/r}} \leq \int_0^T \frac{1}{x^{1/r}}dx = \left[\frac{1}{1-1/r}x^{1-1/r}\right]_0^T = r_\star T^{1/r_\star}$$

$$\implies \sum_{t=1}^T \ell_t(x_t) - \ell_t(u) \leq \frac{L_{\|\cdot\|} D^{1/r} T^{1/r_\star}}{\mu^{1/r}}\left((r_\star-1)^{1/r_\star} + \frac{1}{(r_\star-1)^{1/r}}\right).$$

The proof is concluded by noting that $(r_\star-1)^{1/r_\star} + \frac{1}{(r_\star-1)^{1/r}} = r^{1/r}r_\star^{1/r_\star}$. Lemma C.4 was helpful in finding the optimal step-size.

## C.3 Proof of Theorem 2.5

We have $d \leq T$. Let $k = \lfloor T/d \rfloor \geq 1$. Let $Y_{i,j}$ be i.i.d. Rademacher random variables for $1 \leq i \leq d$, $1 \leq j \leq k$, i.e. $\mathbb{P}(Y_{i,j} = 1) = \mathbb{P}(Y_{i,j} = -1) = 1/2$. Let $e_1, ..., e_d$ be the canonical basis of $\mathbb{R}^d$. Define $g_t = LY_{i,j} \cdot e_i$ where $t = k(i-1) + j$ (for $k$ rounds we stick to the same coordinate and draw i.i.d. Rademacher random variables). Denote the point played by $\mathcal{A}$ by $x_t$ and fix the loss to be $\tilde{\ell}_t(x) = g_t^T x$ (for $t > dk$, fix $\tilde{\ell}_t(x) = 0$). The subgradient is $g_t$, which is bounded by $L$ in $\ell_{p_\star}$-norm. The point $x_t$ depends on the losses up to time $t-1$ but not on $\tilde{\ell}_t$ and is independent of $Y_t$, so for all $t$:

$$\mathbb{E}[\tilde{\ell}_t(x_t)] = \mathbb{E}[Y_t L \cdot e_t^T x_t] = \mathbb{E}[Y_t]Le_t^T \mathbb{E}[x_t] = 0 \implies \mathbb{E}[\sum_{t=1}^T \tilde{\ell}_t(x_t)] = 0.$$

On the other hand, $u = -d^{-1/p} \sum_{i=1}^d \text{sign}\left\{\sum_{j=1}^k Y_{i,j}\right\}e_i \in \mathcal{B}_p$ gives

$$\min_{x \in \mathcal{B}_p} \sum_{t=1}^T \tilde{\ell}_t(x) \leq \sum_{t=1}^T \tilde{\ell}_t(u)$$

$$= -Ld^{-1/p}\left(\sum_{i=1}^d \text{sign}\left\{\sum_{j=1}^k Y_{i,j}\right\}e_i\right)^T\left(\sum_{i=1}^d \sum_{j=1}^k Y_{i,j}e_i\right)$$

$$= -Ld^{-1/p} \sum_{i=1}^d \left|\sum_{j=1}^k Y_{i,j}\right|.$$

We now make use of a result from [13] (proof of Lemma 7.2): fix $B > 0$, consider $X = \sum_{i=1}^{B} t_i R_i$ where $t_i$ are positive integers such that $\sum_{i=1}^{B} t_i = k$ and $R_i$ are i.i.d Rademacher random variables. Then $\mathbb{E}[|X|] \geq k/\sqrt{3B}$.

In our case, with $B = k$ and $t_i = 1$ for all $i$, we have that $\mathbb{E}[|\sum_{j=1}^{k} Y_{i,j}|] \geq \sqrt{k/3} \geq \sqrt{T/6d}$ (since $k = \lfloor T/d \rfloor \geq T/2d$ for $T \geq d$) which gives

$$\mathbb{E}\Big[\sum_{t=1}^{T} \tilde{\ell}_t(x_t) - \min_{x \in \mathcal{B}_p} \sum_{t=1}^{T} \tilde{\ell}_t(x)\Big] \geq 0 + Ld^{-1/p} \sum_{i=1}^{d} \sqrt{\frac{T}{6d}} = Ld^{-1/p}\sqrt{\frac{Td}{6}} = L\sqrt{\frac{Td^{1-2/p}}{6}}$$

The result follows by: $\sup_{\ell_1,\dots,\ell_T} R_T \geq \mathbb{E}\Big[\sum_{t=1}^{T} \tilde{\ell}_t(x_t) - \min_{x \in \mathcal{B}_p} \sum_{t=1}^{T} \tilde{\ell}_t(x)\Big] \geq L\sqrt{\frac{Td^{1-2/p}}{6}}$.

## C.4    Proof of Theorem 2.7

We have $d > T$. For $t \in \{1, \dots, T\}$, let $Y_t$ be i.i.d. Rademacher random variables, i.e. $\mathbb{P}(Y_t = 1) = \mathbb{P}(Y_t = -1) = 1/2$. Let $e_1, \dots, e_d$ be the canonical basis of $\mathbb{R}^d$. At time-step $t$, denote the point played by $\mathcal{A}$ by $x_t$ and fix the loss to be $\tilde{\ell}_t(x) = Y_t L e_t^T x$. The subgradient is $Y_t L e_t$, which is bounded by $L$ in $\ell_{p_\star}$-norm. The point $x_t$ depends on the losses up to time $t-1$ but not on $\tilde{\ell}_t$ and is independent of $Y_t$, so for all $t$:

$$\mathbb{E}[\tilde{\ell}_t(x_t)] = \mathbb{E}[Y_t L e_t^T x_t] = \mathbb{E}[Y_t] L e_t^T \mathbb{E}[x_t] = 0 \implies \mathbb{E}[\sum_{t=1}^{T} \tilde{\ell}_t(x_t)] = 0.$$

On the other hand,

$$\min_{x \in \mathcal{B}_p} \sum_{t=1}^{T} \tilde{\ell}_t(x) = L \min_{x \in \mathcal{B}_p} x^T \Big(\sum_{t=1}^{T} Y_t e_t\Big)$$

is attained at $x = -T^{-1/p} \sum_{t=1}^{T} Y_t e_t \in \mathcal{B}_p$, giving

$$\min_{x \in \mathcal{B}_p} \sum_{t=1}^{T} \tilde{\ell}_t(x) = -LT^{-1/p} \sum_{t,t'=1}^{T} Y_t Y_{t'} e_t^T e_{t'} = -LT^{-1/p} \sum_{t=1}^{T} Y_t^2 = -LT^{1-1/p} = -LT^{1/p_\star}.$$

The result follows by: $\sup_{\ell_1,\dots,\ell_T} R_T \geq \mathbb{E}\Big[\sum_{t=1}^{T} \tilde{\ell}_t(x_t) - \min_{x \in \mathcal{B}_p} \sum_{t=1}^{T} \tilde{\ell}_t(x)\Big] = LT^{1/p_\star}$.

## C.5    Uniform Convexity of $p$-OMD's regulariser

In this section, we provide the proof of Proposition 2.6 on the $\mu$-uniform convexity of degree $p$ of $\psi(x) = \frac{1}{p}\|x\|_p^p$ on $\mathcal{B}_p$ for $p > 2$.

Consider $x, y \in \mathcal{B}_p$. Following the steps in Remark 2.1 of [46], using convexity of $\psi$ we have for $\lambda \in [0, 1/2]$,

$$\psi(\lambda x + (1-\lambda)y) = \psi\Big(2\lambda\Big(\frac{x+y}{2}\Big) + (1-2\lambda)y\Big)$$
$$\leq 2\lambda\psi\Big(\frac{x+y}{2}\Big) + (1-2\lambda)\psi(y)$$
$$= \frac{2\lambda}{p}\Big\|\frac{x+y}{2}\Big\|_p^p + \frac{(1-2\lambda)}{p}\|y\|_p^p.$$

From Clarkson's inequality (equation (2.1) in [2]), we have that

$$\Big\|\frac{x+y}{2}\Big\|_p^p + \Big\|\frac{x-y}{2}\Big\|_p^p \leq \frac{\|x\|_p^p}{2} + \frac{\|y\|_p^p}{2}.$$

Using this in the above we have

$$\psi(\lambda x + (1-\lambda)y) \leq \frac{2\lambda}{p}\frac{\|x\|_p^p}{2} + \frac{2\lambda}{p}\frac{\|y\|_p^p}{2} - \frac{2\lambda}{p}\left\|\frac{x-y}{2}\right\|_p^p + \frac{(1-2\lambda)}{p}\|y\|_p^p$$

$$= \lambda\frac{\|x\|_p^p}{p} + (1-\lambda)\frac{\|y\|_p^p}{p} - \frac{2\lambda}{p}\left\|\frac{x-y}{2}\right\|_p^p$$

$$\leq \lambda\psi(x) + (1-\lambda)\psi(y) - \frac{2\lambda(1-\lambda)}{p}\left\|\frac{x-y}{2}\right\|_p^p. \tag{5}$$

This is an alternative characterisation of uniform convexity, we now show (following steps in Definition 3.2 of [24]) that it is equivalent to our original one (Definition 2.2). From the convexity and differentiability of $\psi$,

$$\psi(y) + \lambda\langle\nabla\psi(y), x-y\rangle = \psi(y) + \langle\nabla\psi(y), [y+\lambda(x-y)] - y\rangle$$

$$\leq \psi(y + \lambda(x-y))$$

$$\leq \lambda\psi(x) + (1-\lambda)\psi(y) - \frac{2\lambda(1-\lambda)}{p}\left\|\frac{x-y}{2}\right\|_p^p.$$

Rearrenging,

$$\implies \lambda\langle\nabla\psi(y), x-y\rangle \leq \lambda(\psi(x) - \psi(y)) - \frac{2\lambda(1-\lambda)}{p}\left\|\frac{x-y}{2}\right\|_p^p$$

$$\implies \langle\nabla\psi(y), x-y\rangle \leq (\psi(x) - \psi(y)) - \frac{2(1-\lambda)}{p}\left\|\frac{x-y}{2}\right\|_p^p$$

$$\implies \psi(x) \geq \psi(y) + \langle\nabla\psi(y), x-y\rangle + \frac{2}{p}\left\|\frac{x-y}{2}\right\|_p^p,$$

as $\lambda \to 0$. So for any $x, y \in \mathcal{B}_p$ we have the condition of uniform convexity with $\mu = 2^{1-p}$. $\quad\square$

**Remark C.3.** *It is not possible to get the parameter of uniform convexity $\mu = 1$. Consider the 1-dimensonal case, $x = 1, y = -1$:*

$$\psi(x) + \langle\nabla\psi(x), y-x\rangle + \frac{\mu}{p}\|x-y\|_p^p = \frac{1}{p} + (y-x) + \frac{\mu}{p}(1+1)^p$$

$$= \frac{1}{p} + (-1-1) + \frac{\mu 2^p}{p}$$

$$= \frac{1 - 2p + \mu 2^p}{p}.$$

*This is less or equal than $\psi(y) = \frac{1}{p}$ when*

$$\frac{1 - 2p + \mu 2^p}{p} \leq \frac{1}{p} \implies 1 - 2p + \mu 2^p \leq 1 \implies \mu \leq p2^{1-p}.$$

*So our constant may be loose by a factor of $p$ but $\mu = 1$ is not possible since $p2^{1-p} < 1$ as soon as $p > 2$.*

*In fact, we can slightly improve $\mu$ from $\frac{1}{2^{p-1}}$ to $\frac{1}{2^{p-1}-1}$ (we present our results with $\frac{1}{2^{p-1}}$ because it only changes the results by a small constant and slightly avoids clutter). Here is how: In the first step of the proof, we used convexity of $\psi$ to obtain the following bound,*

$$\psi\left(2\lambda\left(\frac{x+y}{2}\right) + (1-2\lambda)y\right) \leq 2\lambda\psi\left(\frac{x+y}{2}\right) + (1-2\lambda)\psi(y).$$

*However, from (5), we have that*

$$\psi(\lambda x + (1-\lambda)y) \leq \lambda\psi(x) + (1-\lambda)\psi(y) - \frac{2\lambda}{p}\left\|\frac{x-y}{2}\right\|_p^p, \tag{6}$$

*and this provides a tighter bound than just using convexity:*

$$\psi\left(2\lambda\left(\frac{x+y}{2}\right) + (1-2\lambda)y\right) \leq 2\lambda\psi\left(\frac{x+y}{2}\right) + (1-2\lambda)\psi(y) - \frac{2\cdot2\lambda}{p}\left\|\frac{(x+y)/2 - y}{2}\right\|_p^p$$

$$\leq \lambda\psi(x) + (1-\lambda)\psi(y) - \frac{2\lambda}{p}\left\|\frac{x-y}{2}\right\|_p^p\left(1 + 2^{1-p}\right),$$

*where we followed similar steps as in the original proof (Clarkson's inequality). This provides an even tighter bound than (6) and applying these tighter bounds recursively gives*

$$\psi\Big(2\lambda\Big(\frac{x+y}{2}\Big) + (1-2\lambda)y\Big) \le \lambda\psi(x) + (1-\lambda)\psi(y) - \frac{2\lambda}{p}\Big\|\frac{x-y}{2}\Big\|_p^p \cdot \frac{1}{1-2^{1-p}},$$

*using that $\sum_{t=0}^{\infty}(2^{1-p})^t = 1/(1-2^{1-p})$. Following the same steps for the remainder of the proof gives uniform convexity of $\psi$ with $\mu = \frac{1}{2^{p-1}-1}$.*

## C.6   Helper lemma

**Lemma C.4.** *Fix $a, b > 0$, $n > 1$. Let $f(x) = \frac{a}{x} + bx^{n-1}$ for $x > 0$. Then $f$ is minimised at $x^\star = (a/b(n-1))^{1/n}$ and*

$$f(x^\star) = a^{1-1/n}b^{1/n}\Big(\frac{n}{n-1}\Big)^{(n-1)/n}n^{1/n}.$$

*Proof.* Setting the derivative of $f$ to 0 and solving gives

$$-\frac{a}{x^2} + (n-1)bx^{n-2} = 0 \implies x^\star = \Big(\frac{a}{(n-1)b}\Big)^{1/n}.$$

Plugging into $f$ gives

$$
\begin{aligned}
f(x^\star) &= a \cdot \Big(\frac{(n-1)b}{a}\Big)^{1/n} + b \cdot \Big(\frac{a}{(n-1)b}\Big)^{(n-1)/n} \\
&= a^{1-1/n}(n-1)^{1/n}b^{1/n} + b^{1-1+1/n}a^{1-1/n}(n-1)^{1/n-1} \\
&= a^{1-1/n}b^{1/n}(n-1)^{1/n}\Big(1 + \frac{1}{n-1}\Big) \\
&= a^{1-1/n}b^{1/n}(n-1)^{1/n}\frac{n}{n-1} \\
&= a^{1-1/n}b^{1/n}\Big(\frac{n}{n-1}\Big)^{(n-1)/n}n^{1/n}.
\end{aligned}
$$

$\square$

# D Proofs for Section 3

## D.1 Proof of Theorem 3.1

If $T \leq t_0$, then we have FTRL with fixed regulariser $\phi_p$ and from Corollary 2.4 we have $R_T \leq L(2p_*T)^{1/p_*}$ as in Section 2.3. If $T > t_0$, Theorem 2.3 gives

$$
R_T \leq \psi_T(u) - \min_{x \in V} \psi_1(x) + \sum_{t=1}^{T} \left\{ \frac{(r_t - 1)}{r_t \mu_t^{1/(r_t-1)}} \|g_t\|_{|t\star}^{r_t/(r_t-1)} + \psi_t(x_{t+1}) - \psi_{t+1}(x_{t+1}) \right\}
$$

$$
\leq \frac{\phi_2(u)}{\eta_{T-1}} - \min_{x \in V} \phi_p(x) + \sum_{t=1}^{t_0} \left\{ 2\frac{\eta_{t-1}^{p_\star - 1}}{p_\star} \|g_t\|_{p_\star}^{p_\star} \right\} + \sum_{t=1}^{t_0-1} \left\{ \phi_p(x_{t+1}) \left( \frac{1}{\eta_{t-1}} - \frac{1}{\eta_t} \right) \right\}
$$

$$
+ \frac{\phi_p(x_{t_0+1})}{\eta_{t_0-1}} - \frac{\phi_2(x_{t_0+1})}{\eta_{t_0}} + \sum_{t=t_0+1}^{T} \left\{ \frac{\eta_{t-1}}{2} \|g_t\|_2^2 + \phi_2(x_{t+1}) \left( \frac{1}{\eta_{t-1}} - \frac{1}{\eta_t} \right) \right\}
$$

$$
\leq \frac{\sup_{x \in \mathcal{B}_p} \phi_p(x)}{\eta_{t_0-1}} + \sum_{t=1}^{t_0} \left\{ 2\frac{\eta_{t-1}^{p_\star - 1}}{p_\star} \|g_t\|_{p_\star}^{p_\star} \right\} + \frac{\phi_2(u)}{\eta_{T-1}} + \sum_{t=t_0+1}^{T} \left\{ \frac{\eta_{t-1}}{2} \|g_t\|_2^2 \right\}.
$$

The first two terms correspond to the regret of FTRL with fixed $\phi_p$ regularisation on $t_0$ rounds. Substituting the values of $\eta_{t-1}$ and some algebra gives (see similar steps in the proof of Corollary 2.4)

$$
\frac{\sup_{x \in \mathcal{B}_p} \phi_p(x)}{\eta_{t_0-1}} + \sum_{t=1}^{t_0} \left\{ 2\frac{\eta_{t-1}^{p_\star - 1}}{p_\star} \|g_t\|_{p_\star}^{p_\star} \right\} \leq L(2p_\star t_0)^{1/p_\star}.
$$

The last two terms correspond to the regret of FTRL with fixed $\phi_2$ regularisation over the remaining $T - t_0$ rounds.

$$
\frac{\phi_2(u)}{\eta_{T-1}} + \sum_{t=t_0+1}^{T} \left\{ \frac{\eta_{t-1}}{2} \|g_t\|_2^2 \right\} = \frac{L\sqrt{d^{1-2/p}T}}{\sqrt{2}} + \frac{L\sqrt{d^{1-2/p}}}{2\sqrt{2}} \sum_{t=t_0+1}^{T} \frac{1}{\sqrt{t}}
$$

$$
\leq \frac{L\sqrt{d^{1-2/p}T}}{\sqrt{2}} + \frac{L\sqrt{d^{1-2/p}T}}{\sqrt{2}} - \frac{L\sqrt{d^{1-2/p}t_0}}{\sqrt{2}}
$$

$$
= L\sqrt{2d^{1-2/p}T} - L\sqrt{d^{1-2/p}t_0/2},
$$

where we used that $\sum_{t=t_0+1}^{T} \frac{1}{\sqrt{t}} \leq \int_{t_0}^{T} \frac{1}{\sqrt{x}} dx = \left[ 2\sqrt{x} \right]_{t_0}^{T} = 2(\sqrt{T} - \sqrt{t_0})$. Combining, we have

$$
R_T \leq L\sqrt{2d^{1-2/p}T} + L(2p_\star t_0)^{1/p_\star} - L\sqrt{d^{1-2/p}t_0/2}.
$$

The proof is concluded by $t_0 = 3^{-2p/(p-2)}d$ guaranteeing $(2p_\star t_0)^{1/p_\star} - \sqrt{d^{1-2/p}t_0/2} < 0$ since

$$
(2p_\star t_0)^{1/p_\star} \leq \frac{3}{\sqrt{2}} t_0^{1/p_\star} = 3 t_0^{\frac{p-1}{p} - \frac{1}{2}} \sqrt{t_0/2} = 3 t_0^{\frac{p-2}{2p}} \sqrt{t_0/2} = \sqrt{d^{1-2/p}t_0/2}.
$$

# E Proofs for Section 4

## E.1 Loss construction for proofs

Many of the proofs in this section share the same loss construction, which we describe here. Assume that T is divisible by $4$ (use $T - 1$, $T - 2$ or $T - 3$ if not). We define the following linear losses $\ell_t(x) = L \cdot x^T g_t$ where $g_t \in \mathcal{B}_{p_\star}$ is defined as

$$g_t = \begin{cases} (-1)^t \cdot e_1, & t \leq \frac{T}{2}, \\ -v, & t > \frac{T}{2}, \end{cases}$$

where $v \in \mathcal{B}_{p_\star}$ is a vector with equal entries defined as $v_{t,i} = d^{-1/p_\star}$ (so that $\|v\|_{p_\star} = 1$). Note that $\|v\|_p = d^{1/p - 1/p_\star}$. The cumulative loss of the competitor:

$$\sum_{t=1}^{T} \ell_t(x) = \frac{LT}{2} x^T v \implies \min_{x \in \mathcal{B}_p} \sum_{t=1}^{T} \ell_t(x) = -\frac{LT}{2} \frac{v^T v}{\|v\|_p} = -\frac{LT}{2} \frac{d^{1-2/p_\star}}{d^{1/p-1/p_\star}} = -\frac{LT}{2}. \quad (7)$$

The cumulative sum of sub-gradients used in the FTRL update:

$$L \sum_{s=1}^{t-1} g_s = L \cdot \begin{cases} -e_1, & \text{if } t \leq \frac{T}{2} \text{ is even,} \\ 0, & \text{if } t \leq \frac{T}{2} \text{ is odd,} \\ -\left(t - 1 - \frac{T}{2}\right) \cdot v, & \text{if } t > \frac{T}{2}. \end{cases} \quad (8)$$

## E.2 Proofs of Proposition 4.1 and Proposition 4.5

The two propositions are special cases of the following proposition.

**Proposition E.1.** *For $r \in [2, p]$, define $\phi_r(x) = \frac{1}{r}\|x\|_r^r$. There exists a sequence of linear $L$-Lipschitz losses (in $\ell_p$-norm) for which FTRL with regulariser $\psi_t(x) = \frac{1}{\eta_{t-1}}\phi_r(x)$ and any sequence of decreasing $\eta_{t-1}$ suffers regret*

$$R_T \geq L \cdot \min\left(\frac{T}{8r}, \frac{d^{(r_\star - p_\star)/r_\star p_\star} T^{1/r_\star}}{8}\right).$$

We now prove this proposition. The loss construction is described in Appendix E.1. From Lemma B.1,

$$x_{t+1} = \arg\min_{x \in \mathcal{B}_p} D_{\phi_r}\left(x, \text{sign}\left(-\eta_t \sum_{s=1}^{t} g_s\right)\left|-\eta_t \sum_{s=1}^{t} g_s\right|^{r_\star - 1}\right).$$

Define $\alpha_{t-1} = \min\{1, \eta_{t-1}\}$. Using (8), the points played by FTRL on are given by

- For $t \leq T/2$ odd: $x_t = 0$
- For $t \leq T/2$ even: $x_t = \alpha_{t-1}^{r_\star - 1} \cdot e_1$ by Lemma B.3.
- For $t > T/2$:

$$x_t = \min\left(\frac{1}{\|w\|_p}, \left\{\eta_{t-1} d^{-1/p_\star}\left(t - 1 - \frac{T}{2}\right)\right\}^{r_\star - 1}\right) \cdot w$$

$$= \min\left(1, d^{1-r_\star/p_\star}\left\{\eta_{t-1}\left(t - 1 - \frac{T}{2}\right)\right\}^{r_\star - 1}\right) \cdot \frac{v}{\|v\|_p}$$

by Lemma B.2 where $w$ is a vector with equal entries equal to $1$.

Fix $\eta = \eta_{T/2-1}$, $\alpha = \min\{1, \eta\}$. Using that $\eta_{t-1} \geq \eta_t$, the loss in the first half of the rounds is lower bounded as

$$\sum_{t=1}^{T/2} \ell_t(x_t) = \sum_{k=1}^{T/4} \ell_{2k}(x_{2k}) = \sum_{k=1}^{T/4} \alpha_{2k-1}^{r_\star - 1} e_1^T x_{2k} = \sum_{k=1}^{T/4} \alpha_{2k-1}^{r_\star - 1} e_1^T e_1 \geq \frac{\alpha^{r_\star - 1} T}{4}.$$

If $\alpha \geq 1$, we have $R_T \geq \frac{T}{8} \geq \frac{T}{4r}$ and we are done. So for the rest we assume that $\alpha = \eta \leq 1/2$. Let $k^\star = \lfloor d^{(r_\star/p_\star - 1)/(r_\star - 1)}/\eta \rfloor$, $m = \min(k^\star, T/2 - 1)$. Note that $v^T v = d^{1 - 2/p_\star} = \|v\|_p$. The losses in the second half is lower-bounded as

$$\sum_{t = T/2 + 1}^{T} \ell_t(x_t) = -\sum_{t = T/2 + 1}^{T} x_t^T v \geq -\sum_{k=1}^{T/2 - 1} \min\left\{1, d^{1 - r_\star/p_\star}(\eta k)^{r_\star - 1}\right\} \cdot \frac{v^T v}{\|v\|_p}$$

$$= -\sum_{k=1}^{T/2 - 1} \min\left\{1, d^{1 - r_\star/p_\star}(\eta k)^{r_\star - 1}\right\}$$

$$= -d^{1 - r_\star/p_\star} \sum_{k=1}^{m} (\eta k)^{r_\star - 1} - \left(\frac{T}{2} - 1 - m\right).$$

We bound the sum with an integral as follows,

$$\sum_{k=1}^{m} k^{r_\star - 1} \leq \int_0^m (x + 1)^{r_\star - 1}\, dx = \frac{1}{r_\star}\left[(x + 1)^{r_\star}\right]_0^m \leq \frac{1}{r_\star}(m + 1)^{r_\star}.$$

We get

$$\sum_{t = T/2 + 1}^{T} \ell_t(x_t) \geq -d^{1 - r_\star/p_\star}\frac{\eta^{r_\star - 1}}{r_\star}(m + 1)^{r_\star} - \left(\frac{T}{2} - 1 - m\right).$$

Using the cumulative loss of the competitor from (7), the regret is

$$R_T \geq \frac{T}{2} + \frac{\eta^{r_\star - 1} T}{4} - d^{1 - r_\star/p_\star}\frac{\eta^{r_\star - 1}}{r_\star}(m + 1)^{r_\star} - \left(\frac{T}{2} - 1 - m\right)$$

$$= \frac{\eta^{r_\star - 1} T}{4} - d^{1 - r_\star/p_\star}\frac{\eta^{r_\star - 1}}{r_\star}(m + 1)^{r_\star} + (1 + m).$$

Let's consider two cases:

- $k^\star \geq T/2 - 1$: $m = T/2 - 1$. By the definition of $k^\star$:

$$\frac{d^{(r_\star/p_\star - 1)/(r_\star - 1)}}{\eta} \geq \left\lfloor \frac{d^{(r_\star/p_\star - 1)/(r_\star - 1)}}{\eta} \right\rfloor = k^\star \geq \frac{T}{2} - 1 \implies \frac{\eta}{d^{(r_\star/p_\star - 1)/(r_\star - 1)}}\left(\frac{T}{2} - 1\right) \leq 1$$

$$\implies \frac{\eta}{d^{(r_\star/p_\star - 1)/(r_\star - 1)}}\frac{T}{2} \leq 1 + \frac{\eta}{d^{(r_\star/p_\star - 1)/(r_\star - 1)}} \leq \frac{3}{2},$$

since $\eta \leq 1/2$ and $d^{(r_\star/p_\star - 1)/(r_\star - 1)} \geq 1$ (recall $r \leq p$). Using this in the regret, we get

$$R_T \geq -d^{1 - r_\star/p_\star}\frac{\eta^{r_\star - 1}}{r_\star}\left(\frac{T}{2}\right)^{r_\star} + \frac{T}{2}$$

$$= -\frac{1}{r_\star}\left(\frac{\eta}{d^{(r_\star/p_\star - 1)/(r_\star - 1)}}\frac{T}{2}\right)^{r_\star - 1}\frac{T}{2} + \frac{T}{2}$$

$$\geq -\frac{1}{r_\star}\left(\frac{3}{2}\right)^{r_\star - 1}\frac{T}{2} + \frac{T}{2}$$

$$= \frac{T}{2}\left(1 - \frac{1}{r_\star}\left(\frac{3}{2}\right)^{r_\star - 1}\right)$$

$$\geq \frac{T}{4}\left(1 - \frac{1}{r_\star}\right) = \frac{T}{4r},$$

where we used that $r_\star \in [1, 2]$ and that $f(x) = 1 - \frac{(3/2)^{x-1}}{x} \geq \frac{1}{2}(1 - 1/x)$ for $x \in [1, 2]$.

- $k^\star < T/2 - 1$: $m = k^\star$. By the definition of $k^\star$:

$$\frac{d^{(r_\star/p_\star - 1)/(r_\star - 1)}}{\eta} \geq \left\lfloor \frac{d^{(r_\star/p_\star - 1)/(r_\star - 1)}}{\eta} \right\rfloor = k^\star \implies \frac{\eta}{d^{(r_\star/p_\star - 1)/(r_\star - 1)}}k^\star \leq 1$$

$$\implies \frac{\eta}{d^{(r_\star/p_\star - 1)/(r_\star - 1)}}(k^\star + 1) \leq 1 + \frac{\eta}{d^{(r_\star/p_\star - 1)/(r_\star - 1)}} \leq \frac{3}{2},$$

since again $\eta \leq 1/2$ and $d^{(r_\star/p_\star-1)/(r_\star-1)} \geq 1$. We also have $k^\star + 1 \geq \frac{d^{(r_\star/p_\star-1)/(r_\star-1)}}{\eta}$. Using this in the regret, we get

$$
\begin{aligned}
R_T &= \frac{\eta^{r_\star-1}T}{4} - d^{1-r_\star/p_\star}\frac{\eta^{r_\star-1}}{r_\star}(k^\star+1)^{r_\star} + (1+k^\star) \\
&= \frac{\eta^{r_\star-1}T}{4} - \frac{k^\star+1}{r_\star}\Big(\frac{\eta}{d^{(r_\star/p_\star-1)/(r_\star-1)}}(k^\star+1)\Big)^{r_\star-1} + (1+k^\star) \\
&\geq \frac{\eta^{r_\star-1}T}{4} - \frac{1+k^\star}{r_\star}\Big(\frac{3}{2}\Big)^{r_\star-1} + (1+k^\star) \\
&= \frac{\eta^{r_\star-1}T}{4} + (1+k^\star)\Big(1 - \frac{1}{r_\star}\Big(\frac{3}{2}\Big)^{r_\star-1}\Big) \\
&\geq \frac{\eta^{r_\star-1}T}{4} + \frac{(1+k^\star)}{2}\Big(1 - \frac{1}{r_\star}\Big) \\
&= \frac{\eta^{r_\star-1}T}{4} + \frac{d^{(r_\star/p_\star-1)/(r_\star-1)}}{2r\eta} \\
&\geq \frac{r_\star^{1/r_\star}}{2^{1/r+2/r_\star}}d^{(r_\star/p_\star-1)/r_\star}T^{1/r_\star} \geq \frac{d^{(r_\star-p_\star)/r_\star p_\star}T^{1/r_\star}}{4}
\end{aligned}
$$

where again we used that $1 - \frac{1}{r_\star}\Big(\frac{3}{2}\Big)^{r_\star-1} \geq \frac{1}{2}(1-1/r_\star)$ since $r_\star \in [1,2]$ and in the lstar step we minimised over $\eta$ using Lemma C.4.

Combining both cases, we have that $R_T \geq \min\Big(\frac{T}{4r}, \frac{d^{(r_\star-p_\star)/r_\star p_\star}T^{1/r_\star}}{4}\Big)$. If $T$ is not divisible by 4 and we use $T-1$, $T-2$ or $T-3$, we have $R_T \geq \min\Big(\frac{T-3}{4r}, \frac{d^{(r_\star-p_\star)/r_\star p_\star}(T-3)^{1/r_\star}}{4}\Big) \geq \min\Big(\frac{T}{8r}, \frac{d^{(r_\star-p_\star)/r_\star p_\star}(T-3)^{1/r_\star}}{8}\Big)$ for $T \geq 6$, concluding the proof.

### E.3 Proof of Lemma 4.3

Consider $a \in \arg\min_{z\in\mathbb{R}} \psi(x_1, ..., x_{i-1}, z, x_{i+1}, ...x_d)$. Since $\psi$ is sign-invariant, $-a$ is also in the argmin. Consider $g(z) = \psi(x_1, ..., x_{i-1}, z, x_{i+1}, ...x_d)$. It is straightforward to show that the strong-convexity of $\psi$ applies $g$. By convexity, we have

$$
g(0) = g\Big(\frac{1}{2}a + \frac{1}{2}(-a)\Big) \leq \frac{1}{2}g(a) + \frac{1}{2}g(-a) = g(a) = \min_{z\in\mathbb{R}} g(z),
$$

and by strong convexity, it is actually the unique minimiser. Hence

$$
0 = \arg\min_{z\in\mathbb{R}} \psi(..., x_{i-1}, z, x_{i+1}, ...) \implies \frac{\partial\psi(x)}{\partial x_i}\Big|_{x_i=0} = 0
$$

$$
\implies \nabla\psi(x)^T e_i = 0 \quad \text{for any } x \in \mathcal{B}_p \text{ s.t. } x_i = 0. \quad (9)
$$

For a set $S$ and a vector $x$, denote $x_{-S}$ the vector $x$ with the coordinates in $S$ replaced by $0$. Denote $S_n = \{1, ..., n\}$. We prove the following claim by induction on $n \leq d$:

$$
\psi(x) \geq \psi(x_{-S_n}) + \frac{\mu}{2}\sum_{i=1}^{n} x_i^2.
$$

**Base Case:**, by strong convexity

$$
\begin{aligned}
\psi(x) &\geq \psi(x_{-\{1\}}) + \langle\nabla\psi(x_{-\{1\}}), x - x_{-\{1\}}\rangle + \frac{\mu}{2}\|x - x_{-\{1\}}\|^2 \\
&= \psi(x_{-\{1\}}) + \langle\nabla\psi(x_{-\{1\}}), x_1 e_1\rangle + \frac{x_1^2\mu}{2} \\
&= \psi(x_{-\{1\}}) + \frac{x_1^2\mu}{2} \quad \text{using (9).}
\end{aligned}
$$

**Inductive Step:** suppose true for $k$. Similarly to the base case: by strong convexity,

$$\psi(x_{-S_n}) \geq \psi(x_{-S_{n+1}}) + \langle \nabla\psi(x_{-S_{n+1}}), x_{-S_n} - x_{-S_{n+1}} \rangle + \frac{\mu}{2}\|x_{-S_n} - x_{-S_{n+1}}\|^2$$

$$= \psi(x_{-S_{n+1}}) + x_{n+1}\langle \nabla\psi(x_{-S_{n+1}}), e_{n+1} \rangle + \frac{\mu}{2}x_{n+1}^2$$

$$= \psi(x_{-S_{n+1}}) + \frac{\mu}{2}x_{n+1}^2 \quad \text{using (9).}$$

The result follows by the inductive hypothesis:

$$\psi(x) \geq \psi(x_{-S_n}) + \frac{\mu}{2}\sum_{i=1}^{n} x_i^2$$

$$\geq \psi(x_{-S_{n+1}}) + \frac{\mu}{2}x_{n+1}^2 + \frac{\mu}{2}\sum_{i=1}^{n} x_i^2$$

$$\geq \psi(x_{-S_{n+1}}) + \frac{\mu}{2}\sum_{i=1}^{n+1} x_i^2.$$

When $n = d$, we have $\psi(x) \geq \frac{\mu}{2}\|x\|_2^2$.

### E.4 Proof of Lemma 4.4

As discussed in Remark 4.9, we prove a more general version of Lemma 4.4 for coordinate-wise step-sizes, where the FTRL update is allowed to have a different step-size $\eta_{t-1,i}$ for each coordinate: $x_t = \arg\min_{x \in V}\{\psi(x) + \sum_{i=1}^{d} \eta_{t-1,i} \cdot x_i \sum_{s=1}^{t-1} g_{s,i}\}$.

We consider a slight variation of the loss construction described in Appendix E.1: Assume that T is divisible by 4 (use $T-1$, $T-2$ or $T-3$ if not). We define the following linear losses $\ell_t(x) = L \cdot x^T g_t$ where $g_t \in \mathcal{B}_{p_\star}$ is defined as

$$g_t = \begin{cases} (-1)^t \cdot v, & t \leq 2 \\ (-1)^t \cdot e_{i(t)}, & 2 < t \leq \frac{T}{2}, \\ -v, & t > \frac{T}{2}, \end{cases}$$

where $v \in \mathcal{B}_{p_\star}$ is a vector with equal entries defined as $v_{t,i} = d^{-1/p_\star}$ (so that $\|v\|_{p_\star} = 1$) and $i(t) = \arg\max_{i \in [d]} \eta_{2\lfloor(t-1)/2\rfloor,i}$ (i.e. the coordinate of the largest step-size in the previous even round). Note that $\|v\|_p = d^{1/p-1/p_\star}$. The cumulative loss of the competitor:

$$\sum_{t=1}^{T} \ell_t(x) = \frac{LT}{2}x^T v \implies \min_{x \in \mathcal{B}_p} \sum_{t=1}^{T} \ell_t(x) = -\frac{LT}{2}\frac{v^T v}{\|v\|_p} = -\frac{LT}{2}\frac{d^{1-2/p_\star}}{d^{1/p-1/p_\star}} = -\frac{LT}{2}. \tag{10}$$

The cumulative sum of sub-gradients used in the FTRL update:

$$L\sum_{s=1}^{t-1} g_s = L \cdot \begin{cases} -e_{i(t)}, & \text{if } t \leq \frac{T}{2} \text{ is even,} \\ 0, & \text{if } t \leq \frac{T}{2} \text{ is odd,} \\ -\left(t-1-\frac{T}{2}\right) \cdot v, & \text{if } t > \frac{T}{2}. \end{cases} \tag{11}$$

- First we consider $2 < t \leq T/2$. When $t$ is odd, $x_t = \arg\min_{x \in \mathcal{B}_p} \psi(x) = 0$. When $t$ is even,

$$x_t = \arg\min_{x \in \mathcal{B}_p}\left\{\psi(x) - \eta_{t-1,i(t)}Le_{i(t)}^T x\right\}$$

$$\implies \psi(x_t) - \eta_{t-1,i(t)}Lx_t^T e_{i(t)} \leq \psi(e_{i(t)}) - \eta_{t-1,i(t)}Le_{i(t)}^T e_{i(t)} = 1 - L\eta_{t-1,i(t)}$$

$$\implies \ell_t(x_t) = x_t^T e_{i(t)} \geq 1 - \frac{1}{\eta_{t-1,i(t)}L} \geq 1 - \frac{1}{\eta_{t-1,i(T/2)}L} \quad \text{by def of } i(t)$$

$$\geq 1 - \frac{1}{\eta_{T/2,i(T/2)}L} \geq 1/2,$$

when $\eta := \eta_{T/2,i(T/2)} = \max_{i \in [d]} \eta_{T/2,i} \geq 2/L$. So we have ($-1$ accounts for first 2 rounds not being like the rest)

$$\sum_{t=1}^{T/2} g_t^T x_t \geq \begin{cases} T/4 - 1, & \text{if } \eta \geq 2/L \\ 0, & \text{if } \eta < 2/L. \end{cases}$$

Hence if $\eta \geq 2/L$, we have $R_T \geq LT/4 - 1$ and the statement of the theorem holds. If $\eta < 2/L$, we look to the second half of the rounds.

- Let's now consider $t > T/2$ and assume $\eta < 2/L$. Note that by definition of $\eta$, we have the for all $t \geq T/2$ and for all $i \in [d]$, $\eta_{t,i} \leq \eta \leq 2/L$. Fix $\beta_t = t - T/2 - 1$. The FTRL update is

$$x_t = \arg\min_{x \in \mathcal{B}_p} \Big\{ \psi(x) - L\beta_t \cdot x^T(\eta_{t-1} \odot v) \Big\}.$$

Let $u = v/\|v\|_p$ be the competitor. We can write $x_t = \lambda_t u + \alpha_t u^\perp$ ($\lambda_t > 0$) as a component in the direction of $u$ and a component orthogonal to $u$. We have

$$\psi(x_t) \geq \frac{\mu}{2}\|x_t\|_2^2 = \frac{\mu}{2}(\lambda_t^2\|u\|_2^2 + \alpha_t^2\|u^\perp\|_2^2) \geq \frac{1}{2}\lambda_t^2 \mu d^{1-2/p}.$$

Now from the FTRL update, (in the first implication, we use that $\eta_{t-1,i} \leq \eta$ and $x_{t,i} \geq 0, v_i \geq 0$)

$$\psi(x_t) - L\beta_t x_t^T(\eta_{t-1} \odot v) \leq 0 \implies \frac{1}{2}\lambda_t^2 \mu d^{1-2/p} \leq \eta L\beta_t x_t^T v$$

$$\implies \frac{1}{2}\lambda_t^2 \mu d^{1-2/p} \leq \eta L\beta_t \lambda_t$$

$$\implies \lambda_t \leq \frac{2\eta L\beta_t}{\mu d^{1-2/p}}$$

$$\implies \ell_t(x_t) = -L \cdot v^T x_t = -L\lambda_t \geq -L\frac{2\eta L\beta_t}{\mu d^{1-2/p}} \geq -L\frac{4\beta_t}{\mu d^{1-2/p}},$$

since $\eta \leq 2/L$. If $d \geq (4T/\mu)^{p/(p-2)}$, we have for all $t \leq T$

$$\ell_t(x_t) \geq -L\frac{4\beta_t}{\mu d^{1-2/p}} \geq -L\frac{\beta_t}{T} \geq -\frac{L}{2}$$

$$\implies R_T \geq \frac{LT}{2} + \sum_{t=T/2+1}^{T} \ell_t(x_t) \geq \frac{LT}{2} - \frac{LT}{4} = \frac{LT}{4}.$$

If $T$ is not divisible by $4$ and we use $T-1, T-2$ or $T-3$, we have $R_T \geq \frac{L(T-3)}{4} - 1 \geq \frac{LT}{8}$ for $T \geq 6 + \frac{8}{L}$, concluding the proof.

### E.5 Proof of Lemma 4.7

Assume there exists a constant $c > 0$ such that for all $T$ and any sequence of losses, $R_T \leq cL\sqrt{T}$.

Consider $T > 16c^2$ and a multiple of $4$. We define the following linear losses $\ell_t(x) = x \cdot g_t$ where $g_t \in [-1, 1]$ is defined as

$$g_t = \begin{cases} (-1)^t \cdot L, & t \leq \frac{T}{2}, \\ -L, & t > \frac{T}{2}, \end{cases}$$

Recall that the FTRL update is $x_t = \arg\min_{x \in [-1,1]} \big\{ \eta_{t-1}\big(\sum_{s=1}^{t-1} g_s\big) \cdot x + \psi(x) \big\}$. Set $\eta = \eta_{T/2-1}$. With this sequence of losses, the points played by FTRL satisfy

- for $t \leq T/2 + 1$ and $t$ odd, we have $\sum_{s=1}^{t-1} g_s = 0$, so $x_t = 0$.

- for $t \leq T/2 + 1$ and $t$ even, we have $\sum_{s=1}^{t-1} g_s = -L$ so $x_t = \arg\min_{x \in [-1,1]} \{-\eta_{t-1} x + \psi(x)\}$. For $t < t' \leq T/2$ (both even), we have

$$
\begin{aligned}
-\eta_{t'-1} L x_{t'} + \psi(x_{t'}) &\leq -\eta_{t'-1} L x_t + \psi(x_t) && \text{using the definition of } x_{t'} \\
&= -\eta_{t-1} L x_t + \psi(x_t) + L(\eta_{t-1} - \eta_{t'-1}) x_t \\
&\leq -\eta_{t-1} L x_{t'} + \psi(x_{t'}) + L(\eta_{t-1} - \eta_{t'-1}) x_t && \text{using the definition of } x_t \\
\implies (\eta_{t-1} - \eta_{t'-1}) L x_{t'} &\leq (\eta_{t-1} - \eta_{t'-1}) L x_t \\
\implies x_{t'} &\leq x_t && \text{using that } \eta_{t'-1} \leq \eta_{t-1}.
\end{aligned}
$$

So for all $t \leq T/2$ even, we have $x_t \geq x_{T/2}$.

- for $t > T/2$, we have $\sum_{s=1}^{t-1} g_s = -L(t - T/2 - 1)$ so $x_t = \arg\min_{x \in [-1,1]} \{-\eta_{t-1} L(t - T/2 - 1) \cdot x + \psi(x)\}$.

The regret can then be written as follows

$$
R_T = \sum_{t=1}^{T} \ell_t(x_t) - \left(-\frac{LT}{2}\right) \geq \frac{LT}{2} + \frac{LT}{4} x_{T/2} - L \sum_{t=T/2+1}^{T} x_t, \tag{12}
$$

from which we can show the series of following statements.

1. We first show $\max_{2 \leq t \leq \lceil 2c\sqrt{T} \rceil} x_{\frac{T}{2}+t} \geq \frac{1}{2}$: if not, $x_{\frac{T}{2}+t} < \frac{1}{2}$ for all $t \leq \lceil 2c\sqrt{T} \rceil$ and from (12):

$$
\begin{aligned}
R_T &\geq \frac{LT}{2} - L\left( \sum_{t=T/2+1}^{T/2+\lfloor 2c\sqrt{T} \rfloor} x_t + \sum_{t=T/2+\lceil 2c\sqrt{T} \rceil+1}^{T} x_t \right) \\
&> \frac{LT}{2} - L\left( \sum_{t=T/2+1}^{T/2+\lfloor 2c\sqrt{T} \rfloor} \frac{1}{2} + \sum_{t=T/2+\lceil 2c\sqrt{T} \rceil+1}^{T} 1 \right) \\
&= \frac{LT}{2} - \frac{L}{2} \lfloor 2c\sqrt{T} \rfloor - L\left( T - \lceil 2c\sqrt{T} \rceil - \frac{T}{2} \right) \\
&\geq \frac{L}{2} \lceil 2c\sqrt{T} \rceil \\
&> cL\sqrt{T},
\end{aligned}
$$

which contradicts our initial assumption that $R_T \leq cL\sqrt{T}$ so we must have $\max_{2 \leq t \leq \lceil 2c\sqrt{T} \rceil} x_{\frac{T}{2}+t} \geq \frac{1}{2}$. Note that $2c\sqrt{T} < T/2$ is ensured by $T > 16c^2$.

2. Next, we show that $\eta \geq \frac{\psi(1/2)}{2cL\sqrt{T}}$: let $t^\star = \arg\max_{2 \leq t \leq \lceil 2c\sqrt{T} \rceil} x_{\frac{T}{2}+t}$, by the definition of $x_{\frac{T}{2}+t^\star}$:

$$
\begin{aligned}
0 &\geq -\eta_{\frac{T}{2}+t^\star-1} L\left( \frac{T}{2} + t^\star - 1 - \frac{T}{2} \right) x_{\frac{T}{2}+t^\star} + \psi(x_{\frac{T}{2}+t^\star}) \\
\implies \eta_{\frac{T}{2}+t^\star-1} L\left( t^\star - 1 \right) &\geq \psi(1/2) \\
\implies \eta = \eta_{T/2-1} \geq \eta_{\frac{T}{2}+t^\star-1} &\geq \frac{\psi(1/2)}{L(t^\star - 1)} \geq \frac{\psi(1/2)}{2cL\sqrt{T}}.
\end{aligned}
$$

where in the first implication, we used that $\psi(x_{\frac{T}{2}+t^\star}) \geq \psi(1/2)$ (since $\psi$ is increasing on $[0,1]$ and $x_{\frac{T}{2}+t^\star} \geq 1/2$) and $x_{\frac{T}{2}+t^\star} \leq 1$.

3. From (12), we also have $R_T \geq \frac{LT}{4} x_{T/2}$. To achieve $R_T \leq cL\sqrt{T}$, we must have $x_{T/2} \leq \frac{4c}{\sqrt{T}}$.

4. By the definition of $x_{T/2} = \arg\min_{x \in [-1,1]}\{-\eta Lx + \psi(x)\}$, for any $x \in [4c/\sqrt{T}, 1]$ we have

$$- \eta Lx_{T/2} + \psi(x_{T/2}) \le -\eta Lx + \psi(x)$$

$$\implies \psi(x) \ge \eta L(x - x_{T/2}) \ge \eta L\left(x - \frac{4c}{\sqrt{T}}\right) \ge \frac{\psi(1/2)}{2c\sqrt{T}}\left(x - \frac{4c}{\sqrt{T}}\right)$$

$$\implies \psi\left(\frac{5c}{\sqrt{T}}\right) \ge \frac{\psi(1/2)}{2c\sqrt{T}}\frac{c}{\sqrt{T}} = \frac{\psi(1/2)}{2T}.$$

5. Now fix $x \in [0, 1]$. There exists $T$ (multiple of 4) such that $x \in \left[\frac{5c}{\sqrt{T+4}}, \frac{5c}{\sqrt{T}}\right]$. Using that $\psi$ is increasing on $[0, 1]$ and from the previous point, we have

$$\psi(x) \ge \psi\left(\frac{5c}{\sqrt{T+4}}\right) \ge \frac{\psi(1/2)}{2(T+4)} \ge \frac{\psi(1/2)}{2(T+4)}\frac{T}{25c^2}x^2 \ge \frac{\psi(1/2)}{100c^2}x^2,$$

using that $T/(T+4) \ge 1/2$ for $T \ge 8$. The result is shown with $\mu = \psi(1/2)/100c^2$.

# F  Universal optimality of mirror descent for time-varying regret rates

In this section, we present an extension of the result of [39] on the universality of OMD. We first briefly review the considered setting along with their result. We refer the reader to Part II / Chapters 5 & 7 of [39] for the complete details. We present the results with respect to OMD but they also hold for FTRL up to slightly different constants.

We consider general OCO with linear losses (i.e. online linear optimisation (OLO)): The action set $\mathcal{H} \subset \mathcal{B}$ is a convex and centrally symmetric set that is a subset of an arbitrary real vector space $\mathcal{B}$. The subgradients of the linear losses belong to a set $\mathcal{X} \subset \mathcal{B}^\star$ that is a subset of the dual $\mathcal{B}^\star$ of $\mathcal{B}$. We focus on linear losses for simplicity but the results hold for general OCO where the subgradients of the losses are in $\mathcal{X}$ since the regret can be bounded by the linearised regret, see e.g. Corollary 64 in [39] or Section 2.3 in [36].

The regret is defined as

$$R_T(A, g_1, ..., g_T) = \sum_{t=1^T} \langle A(g_{1:t-1}), g_t \rangle - \inf_{h \in \mathcal{H}} \sum_{t=1}^{T} \langle h, g_t \rangle,$$

where $g_t$ are the subgradients defining the linear losses such that $g_t \in \mathcal{X}$ but otherwise are arbitrary / adversarial and $A$ is a learning algorithm. The minimax regret is

$$\mathcal{V}_T(\mathcal{H}, \mathcal{X}) = \inf_A \sup_{g_1, ..., g_T \in \mathcal{X}} R_T(A, g_1, ..., g_T).$$

We re-state the main the result on the universality of mirror descent from [39].

**Theorem F.1** (Theorem 71 of [39]). *If for some constant $V > 0$ and some $q \in [2, \infty)$, $\mathcal{V}_T(\mathcal{H}, \mathcal{X}) \leq VT^{1-1/q}$ for all $T$, then for any $T > e^{q-1}$, there exists a regularizer function $\Psi$ and step-size $\eta$, such that the regret of the mirror descent algorithm (OMD) $\mathcal{A}_{MD}$ using $\Psi$ against any $g_1, ..., g_T \in \mathcal{X}$ chosen by the adversary is bounded as*

$$R_T(\mathcal{A}_{MD}, g_1, ..., g_T) \leq 6002 \cdot V \cdot \log^2(T) \cdot T^{1-1/q}.$$

The result states that any regret bound with constant rate that is achievable across all time horizons can be matched by OMD up to logarithmic factors. The extension that we discuss next handles the case where there may be multiple regret bounds with different constant rates that exchange ordering at different time horizon intervals.

**Theorem F.2.** *Let $K > 0$ be an integer. If for $k = 1, ..., K$, there exists constants $V_k > 0$ (w.r.t. $T$) and $q_k \geq 2$ such that $\mathcal{V}_T(\mathcal{H}, \mathcal{X}) \leq \min_{k=1,...,K}\left\{V_k T^{1-1/q_k}\right\}$ for all $T$, then for any $T > e^{q-1}$, there exists a procedure $\mathcal{A}_{MD+}$ running OMD over intervals of doubling lengths such that the corresponding regret against any $g_1, ..., g_T \in \mathcal{X}$ chosen by the adversary is bounded as*

$$R_T(\mathcal{A}_{MD+}, g_1, ..., g_T) \leq (2 + \sqrt{2}) \cdot 6002 \cdot \log^2(T) \cdot \min_k\left\{V_k T^{1-1/q_k}\right\}.$$

*The procedure does not require knowledge of $T$.*

This matches up to a factor of $2 + \sqrt{2}$ the regret bound we would get by using Theorem F.1 / Theorem 71 of [39] with advanced knowledge of $T$, and otherwise matches the minimax regret up to constant and logarithmic factors.

This procedure and result could be used to obtain a result similar to Theorem 3.1. However, the bounds are worst due to the additional logarithmic factors and much larger constants. Therefore Theorem 3.1 remains a valuable contribution.

We now provide the proof and procedure based on the doubling-trick.

*Proof.* Fix $T$, $T_i = 2^i$, $B = \min\left\{j \in \mathbb{N} : \sum_{i=0}^{j} T_i \geq T\right\}$. We have:

$$2^B - 1 = \sum_{i=0}^{B-1} T_i < T \leq \sum_{i=0}^{B} T_i = \sum_{i=0}^{B} 2^i = 2^{B+1} - 1 \implies T \in [2^B, 2^{B+1} - 1].$$

Consider the following procedure $\mathcal{A}_{MD+}$. For $i = 0, 1, ..., B$:

- set $k(i) = \arg\min_k \left\{ V_k T_i^{1-1/q_k} \right\}$.

- Use Theorem F.1 / Theorem 71 of [39] to get a regulariser for OMD that achieves the regret upper-bound of $6002 V_{k(i)} \log^2(T_i) \cdot T_i^{1-1/q_{k(i)}}$ over $T_i$ rounds. Use this on the $T_i$ rounds $\left\{ \sum_{j=0}^{i-1} T_j + 1, ..., \sum_{j=0}^{i-1} T_j + T_i \right\}$. When $i = B$, just run it up to round $T$.

Let $k_\star = \arg\min_k \left\{ V_k T^{1-1/q_k} \right\}$. The regret is bounded as follows:

$$R_T(\mathcal{A}_{MD+}, g_1, ..., g_T) \leq \sum_{i=0}^{B} 6002 V_{k(i)} \log^2(T_i) \cdot T_i^{1-1/q_{k(i)}}$$

$$\leq 6002 \log^2(T) \sum_{i=0}^{B} V_{k(i)} \cdot T_i^{1-1/q_{k(i)}} \qquad \text{since for all i, } T_i \leq T$$

$$\leq 6002 \log^2(T) \sum_{i=0}^{B} V_{k_\star} \cdot T_i^{1-1/q_{k_\star}} \qquad \text{by definition of } k(i)$$

$$= 6002 \cdot V_{k_\star} \cdot \log^2(T) \cdot \sum_{i=0}^{B} (2^{1-1/q_{k_\star}})^i$$

$$= 6002 \cdot V_{k_\star} \cdot \log^2(T) \cdot \frac{(2^{1-1/q_{k_\star}})^{B+1} - 1}{2^{1-1/q_{k_\star}} - 1}$$

$$\leq \frac{2^{1-1/q_{k_\star}}}{2^{1-1/q_{k_\star}} - 1} \cdot 6002 \cdot V_{k_\star} \cdot \log^2(T) \cdot (2^B)^{1-1/q_{k_\star}}$$

$$\leq \frac{2^{1-1/q_{k_\star}}}{2^{1-1/q_{k_\star}} - 1} \cdot 6002 \cdot V_{k_\star} \cdot \log^2(T) \cdot T^{1-1/q_{k_\star}} \qquad \text{since } 2^B \leq T$$

$$\leq \frac{\sqrt{2}}{\sqrt{2} - 1} \cdot 6002 \cdot V_{k_\star} \cdot \log^2(T) \cdot T^{1-1/q_{k_\star}} \qquad \text{since } q_{k_\star} \geq 2$$

$$= (2 + \sqrt{2}) \cdot 6002 \cdot \log^2(T) \cdot \min_k \left\{ V_k T^{1-1/q_k} \right\}.$$

$\square$

# G    Proofs of Section 5

Throughout this section, we will use the notation $R_T$ for the pseudo-regret. In fact since a randomized learner is equivalent to a random choice of deterministic learners, we will consider in the proofs below deterministic learners and the regret is equal to the pseudo-regret. In addition, since our loss constructions are oblivious to the learner's actions, even for randomised learners, the pseudo-regret is equal to the regret.

We split the proof into the case where $p > 4/3$ is "large" and the case where $p \in [1, 4/3]$ is "small" and consider separate loss constructions for each. We first highlight the intuition of the loss constructions.

- **For $p > 4/3$**, we take inspiration from the loss construction which [6] use to prove a $\Omega(d\sqrt{T})$ lower bound for low-dimensional ($d^2 < T$) $\ell_p$-balls with $p > 2$. The construction consists of linear Gaussian losses where the mean of each coordinate is the same distance from 0 but the learner does not know the sign. When the dimension is large enough, the learner does not acquire enough information to determine the signs of these means in the $T$ rounds to get sub-linear regret. **This construction will not work when $p \leq 4/3$** because when $p$ is close to 1, the lack of a distinct corner in the $\ell_p$-ball allows any point on the boundary (including $\pm e_1$) with correct signs to achieve similar loss to the competitor (a corner). The learner can therefore focus on $\pm e_1$ simplifying the problem to one-dimension where sub-linear regret is achievable.

- **For $p \leq 4/3$**, the construction consists of linear Gaussian losses where the mean vector has a single non-zero positive entry, unknown to the learner. When the dimension is large enough, it does not acquire enough information to determine the non-zero coordinate of the mean in the $T$ rounds to get sub-linear regret. **This construction will not work when $p > 4/3$** because for $p \gg 1$ the learner can exploit the $\ell_p$-ball's proximity to the hypercube by playing points with all coordinates close to $-1$, bypassing the need to identify the correct non-zero mean coordinate.

We present the proofs with a Lipschitz constant of 1 but they extend straightforwardly to arbitrary $L > 0$.

## G.1    Case $p > 4/3$

**Theorem G.1.** *Fix $T$ and $\delta > 0$. Consider $p > 4/3$ and*

$$d > \max\left\{16T, \frac{1}{c_1}\log\frac{C_1 T}{\delta}, \left(\frac{1}{c_1 p_\star}\log\frac{C_1 T}{\delta}\right)^{p_\star/2}, e^2\right\},$$

*for some universal constants $c_1, C_1$. For any OCO algorithm with bandit feedback on $V = \mathcal{B}_p$, there exists a sequence of random linear losses $(\ell_t)_{t\in[T]}$ with sub-gradients $(g_t)_{t\in[T]}$ such that $\|g_t\|_{p_\star} \leq 1$ for all rounds $t$ with probability at least $1 - \delta$ and*

$$\mathbb{E}[R_T] \geq \frac{T}{80},$$

*where the expectation is with respect to the randomness of the losses.*

### G.1.1    Proof

The following loss construction and analysis is inspired from the proof of Theorem 4 of [6]. Their construction is designed for the low-dimensional setting in such a way that the learner has to balance exploration and exploitation rounds. We only consider the losses corresponding to exploration rounds and generalize the analysis to the high-dimensional setting.

Let $\varepsilon > 0$ be such that $\varepsilon^{p_\star} = 1/d$. Let $T < \alpha d$ (with $\alpha = 1/16$). For a fixed $\xi \in \{-1, 1\}^d$, define the losses as $\ell_t(x) = x^T g_t^\xi$ where $g_t^\xi \sim \mathcal{N}(\varepsilon\xi, \frac{1}{d^{2/p_\star}}I_d)$ (i.i.d.). We show that (when $\xi$ is sampled uniformly at the start and fixed throughout the rounds)

$$\mathbb{E}_\xi \mathbb{E}_{g_t^\xi}[R_T] \geq \frac{T}{16}.$$

We use $\mathbb{E}_\xi$ for the expectation with respect to $\xi$ and $\mathbb{E}_{g_t^\xi}$ for the expectation with respect to $g_t^\xi$ with $\xi$ fixed. We will also use $x_{t,i}$ to mean the $i$-th coordinate of $x_t$.

For fixed $\xi$: $\mathbb{E}_{g_t^\xi}[\ell_t(x)] = \mathbb{E}[x^T g_t^\xi] = \varepsilon \cdot x^T \xi$. So the competitor $x^\star = \arg\min_{x \in V} \varepsilon \cdot \xi^T x = -d^{-1/p}\xi$. Let us define $\bar{x} = \frac{1}{T}\sum_{t=1}^T \mathbb{E}[x_t]$. In particular one has

$$\mathbb{E}_{g_t^\xi}[R_T] = \varepsilon T \cdot \xi^T(\bar{x} - x^\star).$$

The following lemma expresses the expected regret in terms of the expected number of rounds and coordinates for which the learner plays on the wrong side of $\xi$. The proof is in Appendix G.1.3.

**Lemma G.2** (Generalization of Lemma 6 of [6]).

$$\mathbb{E}_{g_t^\xi}[R_T] \geq \frac{\varepsilon^{p_\star}}{p_\star} \cdot \mathbb{E}_{g_t^\xi}\Big[\sum_{t=1}^T \sum_{i=1}^d \mathbb{I}\{x_{t,i}\xi_i \geq 0\}\Big].$$

And now the next lemma shows that the expected number of rounds and coordinates for which the learner plays on the wrong side of $\xi$ is linear in both $T$ and $d$. The proof is in Appendix G.1.4.

**Lemma G.3.** *With $T < \alpha d = \frac{1}{16}$, we have $\mathbb{E}_\xi \mathbb{E}_{g_t^\xi}\sum_{t=1}^T \sum_{i=1}^d \mathbb{I}\{x_{t,i}\xi_i \geq 0\} \geq \frac{dT}{4}$.*

Combining both lemmas, we have

$$\mathbb{E}[R_T] \geq \frac{1}{p_\star}\varepsilon^{p_\star} \cdot \mathbb{E}\Big[\sum_{t=1}^T \sum_{i=1}^d \mathbb{I}\{x_{t,i}\xi_i \geq 0\}\Big] \geq \frac{1}{4p_\star}\varepsilon^{p_\star}Td = \frac{1}{4p_\star}T \geq \frac{T}{16},$$

since $p > 4/3$ so $p_\star \leq 4$. Now to ensure the Lipschitz-condition with high-probability, we get an extra factor of $1/5$ (see the next section), concluding the proof.

### G.1.2  Bound on Sub-gradients

Recall that $p > 4/3$ so $p_\star \leq 4$. Fix $\xi \in \{-1,1\}^d$. $g_t \sim \mathcal{N}(\varepsilon\xi, d^{-2/p_\star}I_d)$. So $g_t = d^{-1/p_\star}X + \varepsilon\xi$ where $X \sim \mathcal{N}(0, I_d)$. From [44], we have

$$\mathbb{E}[\|X\|_{p_\star}] \leq \Big(\mathbb{E}\Big[\sum_{i=1}^d |X_i|^{p_\star}\Big]\Big)^{1/p_\star} = \Big(\mathbb{E}\Big[\sum_{i=1}^d 2^{p_\star/2}\frac{\Gamma((p_\star+1)/2)}{\sqrt{\pi}}\Big]\Big)^{1/p_\star} \leq \sqrt{2}\Big(\frac{3}{4}\Big)^{1/p_\star}d^{1/p_\star} \leq 2d^{1/p_\star}.$$

$$\implies \mathbb{E}[\|g_t\|_{p_\star}] \leq 2 + \varepsilon d^{1/p_\star} = 2 + 1 = 3.$$

Fix $\delta > 0$.

- **For $p_\star \leq 2$:** By Theorem 1.1 in [38] for some constants $C_1, c_1 > 0$,

$$\mathbb{P}\Big(\|X\|_{p_\star} \leq (1+\beta)\mathbb{E}[\|X\|_{p_\star}]\Big) \geq 1 - C_1\exp(-c_1\beta^2 d).$$

  Assuming $d \geq \frac{1}{c_1}\log\frac{C_1 T}{\delta}$, we have $\beta = \sqrt{\frac{1}{c_1 d}\log\frac{C_1 T}{\delta}} \leq 1$ and

$$\mathbb{P}\Big(\|X\|_{p_\star} \leq (1+\beta)\mathbb{E}[\|X\|_{p_\star}]\Big) \geq 1 - \frac{\delta}{T}.$$

- **For $2 < p_\star \leq 4$:** By Theorem 1.1 in [38] for some constants $C_1, c_1 > 0$,

$$\mathbb{P}\Big(\|X\|_{p_\star} \leq (1+\beta)\mathbb{E}[\|X\|_{p_\star}]\Big) \geq 1 - C_1\exp(-c_1\beta p_\star d^{2/p_\star}).$$

  Assuming $d \geq \Big(\frac{1}{c_1 p_\star}\log\frac{C_1 T}{\delta}\Big)^{p_\star/2}$, we have $\beta = \frac{1}{c_1 q d^{2/p_\star}}\log\frac{C_1 T}{\delta} \leq 1$ and

$$\mathbb{P}\Big(\|X\|_{p_\star} \leq (1+\beta)\mathbb{E}[\|X\|_{p_\star}]\Big) \geq 1 - \frac{\delta}{T}.$$

In both cases, with probability at least $1 - \delta/T$

$$\|X\|_{p_\star} \leq (1+\beta)\mathbb{E}[\|X\|_{p_\star}] \leq 2\mathbb{E}[\|X\|_{p_\star}] \leq 4d^{1/p_\star},$$

$$\implies \|g_t\|_{p_\star} \leq d^{-1/p_\star}\|X\|_{p_\star} + \varepsilon d^{1/p_\star} \leq 4 + 1 = 5.$$

By a union bound over all rounds, with probability $1 - \delta$, $\|g_t\|_{p_\star} \leq 5$ for all rounds $t$. So rescaling the losses by a factor of $5$ gives sub-gradients whose $\ell_{p_\star}$-norm is bounded by $1$ with high-probability and a regret bound of:

$$\mathbb{E}[R_T] \geq \frac{T}{80}.$$

### G.1.3 Proof of Lemma G.2

Let $W_t = \left\{ i \in [d] : x_{t,i}\xi_i < 0 \right\}$ and $S = \mathbb{E}\left[ \sum_{t=1}^{T} \sum_{i=1}^{d} \mathbb{I}\{x_{t,i}\xi_i \geq 0\} \right]$. We have that

$$
\mathbb{E}_{g_t^\xi}\left[ R_T \right] = \varepsilon T \cdot \xi^T (\bar{x} - x^\star)
$$

$$
= \varepsilon \sum_{t=1}^{T} \mathbb{E}_{g_t^\xi}\left[ \sum_{i \notin W_t} \xi_i x_{t,i} \right] + \varepsilon \sum_{t=1}^{T} \mathbb{E}_{g_t^\xi}\left[ \sum_{i \in W_t} \xi_i x_{t,i} \right] + \varepsilon T d^{1/p_\star}
$$

$$
\geq \varepsilon \sum_{t=1}^{T} \mathbb{E}_{g_t^\xi}\left[ \sum_{i \in W_t} \xi_i x_{t,i} \right] + \varepsilon T d^{1/p_\star}
$$

Therefore, it is sufficient to show that

$$
\varepsilon \sum_{t=1}^{T} \mathbb{E}_{g_t^\xi}\left[ \sum_{i \in W_t} \xi_i x_{t,i} \right] + \varepsilon T d^{1/p_\star} \geq \frac{\varepsilon^{p_\star} S}{p_\star}.
$$

Since $\|x_{t,W_t}\|_p \leq 1$ (we use $x_{t,W_t}$ to mean that the coordinates of $x_t$ that are not in $W_t$ are 0), by Holder's inequality we know that

$$
\varepsilon \sum_{i \in W_t} \xi_i x_{t,i} = (x_{t,W_t})^T (\varepsilon \xi_{W_t}) \geq -\|x_{t,W_t}\|_p \|-\varepsilon \xi_{W_t}\|_{p_\star} \geq -|W_t|^{1/p_\star}\varepsilon.
$$

Noting that (see (14) below)

$$
|W_t|^{1/p_\star}\varepsilon = ((d - |W_t^C|)\varepsilon^{p_\star})^{1/p_\star} \leq (d\varepsilon^{p_\star})^{1/p_\star} - \frac{1}{p_\star}\varepsilon^{p_\star}|W_t^C|, \tag{13}
$$

we have

$$
\varepsilon \sum_{i \in W_t} \xi_i x_{t,i} \geq \frac{1}{p_\star}\varepsilon^{p_\star}|W_t^C| - (d\varepsilon^{p_\star})^{1/p_\star} = \frac{1}{p_\star}\varepsilon^{p_\star} \sum_{i=1}^{d} \mathbb{I}\{x_{t,i}\xi_i \geq 0\} - (d\varepsilon^{p_\star})^{1/p_\star}
$$

$$
\implies \varepsilon \sum_{t=1}^{T} \mathbb{E}_{g_t^\xi}\left[ \sum_{i \in W_t} \xi_i x_{t,i} \right] \geq \frac{\varepsilon^{p_\star}}{p_\star}\mathbb{E}_{g_t^\xi}\left[ \sum_{t=1}^{T} \sum_{i=1}^{d} \mathbb{I}\{x_{t,i}\xi_i \geq 0\} \right] - T(d\varepsilon^{p_\star})^{1/p_\star} = \frac{\varepsilon^{p_\star} S}{p_\star} - \varepsilon T d^{1/p_\star},
$$

which concludes the proof.

**Proof of (13):** Since $x^{1/p_\star}$ is concave: for all $x, y \in \mathbb{R}$, $x^{1/p_\star} \leq y^{1/p_\star} + \frac{1}{p_\star}y^{-1/p}(x - y)$. In particular, with $x = \varepsilon^{p_\star}(d - s)$, $y = \varepsilon^{p_\star} d$, we have

$$
\varepsilon(d - s)^{1/p_\star} \leq \varepsilon d^{1/p_\star} - \frac{1}{p_\star}\frac{\varepsilon^{p_\star} s}{(\varepsilon^{p_\star} d)^{1/p}} = \varepsilon d^{1/p_\star} - \frac{1}{p_\star}\varepsilon^{p_\star} s, \tag{14}
$$

since $\varepsilon^{p_\star} d = 1$. Using $s = |W_t^C|$ gives the result.

### G.1.4 Proof of Lemma G.3

At round $t$ conditioned on $\xi$, the observed feedback is

$$
f_t^\xi := x_t^T g_t^\xi \sim \mathcal{N}(\varepsilon \cdot x_t^T \xi, \sigma_t^2), \text{ where } \sigma_t^2 = \frac{\|x_t\|_2^2}{d^{2/p_\star}}.
$$

Denote $p_\xi$ for the law of the observed feedback up to time $T$ conditioned on $\xi$, i.e., the law of $(f_1^\xi, ..., f_T^\xi)$. Consider $\xi$ and $\xi'$ differing only in coordinate $i \in d$. By Pinsker's inequality we have

$$
d_{\mathrm{TV}}(p_\xi, p_{\xi'}) \leq \sqrt{\frac{1}{2} D_{\mathrm{KL}}(p_\xi, p_{\xi'})}.
$$

By the chain rule for the KL divergence / operations on conditional densities:

$$D_{\mathrm{KL}}(p_\xi, p_{\xi'}) = \mathbb{E}_{(f_1,...,f_T)\sim p_\xi}\left[\log \frac{p_\xi(f_1,...,f_T)}{p_{\xi'}(f_1,...,f_T)}\right]$$

$$= \sum_{t=1}^T \mathbb{E}_{(f_1,...,f_t)\sim p_\xi}\left[\log \frac{p_\xi(f_t|f_{t-1}...,f_1)}{p_{\xi'}(f_t|f_{t-1}...,f_1)}\right]$$

$$= \sum_{t=1}^T \mathbb{E}_{(f_1,...,f_{t-1})\sim p_\xi}\left\{\mathbb{E}_{f_t\sim p_\xi(\cdot|f_1,...,f_{t-1})}\left[\log \frac{p_\xi(f_t|f_{t-1}...,f_1)}{p_{\xi'}(f_t|f_{t-1}...,f_1)}\right]\right\}.$$

Now since $x_t$ is a deterministic function of $f_1, ..., f_{t-1}$ and given $x_t$, $f_t \sim \mathcal{N}(\varepsilon x_t^T \xi, \sigma_t^2)$ under $p_\xi$, we have that the inner expectation is a KL divergence between Gaussians:

$$\mathbb{E}_{f_t\sim p_\xi(\cdot|f_1,...,f_{t-1})}\left[\log \frac{p_\xi(f_t|f_{t-1}...,f_1)}{p_{\xi'}(f_t|f_{t-1}...,f_1)}\right] = \frac{\left(\varepsilon x_t^T \xi - \varepsilon x_t^T \xi'\right)^2}{2\sigma_t^2} = \frac{4\varepsilon^2 x_{t,i}^2}{2\sigma_t^2} = \frac{2\varepsilon^2 x_{t,i}^2}{\sigma_t^2}$$

$$\implies D_{\mathrm{KL}}(p_\xi, p_{\xi'}) = 2\sum_{t=1}^T \mathbb{E}_{(f_1,...,f_{t-1})\sim p_\xi}\left[\frac{\varepsilon^2 x_{t,i}^2}{\sigma_t^2}\right] = 2\sum_{t=1}^T \mathbb{E}_{p_\xi}\left[\frac{\varepsilon^2 x_{t,i}^2}{\sigma_t^2}\right]$$

$$\implies d_{\mathrm{TV}}(p_\xi, p_{\xi'}) \leq \sqrt{\sum_{t=1}^T \mathbb{E}_{p_\xi}\left[\frac{\varepsilon^2 x_{t,i}^2}{\sigma_t^2}\right]}.$$

Now, using $\xi_{-i}$ to refer to all the coordinates of $\xi$ except the $i$-th and $\xi_{i+}$ (resp. $\xi_{i,-}$) to denote that the $i$-th coordinate of $\xi$ is $+1$ (resp. $-1$),

$$\mathbb{E}_\xi\left[\mathbb{E}_{p_\xi}\left[\sum_{t=1}^T \mathbb{I}\{x_{t,i}\xi_i \geq 0\}\right]\right] = \mathbb{E}_{\xi_{-i}}\mathbb{E}_{\xi_i}\left[\mathbb{E}_{p_\xi}\left[\sum_{t=1}^T \mathbb{I}\{x_{t,i}\xi_i \geq 0\}\right]|\xi_{-i}\right]$$

$$= \frac{1}{2}\mathbb{E}_{\xi_{-i}}\left[\mathbb{E}_{p_{\xi_{i+}}}\left[\sum_{t=1}^T \mathbb{I}\{x_{t,i}\cdot 1 \geq 0\}\right] + \mathbb{E}_{p_{\xi_{i-}}}\left[\sum_{t=1}^T \mathbb{I}\{x_{t,i}\cdot(-1) \geq 0\}\right]\right]$$

$$= \frac{1}{2}\mathbb{E}_{\xi_{-i}}\left[\mathbb{E}_{p_{\xi_{i+}}}\left[\sum_{t=1}^T \mathbb{I}\{x_{t,i} \geq 0\}\right] + \mathbb{E}_{p_{\xi_{i-}}}\left[\sum_{t=1}^T 1 - \mathbb{I}\{x_{t,i} > 0\}\right]\right]$$

$$\geq \frac{T}{2} + \frac{1}{2}\sum_{t=1}^T \mathbb{E}_{\xi_{-i}}\left[\mathbb{E}_{p_{\xi_{i+}}}\left[\mathbb{I}\{x_{t,i} \geq 0\}\right] - \mathbb{E}_{p_{\xi_{i-}}}\left[\mathbb{I}\{x_{t,i} \geq 0\}\right]\right]$$

$$\geq \frac{T}{2} - \frac{1}{4}\sum_{t=1}^T \mathbb{E}_{\xi_{-i}}\left[d_{\mathrm{TV}}(p_{\xi_{i+}}, p_{\xi_{i-}}) + d_{\mathrm{TV}}(p_{\xi_{i-}}, p_{\xi_{i+}})\right] \quad \text{using Pinsker's inequality}$$

$$= \frac{T}{2} - \frac{T}{4}\mathbb{E}_{\xi_{-i}}\left[d_{\mathrm{TV}}(p_{\xi_{i+}}, p_{\xi_{i-}}) + d_{\mathrm{TV}}(p_{\xi_{i-}}, p_{\xi_{i+}})\right]$$

$$\geq \frac{T}{2} - \frac{T}{4}\mathbb{E}_{\xi_{-i}}\left[\sqrt{\sum_{t=1}^T \mathbb{E}_{g_t^{\xi_{i+}}}\frac{\varepsilon^2 x_{t,i}^2}{\sigma_t^2}} + \sqrt{\sum_{t=1}^T \mathbb{E}_{g_t^{\xi_{i-}}}\frac{\varepsilon^2 x_{t,i}^2}{\sigma_t^2}}\right]$$

$$= \frac{T}{2} - \frac{T}{2}\mathbb{E}_\xi\left[\sqrt{\sum_{t=1}^T \mathbb{E}_{g_t^\xi}\frac{\varepsilon^2 x_{t,i}^2}{\sigma_t^2}}\right].$$

Summing over all possible coordinates $i$, we get:

$$\frac{1}{T}\sum_{i=1}^d \mathbb{E}_\xi\left[\mathbb{E}_{p_\xi}\left[\sum_{t=1}^T \mathbb{I}\{x_{t,i}\xi_i \geq 0\}\right]\right] \geq \frac{d}{2} - \frac{1}{2}\sum_{i=1}^d \mathbb{E}_\xi\left[\sqrt{\sum_{t=1}^T \mathbb{E}_{g_t^\xi}\frac{\varepsilon^2 x_{t,i}^2}{\sigma_t^2}}\right].$$

Note that due to the concavity of the square-root:

$$\sum_{i=1}^{d} \mathbb{E}_{\xi}\left[\sqrt{\sum_{t=1}^{T} \mathbb{E}_{g_t^\xi} \frac{\varepsilon^2 x_{t,i}^2}{\sigma_t^2}}\right] \leq \sum_{i=1}^{d} \sqrt{\sum_{t=1}^{T} \mathbb{E}_{\xi,g_t^\xi} \frac{\varepsilon^2 x_{t,i}^2}{\sigma_t^2}}$$

$$= d\frac{1}{d}\sum_{i=1}^{d} \sqrt{\sum_{t=1}^{T} \mathbb{E}_{\xi,g_t^\xi} \frac{\varepsilon^2 x_{t,i}^2}{\sigma_t^2}}$$

$$\leq d\sqrt{\frac{1}{d}\sum_{i=1}^{d}\sum_{t=1}^{T} \mathbb{E}_{\xi,g_t^\xi} \frac{\varepsilon^2 x_{t,i}^2}{\sigma_t^2}}$$

$$= \sqrt{d\sum_{t=1}^{T} \mathbb{E}_{\xi,g_t^\xi} \frac{\varepsilon^2 \|x_t\|_2^2}{\sigma_t^2}}.$$

So we get

$$\frac{1}{T}\mathbb{E}_\xi \mathbb{E}_{g_t^\xi} \sum_{t=1}^{T}\sum_{i=1}^{d} \mathbb{I}\{x_{t,i}\xi_i \geq 0\} \geq \frac{d}{2} - \sqrt{d\sum_{t=1}^{T} \mathbb{E}_{\xi,g_t^\xi} \frac{\varepsilon^2 \|x_t\|_2^2}{\sigma_t^2}}$$

$$= \frac{d}{2} - \sqrt{d^{1+2/p_\star} T \varepsilon^2}$$

$$= \frac{d}{2} - \sqrt{dT}$$

$$\geq d\left(\frac{1}{2} - \sqrt{\alpha}\right)$$

$$= \frac{d}{4}$$

since $\varepsilon = (1/d)^{1/p_\star}$ and $T < \alpha d = d/16$.

## G.2 Case $1 < p \leq 4/3$

**Theorem G.4.** *Fix $T$ and $\delta > 0$. Consider $p \in (1, 4/3]$ and*

$$d > \max\left\{(128 p_\star T)^2, \left(\frac{1}{c_1 p_\star}\log\frac{C_1 T}{\delta}\right)^{p_\star/2}, e^2\right\},$$

*for some universal constants $c_1, C_1$. For any OCO algorithm with bandit feedback on $V = \mathcal{B}_p$, there exists a sequence of random linear losses $(\ell_t)_{t\in[T]}$ with sub-gradients $(g_t)_{t\in[T]}$ such that $\|g_t\|_{p_\star} \leq 1$ for all rounds $t$ with probability at least $1 - \delta$ and*

$$\mathbb{E}[R_T] \geq \frac{T}{16},$$

*where the expectation is with respect to the randomness of the losses.*

### G.2.1 Proof

Before the start of the game, draw $Y \sim Unif(1, ..., d)$ and define the losses as $\ell_t(x) = x^T g_t^Y$ where

$$g_{t,i}^Y \sim \begin{cases} \mathcal{N}(0, \sigma^2), & \text{if } i \neq Y, \\ \mathcal{N}(1/2, \sigma^2), & \text{if } i = Y, \end{cases}$$

where $\sigma = (8\sqrt{p_\star}d^{1/p_\star})^{-1}$. We show that $\mathbb{E}_Y \mathbb{E}_{\ell_1,...,\ell_T} R_T \geq T/16$.

Fix $\alpha \in [0, 1]$ and define $A_i(\alpha) = \{t : y_{t,i} \geq -\alpha\}$. Then

$$
\begin{aligned}
\mathbb{E}\big[R_T | Y = i\big] &= \frac{T}{2} + \frac{1}{2}\mathbb{E}\Big[\sum_{t=1}^{T} y_{t,i} | Y = i\Big] \\
&= \frac{T}{2} + \frac{1}{2}\mathbb{E}\Big[\sum_{t \notin A_i(\alpha)} y_{t,i} + \sum_{t \in A_i(\alpha)} y_{t,i} | Y = i\Big] \\
&\geq \frac{T}{2} + \frac{1}{2}\mathbb{E}\Big[\sum_{t \notin A_i(\alpha)} (-1) + \sum_{t \in A_i(\alpha)} (-\alpha) | Y = i\Big] \\
&= \frac{T}{2} - \frac{1}{2}\mathbb{E}\Big[T - |A_i(\alpha)| + \alpha|A_i(\alpha)| | Y = i\Big] \\
&= (1 - \alpha)\frac{1}{2}\mathbb{E}\Big[|A_i(\alpha)| | Y = i\Big] \\
&= (1 - \alpha)\frac{1}{2}\mathbb{E}\Big[\sum_{t=1}^{T} \mathbb{I}\{y_{t,i} \geq -\alpha\} | Y = i\Big].
\end{aligned}
$$

The following lemma bounds the expected number of rounds where the learner suffers large regret (as measured by $\alpha$) when $Y = i$ compared to an environment where all coordinates of $g_t$ are 0-mean for all $t$ (i.e. there is no better direction). We denote $E_{p_0}$ expectations with respect to this environment. The proof is in Appendix G.2.3.

**Lemma G.5.** *Let* $\sigma_t^2 = \|y_t\|_2^2 \cdot \sigma^2$, *then*

$$
\mathbb{E}\Big[\sum_{t=1}^{T} \mathbb{I}\{y_{t,i} \geq -\alpha\} | Y = i\Big] \geq \mathbb{E}_{p_0}\Big[\sum_{t=1}^{T} \mathbb{I}\{y_{t,i} \geq -\alpha\}\Big] - T\sqrt{\sum_{t=1}^{T} \mathbb{E}_{p_0}\Big[\frac{y_{t,i}^2}{8\sigma_t^2}\Big]}
$$

From the lemma, we have

$$
\mathbb{E}\big[R_T | Y = i\big] \geq (1 - \alpha)\frac{1}{2}\Big\{\mathbb{E}_{p_0}\Big[\sum_{t=1}^{T} \mathbb{I}\{y_{t,i} \geq -\alpha\}\Big] - T\sqrt{\sum_{t=1}^{T} \mathbb{E}_{p_0}\Big[\frac{y_{t,i}^2}{8\sigma_t^2}\Big]}\Big\}.
$$

Taking an expectation with respect to $Y$ we have:

$$
\begin{aligned}
\mathbb{E}\big[R_T\big] &= \frac{1}{d}\sum_{i=1}^{d}\mathbb{E}\big[R_T | Y = i\big] \\
&\geq \frac{(1 - \alpha)}{2d}\sum_{i=1}^{d}\Big\{\mathbb{E}_{p_0}\Big[\sum_{t=1}^{T} \mathbb{I}\{y_{t,i} \geq -\alpha\}\Big] - T\sqrt{\sum_{t=1}^{T} \mathbb{E}_{p_0}\Big[\frac{y_{t,i}^2}{8\sigma_t^2}\Big]}\Big\} \\
&= \frac{(1 - \alpha)}{2d}\Big\{\mathbb{E}_{p_0}\Big[\sum_{t=1}^{T}\sum_{i=1}^{d} \mathbb{I}\{y_{t,i} \geq -\alpha\}\Big] - T\sum_{i=1}^{d}\sqrt{\sum_{t=1}^{T} \mathbb{E}_{p_0}\Big[\frac{y_{t,i}^2}{8\sigma_t^2}\Big]}\Big\}.
\end{aligned}
$$

For the first term: fix $\alpha = (d/2)^{-1/p}$ and note that if $\sum_{i=1}^{d} \mathbb{I}\{y_{t,i} \geq -\alpha\} < d/2$, then there are more than $d/2$ coordinates of $y_t$ whose value is less than $-\alpha$, this means

$$
\|y_t\|_p^p > \frac{d}{2}\alpha^p = \frac{d}{2}\frac{2}{d} = 1,
$$

which contradicts $y_t \in \mathcal{B}_p$ so we must have $\sum_{i=1}^{d} \mathbb{I}\{y_{t,i} \geq -\alpha\} \geq d/2$.

For the second term, using Jensen's inequality and the concavity of $\sqrt{x}$,

$$
\frac{1}{d}\sum_{i=1}^{d}\sqrt{\sum_{t=1}^{T} \mathbb{E}_{p_0}\Big[\frac{y_{t,i}^2}{8\sigma_t^2}\Big]} \leq \sqrt{\frac{1}{d}\sum_{i=1}^{d}\sum_{t=1}^{T} \mathbb{E}_{p_0}\Big[\frac{y_{t,i}^2}{8\sigma_t^2}\Big]} = \sqrt{\frac{1}{d}\sum_{t=1}^{T} \mathbb{E}_{p_0}\Big[\frac{\|y_t\|_2^2}{8\sigma_t^2}\Big]} = \frac{1}{2\sigma}\sqrt{\frac{T}{2d}}.
$$

Combining we have

$$\mathbb{E}[R_T] \geq \frac{1-\alpha}{2}\left(\frac{T}{2} - T\frac{1}{2\sigma}\sqrt{\frac{T}{2d}}\right).$$

- Since $d \geq e^2 \geq 2^{p+1}$, $1 - \alpha = 1 - (2/d)^{1/p} \geq 1 - 2^{-p/p} = 1/2$.
- If $d \geq (128p_\star T)^2$ then $d \geq 2T/\sigma^2$ and:

$$\frac{T}{2} - T\frac{1}{2\sigma}\sqrt{\frac{T}{2d}} \geq \frac{T}{2} - \frac{T}{4} = \frac{T}{2}.$$

The condition on $d$ follows from the definition of $\sigma$ and $p_\star \geq 4$:

$$d \geq (128p_\star T)^2 \geq (128qT)^{1/(1-2/p_\star)} \implies d^{1-2/p_\star} \geq 128qT \implies d \geq \frac{128p_\star T}{d^{2/p_\star}} = \frac{2T}{\sigma^2}.$$

We hence have $\mathbb{E}[R_T] \geq T/16$. The following section ensures that the Lipschitz-condition is satisfied with high-probability.

### G.2.2 Bound on sub-gradients

Recall that $p \leq 4/3$, so $p_\star \geq 4$. Given $Y = i$, $g_t^Y \sim \mathcal{N}(\frac{1}{2}e_i, \sigma^2 I_d)$. So $g_t = \sigma X + \frac{1}{2}e_i$ where $X \sim \mathcal{N}(0, I_d)$. From [44], we have

$$\mathbb{E}[\|X\|_{p_\star}] \leq \left(\mathbb{E}\left[\sum_{i=1}^{d}|X_i|^{p_\star}\right]\right)^{1/p_\star} = \left(\mathbb{E}\left[\sum_{i=1}^{d} 2^{p_\star/2}\frac{\Gamma((p_\star+1)/2)}{\sqrt{\pi}}\right]\right)^{1/p_\star}.$$

From [3] (Theorem 2.2), we have

$$\Gamma\left(\frac{p_\star+1}{2}\right) = \Gamma\left(\frac{p_\star-1}{2}+1\right) \leq \sqrt{2\pi}\left(\frac{p_\star-1}{2}\right)^{(p_\star-1)/2}\exp\left(-\frac{p_\star-1}{2}\right)\sqrt{2\frac{p_\star-1}{2}} \leq 2\sqrt{\pi}p_\star^{p_\star/2}e^{-(p_\star-1)/2}$$

$$\implies \left(2^{p_\star/2}\frac{\Gamma((p_\star+1)/2)}{\sqrt{\pi}}\right)^{1/p_\star} \leq 2\sqrt{p_\star}$$

$$\implies \mathbb{E}[\|X\|_{p_\star}] \leq 2\sqrt{p_\star}d^{1/p_\star}$$

$$\implies \mathbb{E}[\|g_t\|_{p_\star}] \leq 2\sqrt{p_\star}\sigma d^{1/p_\star} + \frac{1}{2}.$$

Fix $\delta > 0$. By Theorem 1.1 in [38] for some constants $C_1, c_1 > 0$,

$$\mathbb{P}\left(\|X\|_{p_\star} \leq (1+\beta)\mathbb{E}[\|X\|_{p_\star}]\right) \geq 1 - C_1\exp(-c_1\beta p_\star d^{2/p_\star}).$$

Assuming $d \geq \left(\frac{1}{c_1 p_\star}\log\frac{C_1 T}{\delta}\right)^{p_\star/2}$, we have $\beta = \frac{1}{c_1 q d^{2/p_\star}}\log\frac{C_1 T}{\delta} \leq 1$ and

$$\mathbb{P}\left(\|X\|_{p_\star} \leq (1+\beta)\mathbb{E}[\|X\|_{p_\star}]\right) \geq 1 - \frac{\delta}{T}.$$

So with probability at least $1 - \delta/T$

$$\|X\|_{p_\star} \leq (1+\beta)\mathbb{E}[\|X\|_{p_\star}] \leq 2\mathbb{E}[\|X\|_{p_\star}] \leq 4\sqrt{p_\star}d^{1/p_\star},$$

$$\implies \|g_t\|_{p_\star} \leq 4\sqrt{p_\star}\sigma d^{1/p_\star} + \frac{1}{2} \leq 1,$$

where the final inequality follows from $\sigma \leq \frac{1}{8\sqrt{p_\star}d^{1/p_\star}}$

By a union bound over all rounds, with probability $1 - \delta$, $\|g_t\|_{p_\star} \leq 1$ for all rounds $t$.

### G.2.3 Proof of Lemma G.5

Given $Y = i$, the observed feedback at round $t$ is exactly

$$f_t^Y := y_t^T g_t^Y \sim \mathcal{N}(\frac{1}{2} y_{t,i}, \sigma_t^2), \text{ where } \sigma_t^2 = \|y_t\|_2^2 \sigma^2.$$

Denote $p_i$ for the law of the observed feedback up to time $T$ given $Y = i$, i.e., the law of $(f_1^i, ..., f_T^i)$. Denote $p_0$ (use $Y = 0$ in notation), the law of the observed feedback up to time $T$ when all coordinates of $g_t$ are 0-mean for all $t$ (i.e. there is no better direction). Under $p_0$, $f_t^0 \sim \mathcal{N}(0, \sigma_t^2)$. By the definition of the total-variation distance ($d_{\mathrm{TV}}$),

$$\mathbb{E}\big[\mathbb{I}\{y_{t,i} \geq -\alpha\}|Y = 0\big] - \mathbb{E}\big[\mathbb{I}\{y_{t,i} \geq -\alpha\}|Y = i\big] \leq d_{\mathrm{TV}}(p_0, p_i)$$

$$\implies \mathbb{E}\big[\sum_{t=1}^T \mathbb{I}\{y_{t,i} \geq -\alpha\}|Y = 0\big] - \mathbb{E}\big[\sum_{t=1}^T \mathbb{I}\{y_{t,i} \geq -\alpha\}|Y = i\big] \leq T\, d_{\mathrm{TV}}(p_0, p_i).$$

By Pinsker's inequality we have

$$d_{\mathrm{TV}}(p_0, p_i) \leq \sqrt{\frac{1}{2} D_{\mathrm{KL}}(p_0, p_i)}.$$

By the chain rule for the KL divergence / operations on conditional densities:

$$D_{\mathrm{KL}}(p_0, p_i) = \mathbb{E}_{(f_1, ..., f_T) \sim p_0}\Big[\log \frac{p_0(f_1, ..., f_T)}{p_i(f_1, ..., f_T)}\Big]$$

$$= \sum_{t=1}^T \mathbb{E}_{(f_1, ..., f_t) \sim p_0}\Big[\log \frac{p_0(f_t | f_{t-1}..., f_1)}{p_i(f_t | f_{t-1}..., f_1)}\Big]$$

$$= \sum_{t=1}^T \mathbb{E}_{(f_1, ..., f_{t-1}) \sim p_0}\Big\{\mathbb{E}_{f_t \sim p_0(\cdot | f_1, ..., f_{t-1})}\Big[\log \frac{p_0(f_t | f_{t-1}..., f_1)}{p_i(f_t | f_{t-1}..., f_1)}\Big]\Big\}.$$

Now since $y_t$ is a deterministic function of $f_1, ..., f_{t-1}$ and given $y_t$, $f_t \sim \mathcal{N}(0, \sigma_t^2)$ under $p_0$ and $f_t \sim \mathcal{N}(\frac{1}{2} y_{t,i}, \sigma_t^2)$ under $p_i$, we have that the inner expectation is a KL divergence between Gaussians:

$$\mathbb{E}_{f_t \sim p_0(\cdot | f_1, ..., f_{t-1})}\Big[\log \frac{p_0(f_t | f_{t-1}..., f_1)}{p_i(f_t | f_{t-1}..., f_1)}\Big] = \frac{(0 - y_{t,i}/2)^2}{2\sigma_t^2} = \frac{y_{t,i}^2}{8\sigma_t^2}$$

$$\implies D_{\mathrm{KL}}(p_0, p_i) = \sum_{t=1}^T \mathbb{E}_{(f_1, ..., f_{t-1}) \sim p_0}\Big[\frac{y_{t,i}^2}{8\sigma_t^2}\Big] = \sum_{t=1}^T \mathbb{E}_{p_0}\Big[\frac{y_{t,i}^2}{8\sigma_t^2}\Big]$$

$$\implies d_{\mathrm{TV}}(p_0, p_i) \leq \sqrt{\sum_{t=1}^T \mathbb{E}_{p_0}\Big[\frac{y_{t,i}^2}{8\sigma_t^2}\Big]}.$$

Combining gives the result.

### G.3 Case $p \to 1$

**Theorem G.6.** *Fix $T$ and $\delta > 0$. Consider $d > \max\big\{T/\delta, 8^4 e^4 T^2, 8\big\}$ and $p \in [1, 1 + 1/\log d]$. For any OCO algorithm with bandit feedback on $V = \mathcal{B}_p$, there exists a sequence of random linear losses $(\ell_t)_{t \in [T]}$ with sub-gradients $(g_t)_{t \in [T]}$ such that $\|g_t\|_{p_\star} \leq 1$ for all rounds $t$ with probability at least $1 - \delta$ and*

$$\mathbb{E}\big[R_T\big] \geq \frac{T}{16},$$

*where the expectation is with respect to the randomness of the losses.*

### G.3.1 Proof

We use almost the same loss construction as the proof of Theorem G.4. Before the start of the game, draw $Y \sim Unif(1, ..., d)$ and define the losses as $\ell_t(x) = x^T g_t^Y$ where

$$g_{t,i}^Y \sim \begin{cases} \mathcal{N}(0, \sigma^2), & \text{if } i \neq Y, \\ \mathcal{N}(1/2, \sigma^2), & \text{if } i = Y, \end{cases}$$

where $\sigma = (4\sqrt{2}\exp(1)\sqrt{\log d})^{-1}$. The only difference with the proof of Theorem G.4 being the value of $\sigma$. We also follow the same steps until we reach for $\alpha = (2/d)^{1/p}$:

$$\mathbb{E}[R_T] \geq \frac{1-\alpha}{2}\left(\frac{T}{2} - T\frac{1}{2\sigma}\sqrt{\frac{T}{2d}}\right).$$

- From $d \geq 8$, we have $p \leq 2$ and $2^{p+1} \leq 8 \leq d$ so $1 - \alpha = 1 - (2/d)^{1/p} \geq 1 - 2^{-p/p} = 1/2$.
- From the definition of $\sigma = (4\sqrt{2}\exp(1)\sqrt{\log d})^{-1}$

$$\frac{1}{2\sigma}\sqrt{\frac{T}{2d}} = 2\exp(1)\sqrt{\frac{T\log d}{d}} \leq 2\exp(1)\sqrt{\frac{T\sqrt{d}}{d}} = 2\exp(1)\sqrt{\frac{T}{\sqrt{d}}} \leq \frac{1}{4}$$

since $d \geq 8^4 e^4 T^2$. This gives:

$$\frac{T}{2} - T\frac{1}{2\sigma}\sqrt{\frac{T}{2d}} \geq \frac{T}{2} - \frac{T}{4} = \frac{T}{2}.$$

We hence have $\mathbb{E}[R_T] \geq T/16$. The following section ensures that the Lipschitz-condition is satisfied with high-probability.

### G.3.2 Bound on sub-gradients

Recall that $p \leq 1 + \frac{1}{\log d}$ so $p_\star \geq \frac{\log d}{1} + 1$. We have

$$\mathbb{E}[\|X\|_\infty] \leq \sqrt{2\log d}.$$

By the Borell-TIS inequality,

$$\mathbb{P}\left(\|X\|_\infty > \mathbb{E}[\|X\|_\infty] + \beta\right) \leq \exp(-\beta^2/2).$$

So with probability $1 - \delta/T$, we have (using $d > T/\delta$)

$$\|X\|_\infty \leq \mathbb{E}[\|X\|_\infty] + \sqrt{2\log\frac{T}{\delta}} \leq \sqrt{2\log d} + \sqrt{2\log\frac{T}{\delta}} \leq 2\sqrt{2\log d}$$

$$\implies \|g_t\|_{p_\star} \leq \sigma\|X\|_{p_\star} + \frac{1}{2} \leq \sigma d^{1/p_\star}\|X\|_\infty + \frac{1}{2} \leq \sigma\exp(1)2\sqrt{2\log d} + \frac{1}{2} = 1$$

where the final equality follows from $\sigma = (4\sqrt{2}\exp(1)\sqrt{\log d})^{-1}$ and we also used that $d^{1/p_\star} \leq e$.

By a union bound over all rounds, with probability $1 - \delta$, $\|g_t\|_{p_\star} \leq 1$ for all rounds $t$.

# H  Results for Online Mirror Descent (OMD)

## H.1  OMD with uniformly-convex regularisation

The results in this section are from [40] (Proposition 7). We include them for completeness.

Let $\psi : \mathbb{R}^d \to \mathbb{R}$ be a proper, closed and differentiable $\mu$-uniformly convex function[5] on $V$ of degree $r > 2$ w.r.t. a norm $\|\cdot\|$. The Bregman Divergence w.r.t. $\psi$ is defined for all $x, y \in \mathbb{R}^d$ as

$$D_\psi(x, y) = \psi(x) - \psi(y) - \langle \nabla\psi(y), x - y \rangle.$$

Given $x_1 \in V$, at time-step $t = 1, ..., T$, Online Mirror Descent (OMD) with step-size $\eta_t > 0$ outputs the following update where $g_t \in \partial\ell_t(x_t)$,

$$x_{t+1} = \arg\min_{x \in V}\Big\{\eta_t\langle g_t, x \rangle + D_\psi(x, x_t)\Big\}. \tag{15}$$

The standard regret bound of OMD stems from the following one-step regret bound lemma (e.g. see Lemma 6.9 in [36]).

**Lemma H.1.** *The iterates (15) of OMD satisfy for all $u \in V$,*

$$\ell_t(x_t) - \ell_t(u) \leq \langle g_t, x_t - x_{t+1} \rangle + \frac{D_\psi(u, x_t) - D_\psi(u, x_{t+1}) - D_\psi(x_{t+1}, x_t)}{\eta_t}$$

From Lemma H.1 and the uniform convexity of $\psi$, we can bound the regret of OMD.

**Theorem H.2.** *The iterates (15) of OMD with decreasing step-size $\eta_{t+1} \leq \eta_t$ ($1 \leq t \leq T$) satisfy for all $u \in V$ (recall that $r_\star$ is the conjugate of $r$, i.e. $1/r + 1/r_\star = 1$),*

$$\sum_{t=1}^{T} \ell_t(x_t) - \ell_t(u) \leq \max_{1 \leq t \leq T} \frac{D_\psi(u, x_t)}{\eta_T} + \frac{1}{r_\star \mu^{r_\star - 1}} \sum_{t=1}^{T} \eta_t^{r_\star - 1}\|g_t\|_\star^{r_\star}. \tag{16}$$

*If the step-sizes are constant: $\eta_t = \eta$ ($1 \leq t \leq T$), we have*

$$\sum_{t=1}^{T} \ell_t(x_t) - \ell_t(u) \leq \frac{D_\psi(u, x_1)}{\eta} + \frac{\eta^{r_\star - 1}}{r_\star \mu^{r_\star - 1}} \sum_{t=1}^{T} \|g_t\|_\star^{r_\star}. \tag{17}$$

*Proof.* By the uniform convexity of $\psi$, $D_\psi(x_{t+1}, x_t) \geq \frac{\mu}{r}\|x_t - x_{t+1}\|^r$. Using this in Lemma H.1 along with Hölder's inequality, we have for all $u \in V$,

$$\ell_t(x_t) - \ell_t(u) \leq \langle g_t, x_t - x_{t+1} \rangle + \frac{D_\psi(u, x_t) - D_\psi(u, x_{t+1})}{\eta_t} - \frac{\mu}{r}\frac{\|x_t - x_{t+1}\|^r}{\eta_t}$$

$$\leq \|g_t\|_\star\|x_t - x_{t+1}\| + \frac{D_\psi(u, x_t) - D_\psi(u, x_{t+1})}{\eta_t} - \frac{\mu}{r}\frac{\|x_t - x_{t+1}\|^r}{\eta_t}.$$

Consider $f(x) = \frac{1}{r}|x|^r$. Then the Fenchel conjugate of $f$ is $f^\star(y) = \frac{1}{r_\star}|y|^{r_\star}$ (see Lemma 2.2 in [24]) and from Fenchel's inequality, we have $xy \leq \frac{1}{r}|x|^r + \frac{1}{r_\star}|y|^{r_\star}$, which we use in the following,

$$\|g_t\|_\star\|x_t - x_{t+1}\| = \Big(\frac{\eta_t^{1/r}}{\mu^{1/r}}\|g_t\|_\star\Big)\cdot\Big(\frac{\mu^{1/r}}{\eta_t^{1/r}}\|x_t - x_{t+1}\|\Big)$$

$$\leq \frac{1}{r_\star}\Big(\frac{\eta_t^{1/r}}{\mu^{1/r}}\|g_t\|_\star\Big)^{r_\star} + \frac{1}{r}\Big(\frac{\mu^{1/r}}{\eta_t^{1/r}}\|x_t - x_{t+1}\|\Big)^r$$

$$= \frac{\eta_t^{r_\star - 1}}{r_\star \mu^{r_\star - 1}}\|g_t\|_\star^{r_\star} + \frac{\mu}{r}\frac{\|x_t - x_{t+1}\|^r}{\eta_t},$$

---

[5]The function $\psi$ can be defined on a subset $X \subseteq \mathbb{R}^d$ but conditions on its behaviour on the boundary of $X$ are then required for OMD to be well defined. For simplicity, we consider $\psi$ defined on $\mathbb{R}^d$, though the results in this section hold more generally (see Theorem 6.7 of [36] for more detail).

where we used that $r = r_\star/(r_\star - 1)$. Plugging this into the above inequality,

$$\ell_t(x_t) - \ell_t(u) \leq \frac{\eta_t^{r_\star-1}}{r_\star \mu^{r_\star-1}}\|g_t\|_\star^{r_\star} + \frac{D_\psi(u, x_t) - D_\psi(u, x_{t+1})}{\eta_t}. \tag{18}$$

Denoting $D = \max_{1 \leq t \leq T} D_\psi(u, x_t)$, the result follows by summing $t$ over all rounds,

$$\sum_{t=1}^{T} \ell_t(x_t) - \ell_t(u) \leq \sum_{t=1}^{T}\Big(\frac{D_\psi(u, x_t)}{\eta_t} - \frac{D_\psi(u, x_{t+1})}{\eta_t}\Big) + \frac{1}{r_\star \mu^{r_\star-1}}\sum_{t=1}^{T}\eta_t^{r_\star-1}\|g_t\|_\star^{r_\star}$$

$$= \frac{D_\psi(u, x_1)}{\eta_1} - \frac{D_\psi(u, x_{T+1})}{\eta_T} + \sum_{t=1}^{T-1}\Big(\frac{1}{\eta_{t+1}} - \frac{1}{\eta_t}\Big)D_\psi(u, x_{t+1}) + \frac{1}{r_\star \mu^{r_\star-1}}\sum_{t=1}^{T}\eta_t^{r_\star-1}\|g_t\|_\star^{r_\star}$$

$$\leq \frac{D}{\eta_1} + D\sum_{t=1}^{T-1}\Big(\frac{1}{\eta_{t+1}} - \frac{1}{\eta_t}\Big) + \frac{1}{r_\star \mu^{r_\star-1}}\sum_{t=1}^{T}\eta_t^{r_\star-1}\|g_t\|_\star^{r_\star}$$

$$= \frac{D}{\eta_1} + D\Big(\frac{1}{\eta_T} - \frac{1}{\eta_1}\Big) + \frac{1}{r_\star \mu^{r_\star-1}}\sum_{t=1}^{T}\eta_t^{r_\star-1}\|g_t\|_\star^{r_\star}$$

$$= \frac{D}{\eta_T} + \frac{1}{r_\star \mu^{r_\star-1}}\sum_{t=1}^{T}\eta_t^{r_\star-1}\|g_t\|_\star^{r_\star}.$$

For constant step-size, the result follows similarly by summing (18) over $t$, giving a telescoping sum,

$$\sum_{t=1}^{T} \ell_t(x_t) - \ell_t(u) \leq \frac{D_\psi(u, x_1) - D_\psi(u, x_{T+1})}{\eta} + \frac{\eta^{r_\star-1}}{r_\star \mu^{r_\star-1}}\sum_{t=1}^{T}\|g_t\|_\star^{r_\star}$$

$$\leq \frac{D_\psi(u, x_1)}{\eta} + \frac{\eta^{r_\star-1}}{r_\star \mu^{r_\star-1}}\sum_{t=1}^{T}\|g_t\|_\star^{r_\star},$$

which concludes the proof. $\qquad\square$

### H.1.1 Regret bounds

When we have $L$-Lipschitz losses w.r.t. $\|\cdot\|$ and we can bound $D_\psi(u, x_1) < D$, then the regret bound (17) for constant step-sizes becomes

$$R_T \leq \frac{D}{\eta} + \eta^{r_\star-1}\frac{TL^{r_\star}}{r_\star \mu^{r_\star-1}}.$$

Assuming the time-horizon $T$ is known, optimising the above bound w.r.t. $\eta$ using Lemma C.4 gives

$$R_T \leq \frac{r^{1/r}}{\mu^{1/r}}LD^{1/r}T^{1/r_\star}, \tag{19}$$

for $\eta = (Dp/T)^{1/r_\star}\mu^{1/r}/L$. With $r = 2$, we recover the standard regret bound of OMD using a strongly-convex regulariser $R_T \leq L\sqrt{2DT/\mu}$.

### H.1.2 Anytime and adaptive bounds

When $T$ is unknown, we can use the bound in (16) and the time-varying step-size

$$\eta_t = \frac{D_{\max}^{1/r_\star}(r-1)^{1/r_\star}\mu^{1/r}}{L} \cdot \frac{1}{t^{1/r_\star}} \quad \text{to get} \quad R_T \leq \frac{LD_{\max}^{1/r}r^{1/r}r_\star^{1/r_\star}T^{1/r_\star}}{\mu^{1/r}}, \tag{20}$$

where $D_{\max}$ is a bound on $\max_{1 \leq t \leq T} D_\psi(u, x_t)$. Though this can be unbounded when $D$ is bounded, for our purposes of $\ell_p$-balls, $D_{\max}$ will only be a constant away from $D$. The doubling trick can also be used to obtain anytime bounds that depend on $D$ instead of $D_{\max}$ (see e.g. [27]). This uses constant step-size OMD on time-horizons of doubling lengths until the unknown true $T$ is reached.

We can also obtain bounds that adapt to the sequence of observed subgradients of the form

$$R_T = \frac{D_{\max}^{1/r} r^{1/r} r_\star^{1/r_\star}}{\mu^{1/r}} \cdot \left( \sum_{i=1}^{T} \|g_i\|_\star^{r_\star} \right)^{1/r_\star}$$

by using $\eta_t = D_{\max}^{1/r_\star}(r-1)^{1/r_\star}\mu^{1/r}/(\sum_{i=1}^{T}\|g_i\|_\star^{r_\star})^{1/r_\star}$. This follows the same lines as for OMD with strongly convex regulariser (see Section 4.2.1 of [36]).

## H.2   OMD on $\ell_p$-balls

- **For the low-dimensional setting,** consider OMD with regulariser $\phi_2(x) = \frac{1}{2}\|x\|_2^2$. We have $D_{\max} = \sup_{x,y \in \mathcal{B}_p} \frac{1}{2}\|x-y\|_2^2 = 2d^{1-2/p}$ and using that $\phi_2$ is 1-strongly-convex with respect to $\|\cdot\|_2$, we have from (20) with $r = 2$ and $\eta_t = \frac{1}{L}\sqrt{\frac{2d^{1-2/p}}{t}}$:

$$R_T \leq 2L\sqrt{2d^{1-2/p}T}.$$

- **For the high-dimensional setting,** consider OMD with regulariser $\phi_p(x) = \frac{1}{p}\|x\|_p^p$. We have $D_{\max} = \sup_{x,y \in \mathcal{B}_p} D_{\phi_p}(x,y) = 2$ (see below) and using that $\phi_p$ is $2^{1-p}$-uniformly-convex of degree $p$ with respect to $\|\cdot\|_p$, we have from (20) with $r = p$ and $\eta = \frac{1}{L}\left(\frac{p-1}{t}\right)^{1-1/p}$:

$$R_T \leq 2p^{1/p}p_\star^{1-1/p_\star}LT^{1-1/p}.$$

To show $D_{\max} = 2$: Fix $x, y \in \mathcal{B}_p$ and $\psi(x) = \frac{1}{p}\|x\|_p^p$. The sign, power and absolute value functions below are applied component-wise to vectors.

$$D_\psi(x,y) = \psi(x) - \psi(y) - \langle \nabla\psi(y), x - y \rangle$$

$$= \frac{1}{p}\|x\|_p^p - \frac{1}{p}\|y\|_p^p + \sum_{i=1}^{d}\left\{ \mathrm{sign}(y_i) \cdot |y_i|^{p-1}(y_i - x_i) \right\}$$

$$= \frac{1}{p}\|x\|_p^p - \frac{1}{p}\|y\|_p^p + \sum_{i=1}^{d}\left\{ |y_i|^p - \mathrm{sign}(y_i) \cdot |y_i|^{p-1}x_i \right\}$$

$$= \frac{1}{p}\|x\|_p^p + \left(1 - \frac{1}{p}\right)\|y\|_p^p - \sum_{i=1}^{d}\left\{ \mathrm{sign}(y_i) \cdot |y_i|^{p-1}x_i \right\}$$

$$\leq \frac{1}{p} + \left(1 - \frac{1}{p}\right) - \sum_{i=1}^{d}\left\{ \mathrm{sign}(y_i) \cdot |y_i|^{p-1}x_i \right\}.$$

We show that the last term is bounded by 1 by using Holder's inequality,

$$\langle \mathrm{sign}(y) \cdot |y|^{p-1}, x \rangle \leq \| |y|^{p-1}\|_q \|x\|_p \leq 1,$$

where in the last inequality we used (recall that $q = p/(p-1)$)

$$\| |y|^{p-1}\|_q = \left( \sum_{i=1}^{d} |y_i|^{q\cdot(p-1)} \right)^{1/q} = \left( \sum_{i=1}^{d} |y_i|^p \right)^{1/q} = \|y\|_p^{p/q} \leq 1.$$

Hence $\sup_{x,y \in \mathcal{B}_p} D_\psi(x,y) \leq 2 = D_{\max}$.

- **We now show how to achieve anytime optimal bounds.** Fix $t_0 = \left( \sqrt{2}p^{1/p}p_\star^{1/p_\star} \right)^{2p/(p-2)} \cdot d$.

  **Proposition H.3.** *Consider running OMD with the following regularizers*

$$\psi_t(x) = \begin{cases} \phi_p(x) = \frac{1}{p}\|x\|_p^p, & \eta_t = \frac{(p-1)^{1/p_\star}}{Lt^{1/p_\star}}, & \text{if } t \leq t_0, \\ \phi_2(x) = \frac{1}{2}\|x\|_2^2, & \eta_t = \frac{\sqrt{2d^{1-2/p}}}{L\sqrt{t}}, & \text{if } t > t_0. \end{cases}$$

  *Assume $\ell_t$ convex, closed, and $\partial\ell_t(x_t)$ not empty. Then, OMD guarantees*

$$R_T \leq \begin{cases} 2p^{1/p}p_\star^{1/p_\star}LT^{1/p_\star}, & \text{if } t \leq t_0, \\ 2L\sqrt{2Td^{1-2/p}}, & \text{if } t > t_0 \end{cases}$$

*Proof.* If $T \leq t_0$, we have just run OMD with $\phi_p$ as regulariser over all rounds and the regret bound is the one for the high-dimensional setting above.

Otherwise, from the standard bounds from the OMD analysis

$$R_T \leq \sum_{t=1}^{t_0} \left( \frac{D_{\psi_p}(u, x_t)}{\eta_t} - \frac{D_{\psi_p}(u, x_{t+1})}{\eta_t} \right) + \frac{L^{p_\star}}{p_\star \mu_p^{p_\star-1}} \sum_{t=1}^{T} \eta_t^{p_\star-1}$$

$$+ \sum_{t=t_0+1}^{T} \left( \frac{D_{\psi_2}(u, x_t)}{\eta_t} - \frac{D_{\psi_2}(u, x_{t+1})}{\eta_t} \right) + \frac{L^2}{2\mu_2} \sum_{t=t_0+1}^{T} \eta_t$$

$$\leq \frac{D_p}{\eta_{t_0}} + \frac{L^{p_\star}}{p_\star \mu_p^{p_\star-1}} \sum_{t=1}^{T} \eta_t^{p_\star-1} + \frac{D_2}{\eta_T} + \frac{L^2}{2\mu_2} \sum_{t=t_0+1}^{T} \eta_t$$

$$\leq 2p^{1/p} p_\star^{1/p_\star} L t_0^{1/p_\star} + \frac{D_2}{\eta_T} + \frac{L\sqrt{D_2}}{2\mu} \sum_{t=t_0+1}^{T} \frac{1}{\sqrt{t}}$$

$$\leq 2p^{1/p} p_\star^{1/p_\star} L t_0^{1/p_\star} + L\sqrt{TD_2} + L\sqrt{D_2}(\sqrt{T} - \sqrt{t_0})$$

$$= 2L\sqrt{2Td^{1-2/p}} + 2p^{1/p} p_\star^{1/p_\star} L t_0^{1/p_\star} - L\sqrt{2d^{1-2/p} t_0},$$

where we used $D_2 = \sup_{x,y \in \mathcal{B}_p} D_{\psi_2}(x,y) \leq 2d^{1-2/p}$, $D_p = \sup_{x,y \in \mathcal{B}_p} D_{\psi_p}(x,y) \leq 2$, $\mu_2 = 1$ and $\mu_p = 2^{1-p}$. The proof is concluded by noting that $2p^{1/p} p_\star^{1/p_\star} L t_0^{1/p_\star} - L\sqrt{2d^{1-2/p} t_0}$ is negative for $t_0 = \left( \sqrt{2} p^{1/p} p_\star^{1/p_\star} \right)^{2p/(p-2)} \cdot d$. $\qquad \square$

## H.3 Failure of fixed separable regularisation for OMD

**Proposition H.4.** *OMD with regulariser $\psi \in \Psi$ and any sequence of decreasing $\eta_t$ cannot be optimal across all dimensions. Specifically there are no constants $c_h, c_l > 0$ such that for all $T$, $R_T \leq c_h L T^{1-1/p}$ for all $d > T$ and $R_T \leq c_l L \sqrt{T d^{1-2/p}}$ for all $d \leq T$.*

The proof is identical to the proof of Theorem 4.6 for FTRL with the corresponding versions of Lemma 4.4 and Lemma 4.7 for OMD given below.

**Lemma H.5.** *[Lemma 4.7 for OMD] Consider $d = 1$ ($V = \mathcal{B}_p = [-1, 1]$) and $\psi \in \mathcal{F}$. OMD with regulariser $\psi$ and arbitrary decreasing step-size $\eta_t$ can only guarantee $R_T \leq cL\sqrt{T}$ for some constant $c > 0$ and all sufficiently large $T$ if for all $x \in [-1, 1]$, $\psi(x) \geq \frac{\psi(1/2)}{100c^2} x^2$.*

*Proof.* Assume there exists a constant $c > 0$ such that for all $T$ and any sequence of losses, $R_T \leq cL\sqrt{T}$. Consider $T > 16c^2$ and a multiple of 4.

By considering the dual-version of OMD, we have that if there are no projections up to time $t$, the update of OMD at time $t + 1$ can be written as

$$x_{t+1} = \nabla \psi_V^\star \left( -\sum_{i=1}^{t} \eta_i g_i \right) = \arg\min_{x \in V} \left\{ \psi(x) + \sum_{i=1}^{t} \eta_i g_i^T x \right\}, \tag{21}$$

where $\psi_V^\star$ is the restriction of the fenchel conjugate of $\psi$ to $V = \mathcal{B}_p = [-1, 1]$.

We now follow the same steps as FTRL with a slight modification to the loss $\ell_t(x) = x \cdot g_t$ where $g_t \in [-1, 1]$ is now defined as

$$g_t = \begin{cases} -\frac{\eta_{t+1}}{\eta_t} \cdot L, & t \leq \frac{T}{2} \text{ odd}, \\ L, & t \leq \frac{T}{2} \text{ even}, \\ -L, & t > \frac{T}{2}, \end{cases}$$

Assume $\eta_2$ is small enough s.t. $x_2 \in \text{int } V = (-1, 1)$ (i.e. no projection is needed). If not, we can modify the losses slightly so that $x_3 = 0$ (if $\eta_3$ is large enough, if it is not then set $g_1 = g_2 = 0$ and start the above losses from $t = 3$) and then proceed similarly (i.e. if $\eta_3$ is still so large that $x_4 = 1$, then again modify the losses slightly so that $x_5 = 0$ etc). Set $\eta = \eta_{T/2}$. With this sequence of losses, the points played by OMD satisfy

- for $t \leq T/2 + 1$ and $t$ odd, we have $\sum_{s=1}^{t-1} \eta_s g_s = 0$, so $x_t = 0$.

- for $t \leq T/2 + 1$ and $t$ even, we have $\sum_{s=1}^{t-1} \eta_s g_s = -\eta_t \cdot L$ so $x_t = \arg\min_{x \in [-1,1]} \{-\eta_t x + \psi(x)\}$.
  For $t < t' \leq T/2$ (both even), we have

$$
\begin{aligned}
-\eta_{t'} L x_{t'} + \psi(x_{t'}) &\leq -\eta_{t'} L x_t + \psi(x_t) && \text{using the definition of } x_{t'} \\
&= -\eta_t L x_t + \psi(x_t) + L(\eta_t - \eta_{t'}) x_t \\
&\leq -\eta_t L x_{t'} + \psi(x_{t'}) + L(\eta_t - \eta_{t'}) x_t && \text{using the definition of } x_t \\
\implies (\eta_t - \eta_{t'}) L x_{t'} &\leq (\eta_t - \eta_{t'}) L x_t \\
\implies x_{t'} &\leq x_t && \text{using that } \eta_{t'} \leq \eta_t.
\end{aligned}
$$

So for all $t \leq T/2$ even, we have $x_t \geq x_{T/2}$.

- for $t > T/2$, we have $\sum_{s=1}^{t-1} \eta_s g_s \geq \sum_{s=1}^{t} \eta_s g_s$ so $x_t \leq x_{t+1}$.

The regret can be written as follows

$$
R_T = \sum_{t=1}^{T} \ell_t(x_t) - \left(-\frac{LT}{2}\right) \geq \frac{LT}{2} + \frac{LT}{4} x_{T/2} - L \sum_{t=T/2+1}^{T} x_t \tag{22}
$$

Following similar steps as the proof of Lemma 4.7 for FTRL, we get

1. We first show that $x_{\frac{T}{2} + \lceil 2c\sqrt{T} \rceil} \geq \frac{1}{2}$: if not, $x_{\frac{T}{2} + \lceil 2c\sqrt{T} \rceil} < \frac{1}{2}$ and from (22):

$$
\begin{aligned}
R_T &\geq \frac{LT}{2} - L \sum_{t=T/2+1}^{T/2 + \lfloor 2c\sqrt{T} \rfloor} x_t - L \sum_{t=T/2+\lceil 2c\sqrt{T} \rceil}^{T} x_t \\
&> \frac{LT}{2} - L \sum_{t=T/2+1}^{T/2 + \lfloor 2c\sqrt{T} \rfloor} \frac{1}{2} - L \sum_{t=T/2+\lceil 2c\sqrt{T} \rceil +1}^{T} 1 \\
&= \frac{LT}{2} - \frac{L}{2} \lfloor 2c\sqrt{T} \rfloor - L \left(T - \lceil 2c\sqrt{T} \rceil - \frac{T}{2}\right) \\
&\geq \frac{L}{2} \lceil 2c\sqrt{T} \rceil \\
&> cL\sqrt{T},
\end{aligned}
$$

which contradicts our initial assumption that $R_T \leq cL\sqrt{T}$ so we must have $x_{\frac{T}{2} + \lceil 2c\sqrt{T} \rceil} \geq \frac{1}{2}$. Note that $2c\sqrt{T} < T/2$ is ensured by $T > 16c^2$.

2. Next, we show that $\eta \geq \frac{\psi(1/2)}{2cL\sqrt{T}}$. Until the points reach 1 there are no projections so we can use (21) to write the OMD update as (note that even if $x_{\frac{T}{2} + \lceil 2c\sqrt{T} \rceil} = 1$ the following still holds)

$$
x_{\frac{T}{2} + \lceil 2c\sqrt{T} \rceil} = \arg\min_{x \in V} \left\{ \psi(x) + \sum_{i=1}^{\frac{T}{2} + \lceil 2c\sqrt{T} \rceil - 1} \eta_i g_i \cdot x \right\} = \arg\min_{x \in V} \left\{ \psi(x) - x \sum_{i=\frac{T}{2}+1}^{\frac{T}{2} + \lceil 2c\sqrt{T} \rceil - 1} \eta_i \right\}
$$

$$
\implies -x_{\frac{T}{2} + \lceil 2c\sqrt{T} \rceil} \sum_{i=\frac{T}{2}+1}^{\frac{T}{2} + \lceil 2c\sqrt{T} \rceil - 1} \eta_i + \psi(x_{\frac{T}{2} + \lceil 2c\sqrt{T} \rceil}) \leq 0
$$

$$
\implies \psi(1/2) \leq \sum_{i=\frac{T}{2}+1}^{\frac{T}{2} + \lceil 2c\sqrt{T} \rceil - 1} \eta_i \leq 2c\sqrt{T}\eta
$$

$$
\implies \eta \geq \frac{\psi(1/2)}{2c\sqrt{T}}.
$$

where in the second implication, we used that $\psi(x_{\frac{T}{2}+\lceil 2c\sqrt{T}\rceil}) \geq \psi(1/2)$ (since $\psi$ is increasing on $[0,1]$ and $x_{\frac{T}{2}+\lceil 2c\sqrt{T}\rceil} \geq 1/2$), $x_{\frac{T}{2}+\lceil 2c\sqrt{T}\rceil} \leq 1$ and $\eta_i \leq \eta_{T/2} = \eta$ for all $i \geq T/2$.

The remaining steps are identical to the proof of Lemma 4.7. $\qquad\square$

**Lemma H.6** (Lemma 4.4 for OMD). *Consider $V = \mathcal{B}_p$ with $p > 2$ and assume losses are $L$-Lipschitz in $\ell_p$-norm. Let $\psi$ be a convex function satisfying for some $\mu > 0$ and any $x \in \mathbb{R}^d$, $\psi(x) \geq \frac{\mu}{2}\|x\|_2^2$. If $d \geq (4T/\mu)^{p/(p-2)}$, there exists a sequence of linear $L$-Lipschitz losses (in $\ell_p$-norm) for which OMD with regulariser $\psi(x)$ and any sequence of decreasing $\eta_t$ suffers regret $R_T \geq \frac{1}{8}LT$.*

*Proof.* We consider the loss construction described in Appendix E.1 with a slight modification to the loss $\ell_t(x) = L \cdot x^T g_t$ where $g_t \in \mathcal{B}_{p_\star}$ is now defined as

$$
g_t = \begin{cases} -\frac{\eta_{t+1}}{\eta_t} L \cdot e_1, & t \leq \frac{T}{2} \text{ odd}, \\ L \cdot e_1, & t \leq \frac{T}{2} \text{ even}, \\ -L \cdot v, & t > \frac{T}{2}. \end{cases}
$$

By again considering the dual-version of OMD, we have that if there are no projections up to time $t$, the update of OMD at time $t+1$ can be written as

$$
x_{t+1} = \nabla\psi_V^\star\left(-L\sum_{i=1}^t \eta_i g_i\right) = \arg\min_{x \in V}\left\{\psi(x) + L\sum_{i=1}^t \eta_i g_i^T x\right\}, \tag{23}
$$

where $\psi_V^\star$ is the restriction of the fenchel conjugate of $\psi$ to $V = \mathcal{B}_p$.

- First we consider $t \leq T/2$. As in the proof of Lemma H.5, assume $\eta_2$ is small enough s.t. $x_2 \in \text{int } \mathcal{B}_p$ (i.e. no projection is needed). By (23), the steps are then the same as for FTRL since when $t$ is odd, $x_t = 0$ and when $t$ is even, $x_t = \arg\min_{x \in \mathcal{B}_p}\left\{\psi(x) - L\eta_t e_1^T x\right\}$. So we have

$$
\sum_{t=1}^{T/2} x_t^T g_t \geq \begin{cases} T/4, & \text{if } \eta_{T/2} \geq 2/L \\ 0, & \text{if } \eta_{T/2} < 2/L. \end{cases}
$$

Hence if $\eta_{T/2-1} \geq 2/L$, we have $R_T \geq LT/4$ and the statement of the theorem holds. If $\eta_{T/2-1} < 2/L$, we look to the second half of the rounds.

- Let's now consider $t > T/2$ and assume $\eta_{T/2-1} < 2/L$. Fix $\beta_t = t - T/2 - 1$. Until the points reaches the boundary there are no projections so we can use (21) to write the OMD update as

$$
x_t = \arg\min_{x \in \mathcal{B}_p}\left\{\psi(x) + L\sum_{i=1}^{t-1} \eta_i g_i^T x\right\} = \arg\min_{x \in \mathcal{B}_p}\left\{\psi(x) - L \cdot v^T x \sum_{i=T/2+1}^{t-1} \eta_i\right\}
$$

$$
= \arg\min_{x \in \mathcal{B}_p}\sum_{i=1}^d\left\{g(x_i) - Lx_i d^{-1/q}\sum_{i=T/2+1}^{t-1}\eta_i\right\}.
$$

Let $u = v/\|v\|_p$ be the competitor. Note that $x_t = \lambda_t u$ (since the update is coordinate invariant) so only reaches the boundary once $x_t = u$ and for which the above equality still holds (this is true because it is true for the last iterate before the projection and then $\sum_{i=1}^{t-1}\eta_i g_i$ is greater than for this last iterate so the argmin will give $x_t = u$). We have $\lambda_t \geq 0$ and

$$
\psi(x_t) \geq \frac{\mu}{2}\|x_t\|_2^2 = \frac{1}{2}\lambda_t^2 \mu d^{1-2/p}.
$$

Now from the OMD update,

$$\psi(x_t) - Lv^T x_t \sum_{i=T/2+1}^{t-1} \eta_i \leq 0 \implies \frac{1}{2}\lambda_t^2 \mu d^{1-2/p} - \lambda_t L \sum_{i=T/2+1}^{t-1} \eta_i \leq 0$$

$$\implies \lambda_t \leq \frac{2L}{\mu d^{1-2/p}} \sum_{i=T/2+1}^{t-1} \eta_i \leq \frac{2L\beta_t \eta}{\mu d^{1-2/p}} \leq \frac{4\beta_t}{\mu d^{1-2/p}}$$

$$\implies \ell_t(x_t) = -Lv^T x_t = -L\lambda_t \geq -\frac{4L\beta_t}{\mu d^{1-2/p}},$$

since $\eta_i \leq \eta_{T/2} \leq 2/L$ for all $i \geq T/2$. If $d \geq (4T/\mu)^{p/(p-2)}$, we have for all $t \leq T$

$$\ell_t(x_t) \geq -\frac{4L\beta_t}{\mu d^{1-2/p}} \geq -\frac{L}{2}$$

$$\implies R_T \geq \frac{LT}{2} + \sum_{t=T/2+1}^{T} \ell_t(x_t) \geq \frac{LT}{2} - \frac{LT}{4} = \frac{LT}{4}.$$

If $T$ is not divisible by $4$ and we use $T-1, T-2$ or $T-3$, we have $R_T \geq \frac{L(T-3)}{4} \geq \frac{LT}{8}$ for $T \geq 6$, concluding the proof. $\qquad\square$

