# OpenReview forum: "On the necessity of adaptive regularisation: Optimal anytime online learning on $\boldsymbol{\ell_p}$-balls"
_NeurIPS.cc/2025/Conference — NeurIPS 2025 spotlight_

### Official Review · Reviewer_QhTE · 2025-06-23

**Clarity:** 3
**Significance:** 3
**Originality:** 3
**Rating:** 5
**Confidence:** 3

**Summary:**

This paper studies online convex optimization over the $\ell_p$ ball for $p>2$, where the loss functions are Lipschitz continuous with respect to the $\ell_p$-norm. The case when $p\in[1,2]$ has already been studied previously. Let $T$ denote the time horizon and $d$ the dimension of the space. Existing work has shown that the optimal regrets differ between the low-dimensional regime ($d<T$) and the high-dimensional regime ($d\geq T$).

For lower bounds, the authors prove that
* FTRL with $\ell_2$-norm regularization, which is known to be optimal in low-dim, is sub-optimal in high-dim.
* FTRL with $\ell_p^p$-norm regularization, which is known to be optimal in high-dim, is sub-optimal in low-dim.
* FTRL with a fixed strongly-convex or fixed separable regularizer cannot achieve optimal regret in both settings simultaneously.

For upper bounds, the authors propose an FTRL algorithm with adaptive regularization and show that it achieves the optimal regret in both settings.

**Questions:**

The following questions are simply out of my curiosity.

1. The results in this paper show that non-adaptive separable regularizers cannot be optimal in both settings simultaneously, which applies to many existing FTRL algorithms. Are you aware of any non-separable regularizers in the literature?
2. Could you provide any intuition or motivation behind the loss construction (Section E.1) used in the lower bound proof?

**Ethical Concerns:**

["NO or VERY MINOR ethics concerns only"]

**Final Justification:**

This is a good paper and there are no significant weaknesses. In their rebuttal, the authors also provide intuitions for their proofs. I am happy to recommend acceptance.

**Limitations:**

Yes.

**Paper Formatting Concerns:**

No issues.

**Quality:**

4

**Strengths And Weaknesses:**

I have read the main paper and skimmed through Appendices A-E.

# Strengths

### Quality and Clarity
This paper is easy to follow. The motivation is clear, and the related work is adequately addressed. The theoretical results are sound and rigorously supported.

---
### Originality
This paper identifies a gap in the existing literature: Although optimal algorithms for both the high-dim ($T<d$) and the low-dim setting ($T\geq d$) are known, there is no single algorithm that is asymptotically optimal for both settings simultaneously. This gap was not previously recognized and is rigorously established in the paper.

Furthermore, the authors extend the lower bounds to non-adaptive strongly convex and non-adaptive separable regularizers. As far as I know, these lower bounds are novel and are the paper's main technical contributions.

---
### Significance
This paper not only closes the above gap but also raises several interesting questions, such as the universal optimality of FTRL and the benefit of non-separable regularizers. It has the potential to guide future research, and I am happy to recommend acceptance.

---

# Weaknesses
No significant weaknesses are found. Below are some minor comments:

* In Line 110, should $T^{1/p}$ be $T^{1-1/p}$?
* In Theorem 4.2, I suggest including the definition of a "sign-invariant regularizer."
* In Line 279, it is stated that "its regret guarantees are provably dimension-independent and consequently sub-optimal in low-dimensions." I don't fully understand this implication. My understanding is that the rate $T^{1/p_\star} = T^{1-1/p}$ in Proposition 4.5 is larger than the optimal rate $T^{1/2}$ for $p>2$, and therefore the uniformly convex regularizer is not optimal. Could you elaborate more on this statement?

---

> ### Author Rebuttal · Authors · 2025-07-27
>
> We thank the reviewer for their positive and constructive feedback. We address your comments, suggestions and questions below.
>
> # Minor comments:
>
> - In line 110, you are right it should be $T^{1-1/p}$, thank you for pointing this out!
> - We will include a definition of “sign-invariant regularizer” in the final version.
> - In Line 279, the phrasing we used was confusing. We meant the following: for $p>2$, the optimal rate in the low-dimensional setting is $O(\sqrt{Td^{1-2/p}})$ which is dimension-dependent. Proposition 4.5 shows that the regret guarantees do not improve for smaller dimensions, so are dimension-independent and therefore sub-optimal (so “provably dimension-independent and sub-optimal” in the sense that the optimal rate is dimension-dependent). We realise this phrasing is confusing and will amend it in the final version.
>
> # Questions:
>
> **Q1: Are you aware of any non-separable regularizers in the literature?**
>
> - There is for example the squared r-norm $\Vert x \Vert_r^2 = (\sum_{i=1}^d \|x_i\|^r)^{2/r}$ for $r\in(1,2]$ which is not separable and has been considered in the literature (e.g. see section 6.7 in [1]). It is however strongly convex with respect to the r-norm so Theorem 4.2 guarantees that it is sub-optimal in high dimension.
>
> **Q2: Could you provide any intuition or motivation behind the loss construction (Section E.1) used in the lower bound proof?**
> - For the first half of the rounds we successively use linear losses with gradient $\pm e_1$.
> - For the second half we use a fixed linear loss with gradient a rescaled version of a point in the corner of the $\ell_p$-ball (so that competitor is this point in the corner which may be hard to reach).
> - The different losses in the two halves of the rounds capture some sort of bias-variance trade-off of the algorithm:
> If the step-size is large then the algorithm will quickly reach the corner of the ball (the competitor) in the second half of the rounds but in the first it will suffer large regret because it moves too fast through the space and every other round it will get to close to $e_1$ which will make it suffer large loss in the next round (it has high “variance”).
> - If the step-size is too small, it will not suffer much regret in the first half of the rounds but in the second half it will not be able to reach the corner of the ball (the competitor) sufficiently quickly and suffer large-regret (it has high “bias”).
> - We chose this loss construction so that a large or small step-size does not benefit the algorithm: A perfectly tuned step-size reaches the correct balance between the regret suffered from both halves and corresponds to the optimal regret (both halves also correspond to the two terms that appear in the regret bounds of OMD/FTRL before tuning of the step-size). We will include this intuition in the corresponding appendix in the final version of the paper.
>
> [1] Francesco Orabona. A modern introduction to online learning. 2019. Preprint, arXiv: 1912.13213.

---

> > ### Comment · Reviewer_QhTE · 2025-08-03
> >
> > I appreciate the authors' detailed and insightful responses. I will keep my rating unchanged and continue to recommend acceptance.

---

### Official Review · Reviewer_5h3T · 2025-07-01

**Clarity:** 2
**Significance:** 3
**Originality:** 3
**Rating:** 5
**Confidence:** 3

**Summary:**

This paper investigates the fundamental question of whether Follow-the-Regularized-Leader (FTRL) algorithms can achieve optimal regret bounds across all dimensional regimes using a single fixed regularizer across time, specifically for online convex optimization on $\ell_p$-balls with $p > 2$.
It is known that optimal regret rates exhibit a dimensional shift: in low-dimensional settings ($d \leq T$), the optimal rate is $O(L\sqrt{T d^{1-2/p}})$ achieved using strongly-convex regularizers such as $||x||^2$, while in high-dimensional settings ($d > T$), the optimal rate is $O(LT^{1-1/p})$ achieved using uniformly-convex regularizers of degree $p$. This creates a challenge for "anytime" algorithms that must perform optimally without knowing the time horizon $T$ in advance as the optimal regularizer depends on the time horizon.

The paper's major contributions are in two directions:
1) Algorithm: The authors design an adaptive FTRL algorithm that achieves anytime optimality by switching regularizers at a carefully chosen threshold $t_0 = 3^{-2p/(p-2)}d$. The algorithm uses uniformly-convex regularization initially (ideal for high dimensions) and switches to strongly-convex regularization when the low-dimensional regime is reached.

2) Fundamental impossibility result: The paper's central contribution proves that this adaptivity is fundamentally necessary for separable regularizers. Specifically, they show that no fixed separable regularizer (of the form $\psi(x) = \sum_i=1^d g(x_i)$) can simultaneously achieve optimal regret in both dimensional regimes. This impossibility result rules out regularizers like $||x||_r^r$.

Additionally, there are also algorithm-specific lower bounds for particular regularizers and extensions of the analysis to bandit feedback, showing that sub-linear regret becomes impossible in sufficiently high dimensions for linear bandits on $\ell_p$-balls.

**Questions:**

- What is the main technical issue with extending the present proofs to non-separable regularizers? Is it the analog of Lemma 4.7?
- Could you explain a bit more about the prior work in this domain? Have negative results of the flavor presented in this work appeared before in the literature?
- Any insights on the algorithmic component to complement the impossibility results in Section 5?

**Ethical Concerns:**

["NO or VERY MINOR ethics concerns only"]

**Final Justification:**

I keep my score as this is a good paper and recommend acceptance.

**Limitations:**

- Yes, assumptions are clearly stated

**Quality:**

3

**Strengths And Weaknesses:**

Strengths

- Correctness: The mathematical analysis is rigorous and sound. The proofs are carefully constructed and the main impossibility result (Theorem 4.6) is established through an argument connecting quadratic growth requirements to high-dimensional failure.

- Studies an interesting question: The paper addresses a fundamental and practically relevant question about the universality of FTRL in online learning algorithms. Understanding when adaptive regularization is truly necessary versus when fixed regularizers suffice has implications for algorithm design, especially in settings where the dimensional regime is unknown a priori.

- The algorithm is simple and intuitive, the lower bound is quite interesting too. The adaptive algorithm switches between two well-understood regularizers at a natural threshold, making it easy to implement and understand. The impossibility result provides insight by showing that the intuitive approach of using a single "compromise" regularizer fundamentally cannot work, which is both (to me) surprising and illuminating.

Weaknesses

- Heavy technical machinery for simple switch: While the core algorithmic idea is straightforward (switch regularizers based on time), the paper employs substantial technical apparatus to analyze this simple procedure. The specific threshold calculation and analysis feel disproportionate to what is essentially an elegant but basic adaptive strategy. The paper would also benefit from more intuition behind the proofs and sketches for some of the important building blocks that are relegated to the appendix (e.g. Lemma 4.7).

- Bandit section is underdeveloped: Section 5 on bandit feedback feels somewhat disconnected from the main narrative and lacks the depth of analysis given to the full-information setting. The impossibility result for high-dimensional bandits is interesting but doesn't provide corresponding positive results or adaptive algorithms for the achievable regimes, leaving this part of the story incomplete.

- Assumptions on separability: The main impossibility result is restricted to separable regularizers, which, while covering many important cases, leaves open the most general version of the question. This limitation feels somewhat artificial and prevents the paper from providing a definitive answer to the broader question of FTRL universality with fixed regularizers.

---

> ### Author Rebuttal · Authors · 2025-07-27
>
> We thank the reviewer for their positive, constructive and thorough feedback. We address your comments, suggestions and questions below.
>
> # Weaknesses:
>
> **Heavy technical machinery for simple switch**
> - While the proof of Theorem 3.1 in Appendix D is tedious and may obscure from the fact that it involves mostly algebraic computations, we attempted to make clear that the result consists of arguing with a sequence of elementary arguments (line 216: “The proof is in Appendix D and consists of a careful application of Theorem 2.3.” where Theorem 2.3 is just the general regret bound for FTRL).
>
> **The paper would also benefit from more intuition behind the proofs and sketches for some of the important building blocks that are relegated to the appendix (e.g. Lemma 4.7)**
> - This is a good point - we will include intuitions for some of these results/proofs in the final version - at least in the appendix and in the main text if space allows (see also answer to reviewer QhTE).
>
> **Bandit section is underdeveloped: Section 5 on bandit feedback feels somewhat disconnected from the main narrative and lacks the depth of analysis given to the full-information setting. The impossibility result for high-dimensional bandits is interesting but doesn't provide corresponding positive results or adaptive algorithms for the achievable regimes, leaving this part of the story incomplete.**
>
> - Our main results are for the full-information setting. We included the bandit results as an addition because we thought it was relevant to provide the separation with full-information that sub-linear regret is not possible in high-dimension. However, there are indeed still many unanswered questions including the optimal rate of regret in the low-dimensional setting (there is no matching lower-bound as far as we know) that leave open many possibilities for future work.
>
> **Assumptions on separability**
>
> - Resolving what result can be achieved for general regularizer that we leave open is of great importance to better understand the limitations and strengths of FTRL (see answer to reviewer vyFS). Also note that our result holds for non-separable regularizers that are a multiplicative constant away from a separable regularizer.
>
> # Questions:
>
> **Q1: What is the main technical issue with extending the present proofs to non-separable regularizers? Is it the analog of Lemma 4.7?**
>
> - Exactly, the difficulty is in extending Lemma 4.7 on the quadratic-growth of the regulariser beyond the 1-dimensional case. In fact having a general result on the necessity for quadratic growth or some form of strong convexity to achieve optimal regret in the low-dimensional setting could be of independent interest.
>
> **Q2: Could you explain a bit more about the prior work in this domain? Have negative results of the flavor presented in this work appeared before in the literature?**
>
> - As far as we are aware, results on the failure of fixed regularization are novel in online learning. Algorithmic specific lower bounds (like the ones we have for specific regularizers - Propositions 4.1 and 4.5) have appeared in prior work (e.g. Theorems 3&4 in [1]).
>
> **Q3: Any insights on the algorithmic component to complement the impossibility results in Section 5?**
>
> - In the low-dimensional setting, the algorithm from [2] that achieves a regret bound of $\tilde{O}(T^{1-1/p} d^{1/p})$ is based on OMD with a carefully designed barrier regularizer. This result holds for all d but becomes loose as d increases. Our lower bound says that sub-linear regret is not possible in high enough dimension. It is unclear whether there are different optimal regret regimes within the low dimensional setting which would require an adaptive algorithm, it is even unclear if the regret bound of [2] is optimal in any dimension. As mentioned in the weakness section, there is still a lot to understand on the optimal rates of regret for the bandit setting.
>
> [1] Francesco Orabona and Dávid Pál. Scale-free online learning. Theoretical Computer Science, 716:50–69, 2018.
>
> [2] Thomas Kerdreux, Christophe Roux, Alexandre d’Aspremont, and Sebastian Pokutta. Linear bandits on uniformly convex sets. Journal of Machine Learning Research, 22(284):1–23, 2021.

---

> > ### Comment · Reviewer_5h3T · 2025-08-01
> >
> > Thanks. I don't remember seeing a note on the main result holding for non-separable regularizers that are a multiplicative constant away from a separable regularizer in the paper---if not would be good to add it. Overall I am happy to stick to my original assessment and recommend acceptance.

---

> > > ### Author Response · Authors · 2025-08-03
> > >
> > > There is a remark on the result holding for non-separable regularizers that are a multiplicative constant away from a separable regularizer in Remark 4.8 at lines 327-329.

---

### Official Review · Reviewer_vyFS · 2025-07-03

**Clarity:** 3
**Significance:** 2
**Originality:** 3
**Rating:** 5
**Confidence:** 3

**Summary:**

The paper studies online convex optimization over $l_p$ norm ball domains with $p > 2$. It investigates the role of regularization in Follow-the-Regularized-Leader (FTRL) and Online Mirror Descent algorithms and shows that the optimal separable regularizers for low- and high-dimensional settings are not equivalent, even up to time-varying scaling factors. The authors further prove that no time-invariant separable regularizer (up to time-varying scalars) can achieve an optimal anytime regret guarantee. However, they establish the existence of a time-varying separable regularizer that does provide such a guarantee. Finally, in the bandit feedback setting, they show that for action sets of $l_p$ norm balls with $p \ge 1$ and linear losses, linear regret is unavoidable when the dimension is sufficiently large.

**Questions:**

- Theorem 5.1 claims that linear regret is unavoidable in the high-dimensional setting. However, this conclusion would also follow from any dimension-dependent lower bound of the form $\Omega(d^a T^b)$. Does the lower bound instance presented in the paper yield any nontrivial results for $l_p$ norm balls that are stronger than those known for the multi-armed bandit setting? This is particularly relevant since, for $p \in [1, 2]$, the regret bounds are already known to be the same.
- Have the authors considered using fixed log-barrier regularizers, which are not part of the class of separable regularizers?

**Ethical Concerns:**

["NO or VERY MINOR ethics concerns only"]

**Limitations:**

yes

**Quality:**

3

**Strengths And Weaknesses:**

Strengths:
- The paper tackles the universality of FTRL algorithms with fixed regularizers, addressing a fundamental and challenging problem. It provides a negative result for the subclass of separable regularizers.
- The analysis and proof techniques are rigorous and carefully executed.
Weaknesses:
- While separable regularizers are indeed widely used, they are not a fundamentally necessary class. As a result, the negative result primarily highlights the need for more sophisticated fixed regularizers rather than a limitation of FTRL itself.
- The negative result on linear bandits appears somewhat trivial, as it essentially demonstrates the dependence of regret on the dimensionality.

---

> ### Author Rebuttal · Authors · 2025-07-27
>
> We thank the reviewer for their positive and insightful feedback. We address your comments, suggestions and questions below.
>
> # Weaknesses
>
> **While separable regularizers are indeed widely used, they are not a fundamentally necessary class. As a result, the negative result primarily highlights the need for more sophisticated fixed regularizers rather than a limitation of FTRL itself.**
> - This is correct. But in fact, we view this as one of the main messages of our work. Our results demonstrate that existing regularizers impose intrinsic limitations on FTRL and open up an interesting avenue of research to discover these more sophisticated alternatives that potentially give algorithms that are fundamentally different. Also note that our main result holds for non-separable regularizers that are a multiplicative constant away from being separable.
>
> **The negative result on linear bandits appears somewhat trivial, as it essentially demonstrates the dependence of regret on the dimensionality.**
> - Yes, it essentially demonstrates that the dependence of the low-dimensional regret on the dimensionality is maintained as the dimension increases but this is not always obvious. For example, in the full-information setting that we consider in most of our work, this is not the case. While it may be more expected that it is the case in the harder bandit feedback setting, proving it formally in the way that we do is not trivial.
>
> # Questions
>
> **Q1: Theorem 5.1 claims that linear regret is unavoidable in the high-dimensional setting. However, this conclusion would also follow from any dimension-dependent lower bound of the form $\Omega(d^a T^b)$. Does the lower bound instance presented in the paper yield any nontrivial results for norm balls that are stronger than those known for the multi-armed bandit setting? This is particularly relevant since, for $p \in [1,2]$, the regret bounds are already known to be the same.**
> - This conclusion would indeed also follow from any dimension-dependent lower bound of the form $\Omega(d^a T^b)$ but usually lower bounds of this form only hold in the low-dimensional setting (e.g. in [1], their lower bound Theorem 4 only holds for $T \geq d^{2/(1-q/2)}$ and does not give linear regret in high dimension). As far as we are aware, for $p \in [1,2]$, the regret bounds are known to be the same as for the multi-armed bandit setting only in the low-dimensional setting, making our results non-trivial in the high-dimensional setting. We will add a remark to clarify this in the final version.
>
> **Q2: Have the authors considered using fixed log-barrier regularizers, which are not part of the class of separable regularizers?**
> - We did not consider specific non-separable log-barrier regularizers but it is unclear how we could use it since they do not satisfy uniform convexity properties that allow us to obtain regret guarantees. However, we would be really happy to receive any pointer in this direction.
>
>
> [1] Sébastien Bubeck, Michael Cohen, and Yuanzhi Li. Sparsity, variance and curvature in multi-armed bandits. In Proceedings of Algorithmic Learning Theory, Proceedings of Machine Learning Research, pages 111–127, 2018.

---

> > ### Comment · Reviewer_vyFS · 2025-08-04
> >
> > Thanks for the insightful response. I suggest highlighting usage of non-separable regularizers as a future direction in conclusion section. Other than that this is a good paper.

---

### Official Review · Reviewer_SbZL · 2025-07-03

**Clarity:** 3
**Significance:** 4
**Originality:** 4
**Rating:** 6
**Confidence:** 4

**Summary:**

The authors established that adaptive regularization is essential for achieving anytime optimal regret for FTRL on high-dimensional $\ell_p$-balls ($p>2$). It demonstrates a shift in optimal regret rates between low-dimensional ($d \leq T$, rate $\mathcal{O}(\sqrt{Td^{1-2/p}})$) and high-dimensional regimes ($d > T$, rate $\mathcal{O}(T^{1-1/p})$), proving that FTRL with time-varying regularization can achieve optimality across all dimensions without prior knowledge of $T$ or the regime. Crucially, the authors show that no fixed separable regularizer (e.g., $\|x\|_r^r$) can be optimal in both regimes simultaneously, resolving an open problem and highlighting the necessity of adaptivity. They further establish that agents suffer unavoidable linear regret under linear bandits in sufficiently high dimensions, underscoring fundamental limitations beyond full-information settings. This paper significantly advances the understanding of geometry-aware online learning.

**Questions:**

- Whether the assumption that $0$ is an interior point of the action set $V$ is fundamentally necessary for this paper? Also, do the authors have some intuition about whether the $\ell_p$-ball structure can be generalized to certain more general convex sets to still have similar negative results for separable regularizers?

**Ethical Concerns:**

["NO or VERY MINOR ethics concerns only"]

**Final Justification:**

This paper is both technically highly nontrivial and significant at a message level. It brings new understanding of the non-asymptotic scaling behavior between the interplay of feature dimension and time horizon for FTPL. Therefore, my final decision is strong accept.

**Limitations:**

Yes.

**Paper Formatting Concerns:**

N/A.

**Quality:**

4

**Strengths And Weaknesses:**

- The regularizer in equation (2) is novel and might be of independent interest.
- The understanding of the necessity of adaptive regularization is clear, clean, and novel.
- This paper leaves a clear open problem on whether a non-separable static regularizer can achieve the same guarantees as the adaptive regularizer in the case of $\ell_p$-balls.

---

> ### Author Rebuttal · Authors · 2025-07-27
>
> We thank the reviewer for their positive feedback. We address your questions below.
>
> **Q1: Whether the assumption that 0 is an interior point of the action set is fundamentally necessary for this paper?**
>
> - We could consider translating the lp-ball away from 0, then our results will hold for regularisers satisfying the conditions we need on this translated ball (e.g. reaches its minimum in the centre of the translated lp-ball). We will add a remark on this in the final version.
>
> **Q2: Do the authors have some intuition about whether the -ball structure can be generalized to certain more general convex sets to still have similar negative results for separable regularizers?**
> - This is an interesting question. What we need for Theorem 4.6 (our main result on the failure of separable regularization) to hold is
> 1. that in the 1-dimensional case, the optimal regret requires quadratic growth of the regularizer.
> 2. In the general d-dimensional case, quadratic growth of the regularizer leads to sub-optimal regret.
>
> Point 1. will hold for any symmetric action set. The question then becomes whether we can show 2. for these more general convex sets, which is not clear We would also require that there is a similar shift in the optimal rate of regret between dimension-regimes, which is not necessarily obvious.
>
> How general this phenomenon might be is likely related to the concept of quadratically convex sets [1] for which several results are known on the posibility of getting regret that grows as $\sqrt{T}$ in the low-dimensional case, while it is likely that the regret is better in the high dimensional case (when taking into account dependence on other quantities like dimension). This is an interesting direction of future research. We will also add a remark on this in the final version.
>
> [1] Daniel Levy and John C Duchi. Necessary and sufficient geometries for gradient methods. In Advances in Neural Information Processing Systems, volume 32, 2019.

---

> > ### Comment · Reviewer_SbZL · 2025-08-01
> >
> > I thank the authors for their insightful rebuttal and decide the keep my positive evaluation unchanged.

---

### Decision · Program_Chairs · 2025-09-17

**Decision:**

Accept (spotlight)

**Comment:**

(a) Summary: The paper studies the problem of achieving optimal anytime regret in online convex optimization over $l_p$-balls, for p > 2. The paper has two main contributions:
  - A positive result: showing that the FTRL algorithm with adaptive regularization achieves anytime optimality, leading to the question of whether optimality is achievable with a fixed regularizer.
  - Accompanying negative results: anytime optimality cannot be achieved by any fixed, separable regularizer.

Additionally, the paper extends its investigation to the bandit feedback setting, showing that for linear bandits on $l_p$-balls, linear regret is unavoidable in sufficiently high dimensions.

(b) Strengths:
  - The paper is well-motivated and studies a fundamental problem, namely the necessity of adaptive regularization for anytime optimality in OCO over lp balls.
  - The results are sounds and the analysis is rigorous. The optimal algorithm is intuitive, and the lower bound is novel.
  - The paper is well-written.

(c) Weaknesses:
  - The main lower bound applies only to separable regularizers, leaving the broader question of whether a non-separable fixed regularizer could achieve anytime optimality open.

(d) Rationale: I recommend acceptance because it is a technically solid paper with clear motivations and interesting results.

(e) Summary of Rebuttal and Discussion
The authors provided detailed responses to all reviewer questions, and the responses were positively received. All reviewers maintained their original positive scores. Main Points Raised by Reviewers:
  - Reviewers vyFS and 5h3T raised concerns about the limitation to separable regularizers, and the completeness of the section on bandit feedback.

How Authors Addressed the Points:
  - On separable regularizers: The authors agreed this was a limitation but they viewed it as a key message of their work, as separable regularizers are common, and the lower bound motivates the search for non-separable ones. They also clarified (in response to 5h3T) that their result already covers non-separable regularizers that are a multiplicative constant away from a separable one.
  - On the bandit section: They acknowledged to 5h3T that this section was an addition to highlight the separation from the full-information setting and that many open questions remain for future work.